# Edit Distance Robust Watermarks for Language Models

**Noah Golowich**
nzg@mit.edu
MIT

**Ankur Moitra**
moitra@mit.edu
MIT

## Abstract

Motivated by the problem of detecting AI-generated text, we consider the problem of watermarking the output of language models with provable guarantees. We aim for watermarks which satisfy: (a) *undetectability*, a cryptographic notion introduced in [CGZ24] which stipulates that it is computationally hard to distinguish watermarked language model outputs from the model's actual output distribution; and (b) *robustness* to channels which introduce a constant fraction of *adversarial* insertions, substitutions, and deletions to the watermarked text. Earlier schemes could only handle stochastic substitutions and deletions, and thus we are aiming for a more natural and appealing robustness guarantee that holds with respect to *edit distance*.

Our main result is a watermarking scheme which achieves both undetectability and robustness to edits when the alphabet size for the language model is allowed to grow as a polynomial in the security parameter. To derive such a scheme, we follow an approach introduced in [CG24], which proceeds via first constructing *pseudorandom codes* satisfying undetectability and robustness properties analogous to those above; our key idea is to handle adversarial insertions and deletions by interpreting the symbols as indices into the codeword, which we call *indexing pseudorandom codes*. Additionally, our codes rely on weaker computational assumptions than used in previous work. Then we show that there is a generic transformation from such codes over large alphabets to watermarking schemes for *arbitrary* language models.

## 1 Introduction

The rapid increase in AI-generated content represents a significant challenge for numerous societal institutions, ranging from education to social media. For instance, the ability of large language models such as GPT-4 to generate long bodies of text such as essays raises the possibility of increasing amounts of plagiarism, and the proliferation of AI-generated text, images, and videos on social media presents challenges regarding large-scale manipulation of online audiences. An important tool at our disposal in preventing the misuse of such content is *watermarking schemes*, which are procedures that embed hidden patterns in AI-produced content using a *secret key*. Watermarking schemes allow a *detection algorithm* with the aid of the secret key to determine with high probability that the content was produced by the AI model. They do so without noticeably altering the content from the perspective of any algorithm not possessing the secret key.

Despite a number of recently proposed watermarking schemes, taking both a theoretical perspective (e.g., [CGZ24, CG24, Zam24, FGJ+23]) as well as a more empirical one (e.g., [KGW+23, KGW+24, KTHL23]), a crucial challenge that remains elusive is that of ensuring the watermark be *robust* to adversaries which can modify the generated content. A watermarking scheme is of little use if it is too easy to change the model's output so as to remove the watermark. On the other hand, a sufficiently

38th Conference on Neural Information Processing Systems (NeurIPS 2024).

resourceful adversary can simply train their own unwatermarked model. Thus it is necessary to strike a balance in terms of the power of adversaries to which a watermarking scheme enjoys robustness. In this paper, we give the first watermarking schemes with provable robustness to adversaries that make a constant fraction of arbitrary substitutions, insertions, and deletions to the output. This is a substantial improvement over previous schemes [CG24], which could only tolerate a constant fraction of adversarial substitutions or a constant fraction of i.i.d. deletions under additional stochasticity assumptions on the language model. Our results therefore represent progress towards the overarching goal of constructing watermarking schemes which are robust to resource-limited adversaries.

## 1.1 Setup: watermarking schemes

We consider the task of watermarking the output of a *language model* Model, which is simply defined as a mapping from a sequence of *tokens* $t_1, \ldots, t_{i-1}$ to a distribution over the next token.[1] Though we use the terminology "language model" and "text" throughout the paper, our framework can be used to model a host of autoregressive models including those for generation of images, audio, and videos (e.g., [YXK$^+$22, TJY$^+$24]) in addition to text. As this paper is theoretical in nature, we work only with this abstraction of an autoregressive model, keeping in mind that single tokens could represent, e.g., sequences of letters [SHB16] or discretized patches of an image [YLK$^+$22, RvdOV19].[2]

A *watermarking scheme* $\mathcal{W}$ for Model consists of a tuple of efficient algorithms,

$$\mathcal{W} = (\mathsf{Setup}, \mathsf{Wat}, \mathsf{Detect})$$

where Setup generates a secret key at random, Wat uses Model and the secret key to produce a sequence of text with a watermark embedded in it, and Detect uses the secret key to determine whether a given sequence of text is watermarked. Moreover, Detect has no knowledge of Model. $\mathcal{W}$ should satisfy the following three properties:

- *Undetectability [CGZ24]:* Text generated from the watermarking protocol Wat should be computationally indistinguishable to text generated by the true model Model, to any polynomial-time algorithm which can repeatedly query either Wat or Model. (Other notions, namely *distortion-freeness* [KTHL23], have been considered in place of undetectability, but are significantly weaker; see Example A.1.)

- *Soundness:* Any fixed sequence of text (not produced by Wat) should not be detected as watermarked, with high probability (over the generation of the secret key).

- **Edit-Robustness:** A sequence which can be obtained by calling the watermarking procedure Wat and then making a constant fraction of adversarial *edits* (i.e., substitutions, insertions, and deletions) to its output will still be detected as watermarked by Detect with high probability.

Whereas several existing watermarking schemes achieve undetectability and soundness, the main contribution of our work is to achieve the notion of *edit-robustness* given above. Robustness, broadly construed, has long been recognized as essential for constructing useful watermarking schemes [ARC$^+$01, ARH$^+$02], yet nearly all of the existing provable guarantees on watermarking do not yield robustness to any constant fraction of edits. We remark that some works (e.g., [CGZ24]) can handle a *subconstant* fraction of edits. However, we argue that the right notion of adversary is that which makes a *constant* fraction of edits: there is a relatively low bar for a malicious adversary attempting to remove the watermark by changing the text at a constant rate while avoiding significant quality degradations, such as randomly replacing $5\%$ of words with a synonym. Moreover, even if an adversary is not actively trying to remove a watermark, it is reasonable to expect that an "honest" editor who is merely trying to improve the text's quality will introduce edits at a constant rate. The lens of coding theory provides further motivation for recovering from such channels that introduce a constant fraction of errors: doing so is a longstanding gold standard in the field [GRS19].

---

[1]Note also that it is typically assumed that Model takes as input a prompt; for simplicity, we absorb such a prompt in the definition of Model, which technically introduces a different model for each possible prompt. Nothing in our results changes if we instead explicitly allow Model to take as input a prompt. (In particular, our watermarking detection algorithms do not assume any knowledge of Model, and therefore of the prompt.)

[2]In the setting of text, transforming a token sequence to readable text (i.e., sequences of letters) and then back to tokens may not necessarily recover the original token sequence. Any errors induced via this transformation would likely be of small edit distance, and therefore we incorporate them as part of the adversary for which our watermarks have robustness. See also [KGW$^+$23, Section 7] for related discussion on "tokenization attacks".

The only prior work in the literature which obtains watermarking schemes with provable robustness guarantees against such "constant-rate" attacks was [CG24], which constructed watermarking schemes robust to a constant fraction of deletions and substitutions. Moreover, in order to handle any deletions at all, [CG24] needs to assume that: (a) the deletions and substitutions are made *independently* at each position with some fixed probability; and (b) the language model is equivalent to a binary symmetric channel in a certain sense, a strong assumption which is contradicted by decades of study in linguistics, which posit that earlier words strongly influence the distribution of subsequent words [Bib09].[3]

Finally, we emphasize that the task of obtaining robustness against adversaries that can make a constant fraction of insertions, deletions, and substitutions, as opposed to merely substitutions, is well-motivated by an extensive line of work on *edit distance* (e.g., [OR07, AKO10, AO11, AN20], amongst many others). The edit distance between two strings is the minimum number of insertions, deletions, and substitutions needed to transform one string into the other. Edit distance has found numerous applications in areas ranging from computational biology to information retrieval [Nav01]. This fundamental role of edit distance results from the fact that many natural processes induce small edit-distance perturbations on strings: for instance, mutations of DNA can involve insertions or deletions as well as substitutions, as do the changes people typically make to documents. Procedures that make such changes therefore represent a reasonable adversary in our present setting of watermarking.

## 1.2 Main result

Our main result states that watermarking schemes with all of the aforementioned properties exist under a standard cryptographic assumption:

**Theorem 1.1** (Informal version of Theorem E.2). *Suppose that local weak PRFs exist (per Assumption 3.1). Then for any security parameter $\lambda \in \mathbb{N}$, there is a watermarking scheme over an alphabet of size* $\text{poly}(\lambda)$ *for model outputs of length* $\text{poly}(\lambda)$*, which is sound, undetectable with respect to algorithms running in time* $\text{poly}(\lambda)$*, and robust to a constant fraction of substitutions, insertions, and deletions when the sequence of generated text has "entropy rate" at least some constant.*

As is typical in cryptographic settings, the guarantee of Theorem 1.1 involves a security parameter $\lambda$, which controls the power of a distinguishing algorithm (in the context of undetectability) as well as the failure probability of soundness and edit-robustness (which are $\text{negl}(\lambda)$; see the formal version in Appendix E). The requirement that the sequence of generated text have constant entropy rate (meaning that on average, the tokens are generated from conditional distributions which have entropy at least a constant fraction of the maximum possible entropy) is easily seen to be necessary,[4] and lower bounds on the entropy are standard amongst essentially all watermarking schemes, even without any robustness requirement [CGZ24, FGJ+23, CG24, Zam24]. Indeed, watermarking a near-deterministic language model is impossible since it can only produce a small number of outputs; the entropy assumption quantifies the extent to which the language model is near-deterministic.

One potential limitation of Theorem 1.1 is the requirement that the alphabet size of the language model (i.e., number of tokens) grow as a polynomial in $\lambda$. This limitation is mitigated by the following observations: first, in many domains, the number of tokens used by existing (or future) tokenizers may already be quite large, i.e., at least in the thousands [SHB16, YLK+22]; second, in practice one could use various heuristics to increase the number of tokens, such as by grouping together consecutive sequences of tokens to create larger alphabets of "mega-tokens". Finally, again invoking the language of coding theory, a common approach to constructing good codes is via *concatenation*, which combine an *outer code* with large alphabet together with an *inner code* that encodes individual symbols of the large alphabet using a smaller alphabet. Theorem 1.1 (as well as Theorem 4.1, discussed below) could play the role of the outer code in such a concatenation strategy for watermarking schemes; finding the analogue of an inner code to decrease the alphabet size represents an important direction for future work.

---

[3]We remark that, even for the case of substitutions, the result of [CG24] (Lemma 23 & Theorem 5 within) only states robustness against i.i.d. substitutions, i.e., a binary symmetric channel. Nevertheless, a straightforward modification to the proof of [CG24, Theorem 5] allows one to establish robustness a constant fraction of *adversarial* substitutions. However, entirely different techniques are needed to handle the case where insertions and deletions are also allowed.

[4]If only an $\alpha = o(1)$ fraction of tokens have nonzero entropy, then an adversary could simply replace those tokens with any fixed token and thus remove all entropy.

**Roadmap of techniques.** The proof of Theorem 1.1 proceeds via constructing new *pseudorandom codes (PRCs)*, which are cryptographic objects originally introduced by [CG24] to derive watermarking schemes robust to i.i.d. random (i.e., non-adversarial) subsitutions and deletions. Roughly speaking, a pseuorandom code is an error-correcting code equipped with a secret key used for encoding and decoding, which looks random to any polynomial-time algorithm that does not hold a secret key. We establish the following ingredients pertaining to PRCs:

- First, we design a a PRC over the binary alphabet which is robust to a constant fraction of adversarial substitutions, under Assumption 3.1 (Theorem 3.2; see Section 3). This assumption is, qualitatively speaking, *weaker* than the cryptographic assumptions in [CG24].

- Next, we give a generic reduction which, given any PRC over the binary alphabet robust to substitutions, yields a PRC over a polynomial-sized alphabet robust to any constant fraction of substitutions, insertions, and deletions (Theorem 4.1; see Section 4). A central idea in this latter PRC is to interpret symbols of the larger alphabet as *indices* into codewords of the former PRC; hence, we call it an *indexing PRC*.

- Finally, we establish a generic reduction which converts any PRC over a larger alphabet robust to substitutions, insertions, and deletions to a watermarking scheme with analogous robustness properties (Theorem E.1; see Section 5).

Theorem 1.1 follows by composing the components above. These components have the advantage of being modular in nature, meaning that one could, for instance, plug in the substitution PRCs of [CG24] in place of the first item to obtain a guarantee analogous to Theorem 1.1 but under the (qualitatively stronger) cryptographic assumptions of [CG24].

In order to handle deletions, [CG24] uses a type of PRC they call a *majority code*. This code is only robust when the substitutions and deletions are made in an i.i.d. manner. In their reduction that converts a PRC to a watermarking scheme, the language model may induce certain errors on the PRC codewords. Since these errors may not be correctable by the majority code unless they are assumed to be i.i.d., [CG24] need to assume that the language model is equivalent to a BSC. Our use of indexing PRCs avoids this significant shortcoming.

## 2 Preliminaries

We begin with the definition of a *pseudorandom code (PRC)*, as introduced in [CG24]. A pseudorandom code is defined over an *alphabet* $\Sigma$, which is simply a finite set. We denote the set of all finite-length strings over $\Sigma$ by $\Sigma^\star$. As is typical when defining cryptographic primitives, our pseudorandom codes depend on a *security parameter* $\lambda \in \mathbb{N}$; as $\lambda$ is increased, the amount of security afforded by the PRC also increases. A function $f : \mathbb{N} \to \mathbb{R}_{\geq 0}$ is called *negligible* if for any $C > 0$, there exists $\lambda_0$ so that $f(\lambda) \leq \lambda^{-C}$ for all $\lambda > \lambda_0$. We use $\mathsf{negl}(\lambda)$ to denote a negligible function, whose precise value can change from line to line.

Given an alphabet $\Sigma$, a *channel* is a mapping $\mathcal{E}$ which associates to any $x \in \Sigma^\star$ a distribution $\mathcal{E}(x)$ over $\Sigma^\star$. With slight abuse of notation, we will often let $\mathcal{E}(x)$ denote a random variable $y$ drawn from $\mathcal{E}(x)$: for instance, when we write a statement of the form "with probability $1 - \mathsf{negl}(n)$, $D_{\mathsf{Ham}}(x, \mathcal{E}(x)) \leq pn$", we mean that $D_{\mathsf{Ham}}(x, y) \leq pn$ with probability $1 - \mathsf{negl}(n)$ over $y \sim \mathcal{E}(x)$.

We primarily focus on *secret key* PRCs for simplicity; our arguments (in particular, those regarding edit-robust PRCs) apply identically to the case of public-key PRCs [CG24, Definition 2], though for simplicity we focus solely on secret-key PRCs. A secret-key PRC may be viewed as a variant of a secret-key encryption scheme in which the ciphertext enjoys certain robustness properties to adversarial perturbations. Formally, a PRC is specified by functions KeyGen, Encode, Decode, which behave as follows: KeyGen outputs a secret key, Encode uses the secret key to "encrypt" a message, and Decode uses the secret key to "decode" a string. If the string passed to Decode is the output of Encode, perhaps with a bounded number of adversarial perturbations, then Decode should output the message originally passed to Encode (*robustness*). Outputs of Encode should also look random to polynomial-time algorithms (*undetectability*), and Decode should almost always fail when given any fixed string (i.e., not an output of Encode) as input. These definitions are formalized below:

**Definition 2.1** (Secret-key Pseudorandom code (PRC)). Let $\lambda \in \mathbb{N}$ denote a security parameter, and suppose that for each $\lambda \in \mathbb{N}$ an associated alphabet $\Sigma(\lambda)$ is given. Moreover, suppose that

for each $\lambda$, a collection $\mathscr{E}(\lambda)$ of channels $\mathcal{E} : \Sigma(\lambda)^\star \to \Delta(\Sigma(\lambda)^\star)$ is given. A *secret-key pseudorandom code (PRC)* with robustness to $\mathscr{E}$ is a triple of probabilistic polynomial-time algorithms (KeyGen, Encode, Decode) satisfying the following:

- For some functions $n, k, \ell_{\sf sk} : \mathbb{N} \to \mathbb{N}$, for all $\lambda \in \mathbb{N}$, we have $\mathsf{KeyGen}(1^\lambda) \in \{0,1\}^{\ell_{\sf sk}(\lambda)}$, $\mathsf{Encode} : \{1^\lambda\} \times \{0,1\}^{\ell_{\sf sk}(\lambda)} \times \Sigma^{k(\lambda)} \to \Sigma^{n(\lambda)}$, and $\mathsf{Decode} : \{1^\lambda\} \times \{0,1\}^{\ell_{\sf sk}(\lambda)} \times \Sigma^\star \to \Sigma^{k(\lambda)} \cup \{\bot\}$.

- For any $\lambda \in \mathbb{N}$ and message $\mathsf{m} \in \Sigma^{k(\lambda)}$, the code is *robust* to any channel $\mathcal{E} \in \mathscr{E}(\lambda)$, which means that:

$$\Pr_{\mathsf{sk} \leftarrow \mathsf{KeyGen}(1^\lambda)} \big( \mathsf{Decode}(1^\lambda, \mathsf{sk}, \mathcal{E}(x)) = \mathsf{m} \mid x \leftarrow \mathsf{Encode}(1^\lambda, \mathsf{sk}, \mathsf{m}) \big) \geq 1 - \mathsf{negl}(\lambda).$$

Moreover, for any fixed $y \in \Sigma^\star$, the code is *sound*, which means that:

$$\Pr_{\mathsf{sk} \leftarrow \mathsf{KeyGen}(1^\lambda)} \big( \mathsf{Decode}(1^\lambda, \mathsf{sk}, y) = \bot \big) \geq 1 - \mathsf{negl}(\lambda).$$

- The code is *undetectable* which means that: for any $\lambda \in \mathbb{N}$ for any probabilistic polynomial-time adversary $\mathsf{Adv}$,

$$\left| \Pr_{\mathsf{sk} \leftarrow \mathsf{KeyGen}(1^\lambda)} \big( \mathsf{Adv}^{\mathsf{Encode}(1^\lambda, \mathsf{sk}, \cdot)}(1^\lambda) = 1 \big) - \Pr_{\mathsf{sk} \leftarrow \mathsf{KeyGen}(1^\lambda)} \big( \mathsf{Adv}^{\mathcal{U}}(1^\lambda) = 1 \big) \right| \leq \mathsf{negl}(\lambda), \ (1)$$

where $\mathcal{U}$ denotes an oracle which responds with a freshly drawn uniformly random string in $\Sigma^{n(\lambda)}$ on each call (to any input).

The *block length* of the PRC is defined to be $n(\lambda)$. For all of our pseudorandom codes, we will have $n(\lambda), k(\lambda), \ell_{\sf sk}(\lambda) \leq \mathsf{poly}(\lambda)$. Moreover, we take as a convention that $n(\lambda) \geq \lambda$ for all $\lambda$ (this may be ensured without loss of generality by rescaling $n(\lambda)$). Our focus will primarily be on *zero-bit PRCs*, which are particularly useful for watermarking language model outputs.

**Definition 2.2** (Zero-bit PRC). A *zero-bit* PRC is one for which $k(\lambda) = 0$ for all $\lambda$, i.e., the only possible message is $\mathsf{m} = \emptyset$.

[CG24, Section 6] shows a generic reduction that converts any zero-bit PRC into a general PRC with constant rate, meaning that $k(\lambda)/n(\lambda) = \Omega(1)$. We remark that the same reduction can be applied to our PRCs with edit robustness, though we do not pursue this direction further in this paper.

**Definition 2.3** (Substitution-bounded). For $p \in (0,1)$, a channel $\mathcal{E}$ over alphabet $\Sigma$ is *$p$-substitution-bounded* if for any $n \in \mathbb{N}$, $y \in \Sigma^n$, for $z \sim \mathcal{E}(y)$, $D_{\mathsf{Ham}}(y, z) \leq pn$ holds almost surely.

**Definition 2.4** (Edit-bounded channel). Fix an alphabet $\Sigma$ together with $p \in (0,1)$. A channel $\mathcal{E}$ over $\Sigma$ is defined to be *$p$-edit-bounded* if for any $x \in \Sigma^\star$ with $n := |x|$, $y \sim \mathcal{E}(x)$ may be obtained from $x$ by applying a total of at most $pn$ substitutions, insertions, and deletions, almost surely. Moreover, there exists a probabilistic polynomial-time algorithm which, given $x$, outputs a sample $y \sim \mathcal{E}(x)$.[5]

## 3 Secret-key substitution PRCs from weaker assumptions

In this section, we discuss a new construction of binary PRCs for substitution channels. Though such PRCs were also obtained by [CG24], the codes in [CG24] relied on relatively strong average-case hardness assumptions, in the sense that they imply the existence of public-key cryptography (i.e., in the context of Impagliazzo's Five Worlds [Imp95], they imply primitives in "Cryptomania"). In contrast, our construction relies only on the hardness of the existence of a family of pseudorandom functions that enjoys a certain locality property; such an assumption is generally believed to be weaker than the ones in [CG24], in the sense that it is only known to yield cryptographic primitives in "Minicrypt", though we are not aware of a formal separation.

We first recall the definition of pseudorandom function (PRF) families, which are PRF families for which the adversary can only query the pseudorandom function at a uniformly random input $x$.

---

[5] The computational efficiency of $\mathcal{E}$ is only needed for our watermarking schemes. It is not necessary to obtain edit-robust PRCs; see Remark D.2

**Definition 3.1** (Weak PRF family). Fix functions $\ell(\lambda), n(\lambda) : \mathbb{N} \to \mathbb{N}$ of a security parameter $\lambda$, and a collection of functions $\{F_s : \{0,1\}^{n(\lambda)} \to \{0,1\}\}$, indexed by $s \in \{0,1\}^{\ell(\lambda)}$, for $\lambda \in \mathbb{N}$. We say that the collection $\{F_s\}_s$ is a *weak pseudorandom function family (weak PRF)* if for every probabilistic polynomial-time algorithm Adv which outputs a single bit (i.e., 0 or 1), it holds that

$$\left| \mathbb{E}_{s \sim \{0,1\}^{\ell(\lambda)}} \left[ \widetilde{\mathsf{Adv}}^{F_s(\cdot)}(1^{n(\lambda)}) \right] - \mathbb{E}_{F_{\mathsf{Unif}}} \left[ \widetilde{\mathsf{Adv}}^{F_{\mathsf{Unif}}}(1^{n(\lambda)}) \right] \right| \leq \mathsf{negl}(\lambda), \tag{2}$$

where $\widetilde{\mathsf{Adv}}^G$, for a mapping $G : \{0,1\}^{n(\lambda)} \to \{0,1\}$ means that Adv can make calls to the function $G$, and for each call receives a tuple $(x, G)$, for $x \sim \mathrm{Unif}(\{0,1\}^n)$. (In particular, the tilde refers to the fact that Adv can only call $G(x)$ on a *uniformly chosen $x$*.) Moreover, $F_{\mathsf{Unif}} : \{0,1\}^n \to \{0,1\}$ denotes a uniformly random function. We will often refer to the function $G$ that Adv can make queries to as the *oracle* Adv has access to. Given $q \in [0, 1/2)$, we say that $\{F_s\}_s$ is a *weak PRF family with noise level $q$* if (2) holds where each call to $F_s(\cdot)$ by Adv returns $F_s(x) \oplus e$ where $x \sim \mathrm{Unif}(\{0,1\}^n)$ and $e \sim \mathrm{Ber}(q)$.

Our new construction of PRCs is based off of *local (weak) PRFs*, which are PRFs for which the output depends on a small number of input bits.

**Definition 3.2** (Local function family). Let $\tau \in \mathbb{N}$. A family of functions $\{F_s : \{0,1\}^n \to \{0,1\}\}$ indexed by $s$ is defined to be $\tau$-*local* if for each $s$, $F_s(x)$ only depends on at most $\tau$ bits of $x$, i.e., for each $s$ there are distinct indices $j_1, \ldots, j_\tau \in [n]$ together with a function $G_s : \{0,1\}^\tau \to \{0,1\}$ so that $F_s(x) = G_s(x_{j_1}, \ldots, x_{j_\tau})$ for all $x \in \{0,1\}^n$.

Our main computational assumption is the existence of a weak PRF family which is $\tau$-local for $\tau$ of *logarithmic* size:

**Assumption 3.1** (Local Weak PRFs). *For some functions $\ell(\lambda), n(\lambda), \tau(\lambda) : \mathbb{N} \to \mathbb{N}$ with $\ell(\lambda), n(\lambda) \leq \mathrm{poly}(\lambda)$ and $\tau(\lambda) \leq \log n(\lambda)$, there exists a weak PRF family $\{F_s : \{0,1\}^{n(\lambda)} \to \{0,1\}\}_{s \in \{0,1\}^{\ell(\lambda)}}$ for some noise level $q < 1/2$ which is $\tau(\lambda)$-local, for each $\lambda \in \mathbb{N}$.*

In Appendix C.2, we discuss how Assumption 3.1 follows from standard average-case hardness assumptions, notably the hardness of learning $\log(n)$-juntas over $\mathrm{Unif}(\{0,1\}^n)$. As specific examples, either hardness of the $\log(n)$-sparse noisy parity problem [FGKP09, BFJ$^+$94, GRV11, Val15] or hardness of weakly learning a particular family of functions presented in [BFKL94] (see also [Blu03]) implies Assumption 3.1.

**The PRC construction.**  Our construction of PRCs based on Assumption 3.1 is presented in Algorithm 2: given a function family $\mathcal{F}$ satisfying Assumption 3.1 with noise level $q$ together with some $p < 1/2$ representing the maximum fraction of substitutions to correct, we construct $\mathsf{PRF\text{-}PRC}[\mathcal{F}, p, q] = (\mathsf{KeyGen}, \mathsf{Encode}, \mathsf{Decode})$ as follows. The construction depends on some parameters $N(\lambda), m(\lambda) \leq \mathrm{poly}(\lambda)$, specified in (3):

- $\mathsf{KeyGen}(1^\lambda)$ chooses a function in $\mathcal{F}$ (indexed by $s$) together with a uniformly random $z \sim \mathrm{Unif}(\{0,1\}^{N(\lambda)})$ and a uniform permutation $\pi : [N(\lambda)] \to [N(\lambda)]$, and returns $\mathsf{sk} = (s, z, \pi)$.

- $\mathsf{Encode}(1^\lambda, (s, z, \pi), \emptyset)$ draws $m(\lambda)$ uniformly random elements of $\{0,1\}^{n(\lambda)}$, $x_1, \ldots, x_{m(\lambda)}$, applies $F_s$ to each of them and flips the result with probability $q$ to obtain bits $(w_1, \ldots, w_{m(\lambda)})$, and perturbs the concatenation $((x_i, w_i))_{i \in [m(\lambda)]}$ according to $z, \pi$ as on Line 9.

- $\mathsf{Decode}(1^\lambda, (s, z, \pi), y)$ first "unperturbs" $y$ to obtain a string $((x_i, w_i))_{i \in [m(\lambda)]}$ as in Encode, and then outputs $\emptyset$ if $\sum_{j=1}^{m(\lambda)} \mathbb{1}\{w_j = F_s(x_j)\}$ is above a threshold; otherwise, it outputs $\perp$.

**Theorem 3.2.** *Given $p, q < 1/2$ and a function family $\mathcal{F}$ together with noise level $q$ satisfying Assumption 3.1, then $\mathsf{PRF\text{-}PRC}[\mathcal{F}, p, q]$ (Algorithm 2) is a zero-bit binary-alphabet secret-key PRC (per Definition 2.2) with robustness to all $p$-bounded substitution channels.*

The proof of Theorem 3.2 is given in Appendix C.3.

## 4   From substitution PRCs to edit-robust PRCs

In this section, we discuss our construction of PRCs which are robust to edit-bounded channels. To do so, we reduce to PRCs robust to substitution-bounded channels. Suppose we are given a PRC

$\mathsf{PRC}_{\mathsf{Sub}}$ with block length $n(\lambda)$ over the binary alphabet which is robust to any $(1/2 - p_0)$-bounded substitution channel, for $p_0 \in (0, 1/2)$. Given a parameter $\rho \geq 1$, we construct an *indexing PRC*, $\mathsf{PRC}_{\mathsf{Idx}}[\mathsf{PRC}_{\mathsf{Sub}}, \rho]$ (in Algorithm 1), which is robust to any $p$-edit-bounded channel, where the parameter $p$ depends on $p_0, \rho$ in a manner that will be explained below. The code $\mathsf{PRC}_{\mathsf{Idx}}[\mathsf{PRC}_{\mathsf{Sub}}, \rho]$ has polynomially large alphabet $\Sigma(\lambda) := [q(\lambda)]$, where $q(\lambda) = \rho \cdot n(\lambda)$. We denote the block length of $\mathsf{PRC}_{\mathsf{Idx}}[\mathsf{PRC}_{\mathsf{Sub}}, \rho]$ by $m(\lambda)$ (which is defined to be $\lceil \ln(2) \cdot n(\lambda) \rceil$) to distinguish it from the block length $n(\lambda)$ of $\mathsf{PRC}_{\mathsf{Sub}}$.

**Construction of** $\mathsf{PRC}_{\mathsf{Idx}}[\mathsf{PRC}_{\mathsf{Sub}}, \rho]$**.** The idea behind $\mathsf{PRC}_{\mathsf{Idx}}[\mathsf{PRC}_{\mathsf{Sub}}, \rho]$ is simple: we interpret each symbol of $\mathsf{PRC}_{\mathsf{Idx}}[\mathsf{PRC}_{\mathsf{Sub}}, \rho]$ as an *index* into a codeword of $\mathsf{PRC}_{\mathsf{Sub}}$, so that the existence of a symbol in a given codeword of $\mathsf{PRC}_{\mathsf{Idx}}[\mathsf{PRC}_{\mathsf{Sub}}, \rho]$ should be interpreted as the corresponding codeword for $\mathsf{PRC}_{\mathsf{Sub}}$ as having a "1" in the position corresponding to that symbol. To ensure stronger robustness guarantees, it turns out to be necessary to introduce redundancy in the sense that for each integer $j \in [n(\lambda)]$ (representing an index of a codeword of $\mathsf{PRC}_{\mathsf{Sub}}$), there are $\rho$ different elements of $[q(\lambda)]$ which correspond to index $j$. The choice of these $\rho$ elements for each $j$ is specified a mapping $\psi : [q(\lambda)] \to [n(\lambda)]$ with $|\psi^{-1}(j)| = \rho$ for all $j$, which is chosen randomly in the $\mathsf{KeyGen}$ function of $\mathsf{PRC}_{\mathsf{Idx}}[\mathsf{PRC}_{\mathsf{Sub}}, \rho]$. With this intuition in mind, we proceed to overview the individual $\mathsf{KeyGen}, \mathsf{Encode}, \mathsf{Decode}$ functions of $\mathsf{PRC}_{\mathsf{Idx}}[\mathsf{PRC}_{\mathsf{Sub}}, \rho]$ in Algorithm 1:

- The $\mathsf{KeyGen}(1^\lambda)$ function generates a secret key $\mathsf{sk}$ for $\mathsf{PRC}_{\mathsf{Sub}}$ using $\mathsf{KeyGen}_{\mathsf{Sub}}$. It also generates a random function $\psi$ as described above, and returns the tuple $(\mathsf{sk}, \psi)$, which is the secret key for $\mathsf{PRC}_{\mathsf{Idx}}[\mathsf{PRC}_{\mathsf{Sub}}, \rho]$.

- The $\mathsf{Encode}(1^\lambda, (\mathsf{sk}, \psi), \mathsf{m})$ function calls the encoding method $\mathsf{Encode}_{\mathsf{Sub}}$ for $\mathsf{PRC}_{\mathsf{Sub}}$, which yields a string $y^0 \in \{0, 1\}^{n(\lambda)}$. It then chooses a string $y \in [n(\lambda)]^{m(\lambda)}$ which has the property that the set of distinct elements of $y$, which we denote by $\mathsf{Unique}(y)$, has small set difference with the set $\mathcal{S}^0 := \{i \in [n(\lambda)] : y_i^0 = 1\}$ of indices at which $y^0$ has a "1". The precise way in which the sets $\mathsf{Unique}(y)$ and $\mathcal{S}^0$ differ is determined by the function $\mathsf{PerturbDifference}$ in Algorithm 1, and is needed to ensure that the output of $\mathsf{Encode}(1^\lambda, \mathsf{sk}, \mathsf{m})$ is indistinguishable from the uniform distribution over $[n(\lambda)]^{m(\lambda)}$ (i.e., that the PRC is undetectable). Finally, $\mathsf{Encode}$ returns a string $z \in [q(\lambda)]^{m(\lambda)}$ where each coordinate $z_j$ is a uniformly random pre-image of $y_j$ under $\psi$.

- The $\mathsf{Decode}(1^\lambda, (\mathsf{sk}, \psi), z)$ function calls the substitution PRC decode function, $\mathsf{Decode}_{\mathsf{Sub}}$, on the string $y' \in \{0, 1\}^{n(\lambda)}$ which has a 1 in position $i \in [n(\lambda)]$ if and only if $i \in \mathsf{Unique}(\psi(z))$. For future reference, we denote this string by $y' = D_\psi(z)$, i.e., $D_\psi(z)_i = \mathbb{1}\{i \in \mathsf{Unique}(\psi(z))\}$.

Theorem 4.1 below shows that $\mathsf{PRC}_{\mathsf{Idx}}[\mathsf{PRC}_{\mathsf{Sub}}, \rho]$ has robustness to any channel which makes a large fraction (at most $1 - C_{\mathsf{rob}} p_0$) of substitutions, insertions, and deletions.

**Theorem 4.1.** *There are constants $C_0, C_{\mathsf{rob}} \geq 1$ so that the following holds. For any $p_0 < (10 C_{\mathsf{rob}})^{-1}$ and PRC $\mathsf{PRC}_{\mathsf{Sub}}$ with block length $n(\lambda)$ which is robust to all $(1/2 - p_0)$-substitution-bounded channels, for any $\rho \geq C_0 / p_0$, $\mathsf{PRC}_{\mathsf{Idx}}[\mathsf{PRC}_{\mathsf{Sub}}, \rho]$ (Algorithm 1) is a pseudorandom code over an alphabet of size $\lceil \rho \cdot n(\lambda) \rceil$ and block length at most $n(\lambda)$, which has robustness to any $(1 - C_{\mathsf{rob}} p_0)$-edit-bounded channel (per Definition 2.4).*

**Proof overview for Theorem 4.1.** To prove Theorem 4.1, we need to establish the soundness, undetectability, and robustness of $\mathsf{PRC}_{\mathsf{Idx}}[\mathsf{PRC}_{\mathsf{Sub}}, \rho]$. Soundness is an immediate consequence of soundness of $\mathsf{PRC}_{\mathsf{Sub}}$ (Lemma D.2). Undetectability is likewise straightforward, using undetectability of $\mathsf{PRC}_{\mathsf{Sub}}$ together with the fact that the output of $\mathsf{PerturbDifference}(n, m, y^0)$, for $y^0 \sim \mathrm{Unif}(\{0, 1\})^n$, is uniform on $[n]^m$ (Lemma D.2). The bulk of the proof consists in establishing robustness.

A natural attempt to establish robustness would proceed as follows: given a string $z \in [n(\lambda)]^{m(\lambda)}$ (to be interpreted as an output of $\mathsf{Encode}(1^\lambda, (\mathsf{sk}, \psi), \mathsf{m})$), a single insertion or deletion in $z$ can change at most one symbol of $D_\psi(z)$, and a single substitution in $z$ can change at most two symbols of $D_\psi(z)$. Thus, it is straightforward to show a statement of the following form: if $\mathsf{PRC}_{\mathsf{Sub}}$ is $(1/2 - p_0)$-substitution bounded and $2p < 1/2 - p_0$, then $\mathsf{PRC}_{\mathsf{Idx}}[\mathsf{PRC}_{\mathsf{Sub}}, \rho]$ is robust to the class of $p$-edit-bounded channels. (Technically, some additional work is needed since the $\mathsf{PerturbDifference}$ function introduces some additional substitutions in the underlying binary codeword, though we ignore this detail for now since with an appropriate choice of parameters the number of such errors will be of lower order.)

---

**Algorithm 1** Indexing PRC: $\mathsf{PRC_{Idx}}[\mathsf{PRC_{Sub}}, \rho]$

---

**Require:** PRC for substitutions: $(\mathsf{KeyGen_{Sub}}, \mathsf{Encode_{Sub}}, \mathsf{Decode_{Sub}})$, together with functions $n : \mathbb{N} \to \mathbb{N}, \ell : \mathbb{N} \to \mathbb{N}$ characterizing the block length and key length, respectively; parameter $\rho > 1$. Functions $m(\lambda) := \lceil \ln(2) \cdot n(\lambda) \rceil, q(\lambda) := \rho \cdot n(\lambda)$.

 1: **function** $\mathsf{KeyGen}(1^\lambda)$
 2:      Define $\mathsf{sk} \leftarrow \mathsf{KeyGen_{Sub}}(1^\lambda)$.
 3:      Let $\psi : [q(\lambda)] \to [n(\lambda)]$ be a uniformly random function conditioned on $|\psi^{-1}(j)| = \rho$ for each $j \in [n(\lambda)]$.
 4:      **return** $(\mathsf{sk}, \psi)$.

 5: **function** $\mathsf{Encode}(1^\lambda, (\mathsf{sk}, \psi), \mathsf{m})$
 6:      Set $y^0 \leftarrow \mathsf{Encode_{Sub}}(1^\lambda, \mathsf{sk}, \mathsf{m}) \in \{0,1\}^{n(\lambda)}$.
 7:      Set $y \leftarrow \mathsf{PerturbDifference}(n(\lambda), m(\lambda), y^0)$.
 8:      For each $j \in [m(\lambda)]$, choose $z_j \sim \mathrm{Unif}(\{a \ : \ \psi(a) = y_j\})$.
 9:      **return** $z = (z_1, \ldots, z_{m(\lambda)})$.

10: **function** $\mathsf{Decode}(1^\lambda, (\mathsf{sk}, \psi), z)$
11:      Define $y := \psi(z) = (\psi(z_1), \ldots, \psi(z_{m(\lambda)}))$.
12:      Define $y' \in \{0,1\}^n$ by $y'_i = \mathbb{1}\{i \in \mathsf{Unique}(y)\}$.
13:      **return** $\mathsf{Decode_{Sub}}(1^\lambda, \mathsf{sk}, y')$.

14: **function** $\mathsf{PerturbDifference}(n, m, y^0)$
15:      Set $\mathcal{S}^0 := \{i \in [n] \ : \ y_i^0 = 1\}$.
16:      Sample $y^1 \sim \mathrm{Unif}([n]^m)$.
17:      Set $\mathcal{S}^1 := \mathsf{Unique}(y^1) \subset [n]$.
18:      **if** $|\mathcal{S}^0 \backslash \mathcal{S}^1| \geq |\mathcal{S}^1 \backslash \mathcal{S}^0|$ **then**
19:          Let $\sigma : \mathcal{S}^1 \backslash \mathcal{S}^0 \to \mathcal{S}^0 \backslash \mathcal{S}^1$ denote a uniformly random injective mapping.
20:          For each $a \in \mathcal{S}^1 \backslash \mathcal{S}^0$, let $y$ be formed by replacing each instance of $a$ in $y^1$ with $\sigma(a)$.
21:      **else**
22:          Let $\tau : \mathcal{S}^0 \backslash \mathcal{S}^1 \to \mathcal{S}^1 \backslash \mathcal{S}^0$ denote a uniformly random injective mapping.
23:          For each $a \in \mathcal{S}^0 \backslash \mathcal{S}^1$, let $y$ be formed by replacing each instance of $\tau(a)$ in $y^1$ with $a$.
24:      **return** $y$.

---

Unfortunately, such a result does not have sufficiently good robustness for our application to watermarking. As will be discussed in Section 5, our procedure which watermarks a language model Model by "embedding" a codeword $z$ of a PRC in a sequence of text output by Model introduces a fraction $1 - \alpha$ of errors to $z$, all of which are substitutions. Here $\alpha$ is some constant which is related to the entropy rate of Model. Using the naive approach above, we are constrained to a rate of substitutions $p$ bounded as $p < 1/4$, thus forcing $1 - \alpha < 1/4$ and so disallowing all but high entropy rates.

To compensate, we make use of the fact that the randomly chosen mapping $\psi : [q(\lambda)] \to [n(\lambda)]$ maps multiple (namely, $\rho$) symbols in $[q(\lambda)]$ to each symbol in $[n(\lambda)]$, when performing decoding. In particular, consider any fixed channel $\mathcal{E}$ over the alphabet $[q(\lambda)]$. For simplicity in our overview here, we assume that $\mathcal{E}$ is deterministic (so that it is specified by a mapping $\mathcal{E} : [q(\lambda)]^m \to [q(\lambda)]^\star$ for each $m \in \mathbb{N}$), though essentially the same argument works for randomized $\mathcal{E}$. Consider a codeword $z \in [q(\lambda)]^{m(\lambda)}$ which is output by $\mathsf{Encode}(1^\lambda, (\mathsf{sk}, \psi), \mathsf{m})$. By undetectability of the code, with high probability over Encode, we will have that $|\mathsf{Unique}(\psi(z))| \approx n(\lambda)/2 \pm O(\sqrt{n(\lambda)})$, since by our choice of $m(\lambda) = \lceil n(\lambda) \cdot \ln(2) \rceil$, a uniformly random string $\tilde{z} \sim \mathrm{Unif}([q(\lambda)]^{m(\lambda)})$ satisfies $|\mathsf{Unique}(\psi(\tilde{z}))| \approx n(\lambda)/2 \pm O(\sqrt{n(\lambda)})$ with high probability.[6]

---

[6]Here we let $\psi(z) := (\psi(z_1), \ldots, \psi(z_{m(\lambda)}))$.

Supposing that $\mathcal{E}$ is promised to make at most a fraction $1-p$ of edits, then $z' := \mathcal{E}(z)$ shares at least $p \cdot m(\lambda)$ symbols with $z$. Let us suppose for simplicity here that $z' \in [q(\lambda)]^{m(\lambda)}$, so that the insertions and deletions of $\mathcal{E}$ are balanced (a slight modification of the argument handles the general case). Of the remaining $(1-p) \cdot m(\lambda)$ symbols of $z'$, by the random choice of $\psi$ and since $|\mathsf{Unique}(\psi(z))| \approx n(\lambda)/2$, each one is roughly equally likely to map (under $\psi$) to an element in $\mathsf{Unique}(\psi(z))$ as to an element in $[n(\lambda)]\backslash\mathsf{Unique}(\psi(z))$. Thus, in going from the set $\mathsf{Unique}(\psi(z))$ to the set $\mathsf{Unique}(\psi(z'))$, we should expect to change at most roughly $(1-p) \cdot n(\lambda)/2$ elements. This intuition is made precise in the following lemma:

**Lemma 4.2** (Informal version of Lemma D.7). *Given a channel $\mathcal{E}$ as above which makes a $(1-p)$-fraction of edits (i.e., substitutions, insertions, and deletions), with $1 - \mathsf{negl}(\lambda)$ probability over the draw of of $(\mathsf{sk}, \psi) \leftarrow \mathsf{KeyGen}(1^\lambda)$ and $z \leftarrow \mathsf{Encode}(1^\lambda, (\mathsf{sk}, \psi), \mathsf{m})$ we can bound the set difference between $\mathsf{Unique}(\psi(z)), \mathsf{Unique}(\psi(\mathcal{E}(z)))$ as follows:*

$$|\Delta(\mathsf{Unique}(\psi(z)), \mathsf{Unique}(\psi(\mathcal{E}(z))))| \leq (1 - \Omega(p)) \cdot n(\lambda).$$

Since the Hamming distance $D_{\mathsf{Ham}}(D_\psi(z), D_\psi(z'))$ is equal to the size of the set difference $|\Delta(\mathsf{Unique}(\psi(z)), \mathsf{Unique}(\psi(z')))|$, we arrive at the conclusion that $D_{\mathsf{Ham}}(D_\psi(z), D_\psi(z')) \lesssim (1 - \Omega(p)) \cdot n(\lambda)/2$ with high probability over the draw of $\psi$ in KeyGen. Since $p_0$ can be chosen arbitrarily small, we have substitution PRCs $\mathsf{PRC}_{\mathsf{Sub}}$ which can correct a $(1 - \Omega(p))/2$ fraction of substitutions. Thus, we obtain as a consequence that $\mathsf{PRC}_{\mathsf{Idx}}[\mathsf{PRC}_{\mathsf{Sub}}, \rho]$ can correct a $(1-p)$ fraction of substitutions, insertions, and deletions.

The argument above omits many details, notably involving the high-probability event referenced in Lemma 4.2. To establish that such an event occurs with high probability (over the draw of $\psi$), we need to use Dobrushin's concentration inequality (Theorem F.3) for data with limited dependencies. Roughly speaking, this inequality comes into play because the sets $\psi^{-1}(1), \ldots, \psi^{-1}(n(\lambda)) \subset [q(\lambda)]$ are not fully independent (since, e.g., they must be disjoint). Nevertheless, we may bound their dependencies, assuming that $n(\lambda)$ is sufficiently large as compared to $\rho$.

## 5 From large-alphabet PRCs to watermarking

In this section, we overview our reduction (stated in Theorem 5.1) that converts a PRC with robustness to channels making a bounded number of adversarial edits to a watermarking scheme with robustness to adversarial edits. At a high level, this reduction uses rejection sampling at each step of the language model generation to make the output of the model align with a PRC codeword. To formally state the result, we need the notion of *empirical entropy*: given a token sequence $\mathsf{t} \in \Sigma^L$ and $i, j \in [L]$, the empirical entropy of Model on the subsequence $[i, j]$ is

$$H_{\mathsf{e}}^{[i:j]}(\mathsf{t}, \mathsf{Model}) := - \log \Pr_{\mathsf{t}'_{i:j} \sim \mathsf{Model}(\cdot | \mathsf{t}_{1:i-1})}(\mathsf{t}'_{i:j} = \mathsf{t}_{i:j} \mid \mathsf{t}_{1:i-1}),$$

where the probability is over a sequence that is drawn, token by token, from the per-token distributions induced by Model (see Definition B.1 for discussion). The empirical entropy quantifies the degree to which the sequence $\mathsf{t}_{i:j}$ is "far from being deterministic" under Model given the prefix $\mathsf{t}_{1:i-1}$.

The watermarking schemes we derive from PRCs satisfy a stronger property known as *substring robustness* [CGZ24, CG24], which means that if any sufficiently high-entropy *substring* of watermarked text is passed through a channel inducing a bounded number of edits, then the watermark will still be detected. More precisely, for a function $\beta : \mathbb{N} \to \mathbb{N}$, a watermarking procedure $\mathcal{W} = (\mathsf{Setup}, \mathsf{Wat}, \mathsf{Detect})$ over an alphabet $\Sigma$ is $\beta$-*substring robust* to a channel $\mathcal{E}$ if

$$\Pr_{\substack{\mathsf{sk} \leftarrow \mathsf{Setup}(1^\lambda) \\ \mathsf{t} \leftarrow \mathsf{Wat}(1^\lambda, \mathsf{sk}), \mathsf{t}' \leftarrow \mathcal{E}(\mathsf{t})}} \left( \exists i, \ell \text{ s.t. } \mathsf{Detect}(1^\lambda, \mathsf{sk}, \mathsf{t}') = \mathsf{False} \text{ and } H_{\mathsf{e}}^{[i:i+\ell-1]}(\mathsf{t}) \geq \beta(\ell) \cdot \ln |\Sigma| \right) \leq \mathsf{negl}(\lambda).$$

We remark that the channel $\mathcal{E}$ is required to be *non-adaptive* in that it cannot depend on the particular draw of the secret key $\mathsf{sk}$. Some details are omitted; see Definition B.2 for a completely formal definition of substring-robustness. Our main result of this section shows that a PRC with robustness to edit-bounded channels yields a watermarking scheme with substring-robustness to edit-bounded channels:

**Theorem 5.1** (Informal version of Theorem E.1). *Let $\alpha \in (0,1)$ be a given constant. Suppose* PRC, *defined over alphabet $\Sigma_{\mathsf{PRC}}$, has block length $n$ and is robust to any $(1 - \frac{\alpha}{16})$-edit-bounded channel. Further suppose* Model *is a language model over alphabet $\Sigma$ satisfying $|\Sigma| \geq (\frac{8}{\alpha}|\Sigma_{\mathsf{PRC}}|)^{2/\alpha}$. Then there is a watermarking scheme $\mathcal{W}[\mathsf{PRC}, \mathsf{Model}]$ (Algorithm 3) which is sound, undetectable, and $\beta(\ell)$-substring robust to any $\frac{\alpha^2}{48}$-edit-bounded channel, for $\beta(\ell) := 8n + 6\alpha\ell$.*

Theorem 5.1 establishes that, as long as the entropy from a substring of generated text is roughly a $\Omega(\alpha)$-fraction of the maximum possible entropy, then a constant ($O(\alpha^2)$) fraction of edits to that substring cannot remove the watermark. We remark that this constant fraction of edits cannot be impoved to beyond an $\alpha$-fraction (as is evident from the example in Footnote 4). The proof of Theorem 5.1 builds off of the reduction of [CG24], for which $\Sigma = \Sigma_{\mathsf{PRC}} = \{0, 1\}$. Their reduction, however, breaks down in the setting when $\Sigma_{\mathsf{PRC}}$ is no longer binary, and we introduce some new ideas (roughly, involving a hashing technique) to deal with the setting of larger alphabets. Details of the proof may be found in Appendix E. Finally, we remark that by combining Theorems 3.2, 4.1 and 5.1, we obtain Theorem 1.1, which establishes the existence of edit-robust watermarking schemes under Assumption 3.1.

**On implementation of the watermarking scheme.** A natural question is how feasible it is to implement the watermarking scheme $\mathcal{W}[\mathsf{PRC}, \mathsf{Model}]$ of Theorem 5.1. The main limitation of our present theoretical results which may complicate a practical implementation is as follows: The alphabet size $|\Sigma(\lambda)|$ is required to grow exponentially in the inverse of the parameter $\alpha$ (see the statement of Theorem E.2). In turn, the parameter $\alpha$ is proportional to the entropy rate of the text needed to guarantee substring robustness (see Definition B.2 and the setting of $\beta_\lambda(\ell) \asymp \alpha \cdot \ell$ in Theorem E.2). For typical LLMs, the alphabet size is likely smaller than our required value of $|\Sigma(\lambda)|$ given the entropy rates observed empirically in natural language. On the other hand, we believe that future work aimed at developing modifications of our watermarking scheme with an eye towards practical implementation will be successful. One idea which seems promising is to simulate a larger alphabet by grouping tokens together, and to aim accordingly for a slightly weaker robustness guarantee.

## Acknowledgements

NG was supported by a Fannie & John Hertz Foundation Fellowship and an NSF Graduate Fellowship. AM is supported in part by a Microsoft Trustworthy AI Grant, an ONR grant and a David and Lucile Packard Fellowship. We thank Gabe Schoenbach for helpful comments on an earlier draft of this work.

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

# A  Additional related work

Watermarking has a long history, dating back to the work of [TTI05, ARC⁺01, ARH⁺02]: generally speaking, these early works considered procedures which embed a watermark into a *given* text, by *altering* the text in some subtle way. Spurred by the remarkable abilities of generative models, in particular large language models (LLMs), there has been a recent resurgence of interest in watermarking. In contrast to the classical works, most recent works (including this paper) aim to embed the watermark in a way that does not significantly alter the *distribution* of the generated content. This task is enabled by the autoregressive nature of LLMs, which generate each successive token using some fresh randomness.

Recent works [ZALW24, AK22, KGW⁺23] construct watermarking schemes by pereferentially generating certain tokens at each step of generation. For instance, [ZALW24] partitions tokens into two equal-sized sets, a "green set" and a "red set". It slightly increases the probabilities of green tokens while slightly decreasing the probabilities of red tokens. A watermark is then detected if the proportion of green tokens in a text segment is noticeably higher than $\frac{1}{2}$. The works of [AK22, KGW⁺23] build on this protocol by generating the green and red sets at each position using a pseudorandom function seeded with a window of some number $k$ of previous tokens. These schemes all introduce some degree of noticeable bias to the watermarked model's output, by increasing the probability of certain $k$-grams.

**Distortion-freeness vs undetectability.**   [KTHL23] introduced a notion called *distortion-freeness*, which posits that a *single text sample* from the watermarked model is distributed identically to a single sample from the true model. [KTHL23] construct watermarking schemes satisfying distortion-freeness, with nontrivial edit-robustness guarantees: their watermarks have robustness to a constant fraction of substitutions (Lemma 2.5 within), but only to a slightly sub-constant fraction of edits when insertions and deletions are also allowed (Lemma 2.6 within).[7] [HCW⁺24, WHZH23] also developed a similar approach to [KTHL23].

[CGZ24] defined the stronger notion of *undetectability* (the focus of the present work), which requires that no computationally efficient algorithm which makes *any number* of queries to the language model can distinguish between the watermarked model and the original one. [CGZ24] constructed the first provably undetectable watermark by using a similar technique to [AK22, KGW⁺23] with the modification that they dynamically adjust the parameter $k$ denoting the length of text used to seed the pseudorandom function. This dynamic adjustment ensures that the sequence of $k$ tokens has sufficiently high entropy, which ensures undectability.[8] Undetectable watermarking schemes were also constructed by [Zam24], which focused specifically on the application to *steganography* (see also [CG24] for robust steganography schemes), and [FGJ⁺23], which concentrated on schemes with *public attribution*.

It is natural to wonder: *why should one should prefer undetectable watermarking schemes over distortion-free ones?* The simplest reason is that, as a requirement on generative models, distortion-freeness is very weak, as illustrated in the below example.

**Example A.1** (Distortion-freeness).  Consider the alphabet $\Sigma = \{0, 1\}$ and suppose that $\overline{\mathsf{Model}}$ is the *uniformly random* distribution on outputs $\mathsf{t} \in \{0, 1\}^n$, for some $n \in \mathbb{N}$. Consider the watermarking scheme which draws a key $\mathsf{sk} \sim \mathrm{Unif}(\{0, 1\}^n)$ and at each call to $\mathsf{Wat}(\mathsf{sk})$, simply returns $\mathsf{sk}$. This scheme is distortion-free since the distribution of $\mathsf{sk}$ is uniform, but has the property that once $\mathsf{Wat}$ is called once, the output of all future calls is determined (and so it is clearly not undetectable).

Example A.1 indicates that distortion-free watermarks may suffer from insufficient diversity in model outputs: in Example A.1 there is no variation whatsoever in the outputs. The distortion-free schemes of [KTHL23] are only slightly better on this front: they sample a long random key $\mathsf{sk}$ and then

---

[7]In particular, in order for [KTHL23, Lemma 2.6] to be nonvacuous, the parameter $\gamma$ must be chosen so that $\gamma > \log k$, which necessitates $\epsilon < 1/\gamma < 1/\log k$, and $k$ must grow faster than a constant for the probability in [KTHL23, Lemma 2.6] to decay to 0.

[8]Notice that undetectability is not strictly speaking stronger than distortion-freeness, since an undetectable watermarking scheme can perturb the distribution of even a single text sample output by the model, though only in a way that can not be detected by any computationally efficient algorithm. Since in practice, any downstream applications of language model outputs will proceed via computationally efficient algorithms, we view undetectability as stronger than distortion-freeness "for all intents and purposes".

generate a sequence of text from Model in a way that aims to be close to a *random shift* of sk. Having fixed the key sk, the output text from the scheme of [KTHL23] is a deterministic function of the shift, meaning that at most $\mathrm{len}(\mathsf{sk})$ distinct sequences of text can be generated altogether. Moreover, by the birthday paradox, one would expect to see a repeat answer after only $\sqrt{\mathrm{len}(\mathsf{sk})}$ calls. The effective number of distinct samples of the distribution of output text of [KTHL23] may be even smaller than $\mathrm{len}(\mathsf{sk})$, if different shifts of sk lead to similar text sequences (as they will for, e.g., the uniform Model in Example A.1).

Beyond avoiding the issue of insufficient diversity, the stronger guarantees afforded by undetectability offer a more compelling argument for adopting watermarking in the first place. Watermarking is just one of many technologies that can be built into AI models to achieve better alignment with human values. Even if each of these degrades performance in minor but non-negligible ways, the aggregate impact of all of these additions could potentially potentially be substantial. The guarantee of undetectability, namely that *no matter what efficient downstream algorithms are applied to the model's output, the result will be essentially the same*, means that the addition of a watermark can only be blamed for a negligible amount of the overall performance degradation.

**Additional work on watermarking.**   [HZZ+24] formulate a variant of the watermarking problem which is purely statistical in nature, and characterize the optimal rate of watermarking. [GLLH24] show that watermarking schemes can be *learned*, using a student-teacher framework. [KGW+23, CG24, PHZS24, ZEF+23] develop various attacks on watermarking schemes. Generally speaking, these attacks are relatively expensive in terms of calls to the language model (as in [CG24, PHZS24]) or an additional "quality oracle" (as in [ZEF+23]). Moreover, a sufficiently motivated adversary could simply train their own (unwatermarked) AI model and generate content from it. Therefore, we view robust watermarking as targeted more at "honest" or "lazy" adversaries who are making edits to either improve the quality of content without explicitly trying to remove a watermark (e.g., for AI-generated news articles) or have very few resources to remove the watermark (e.g., a student asking an LLM to write their essay right before the deadline).

Finally, we remark that there is a sizeable body of work focusing on empirical approaches to watermarking (e.g., [LPH+24b, LPH+24a, LB24, CBD+24, KGW+24]).

**Error-correcting codes for insertions and deletions.**   A recent line of work (e.g., [GL16, BGH17, GW17, HS21a, HS21b]) has focused on developing error-correcting codes for insertions and deletions. We remark that the idea of indexing is implicit in some of this work. One focus of these papers has been on obtaining codes which can correct a constant fraction of insertions and deletions with smaller (e.g., constant-size) alphabets. A common technique for this task is that of of *synchronization strings* [HS21a, HS21b]: it proceeds by increasing the alphabet size by a constant factor and attaching to each message character an "auxiliary character" coming from a so-called synchronization string, which has certain properties that make it easier to align a noisy codeword with the clean codeword. Unfortunately, the same synchronization string must be used with each call to the encoding algorithm, which makes it unclear how to construct such codes that are pseudorandom.

# B   Additional preliminaries

## B.1   Watermarking language models

A language model Model over alphabet $\Sigma$ which, for any positive integer $i$, takes as input a sequence of tokens $\mathsf{t}_{1:i-1} = (\mathsf{t}_1, \ldots, \mathsf{t}_{i-1})$ already output by the model and produces a distribution $\mathsf{Model}(\mathsf{t}_i = \cdot \mid \mathsf{t}_{1:i-1}) \in \Delta(\Sigma)$, representing the distribution of the next token, conditioned on $\mathsf{t}_{1:i-1}$. We assume that Model is computationally efficient, meaning that it runs in time $\mathrm{poly}(i, \log |\Sigma|)$. We also assume that there is some token $\mathsf{END} \in \Sigma$ representing the end of the model's response. As a matter of convention, we assume that all subsequent tokens after a $\mathsf{END}$ token are also $\mathsf{END}$ (i.e., $\mathsf{Model}(\mathsf{END} \mid \mathsf{t}_{1:i-1}) = 1$ if some $\mathsf{t}_j$ for $j < i$ is $\mathsf{END}$). Given a language model Model, we introduce the notation $\overline{\mathsf{Model}}$: given any sequence $\mathsf{t}_{1:i-1} \in \Sigma^{i-1}$ (which may be the empty sequence) and $m > 0$, the distribution $\overline{\mathsf{Model}}(\mathsf{t}_{i:i+m-1} = \cdot \mid \mathsf{t}_{1:i-1}) \in \Sigma^m$ is defined as follows: we sequentially generate $\mathsf{t}_i, \ldots, \mathsf{t}_{i+m-1}$, where for each $0 \le j < m$, $\mathsf{t}_j$ is distributed as $\mathsf{Model}(\mathsf{t}_j = \cdot \mid \mathsf{t}_{1:j-1})$. In the case $i = 1$ (i.e., $\mathsf{t}_{1:i-1} = \emptyset$), then we denote an output of Model as $\mathsf{t} \sim \overline{\mathsf{Model}}$.

An intuitive requirement on Model that allows for watermarking is that outputs of Model must be *high-entropy*; indeed, a model that outputs a deterministic sequence of text is impossible to watermark, since the only distribution which is indistinguishable from its output is that same deterministic sequence of text. As we aim to measure the entropy of individual model outputs, we use the following notion of *empirical entropy*, following [CGZ24]:

**Definition B.1** (Empirical entropy). Given a language model Model, a sequence $t \in \Sigma^L$, and $i, j \in t$, we define the *empirical entropy of* Model *for* $t$ *on subsequence* $[i, j]$ to be:

$$H_e^{[i:j]}(t, \text{Model}) := -\log \overline{\text{Model}}(t_{i:j} \mid t_{1:i-1}).$$

By definition of $\overline{\text{Model}}$, we have $-\log \overline{\text{Model}}(t_{i:j} \mid t_{1:i-1}) = -\log \Pr_{\bar{t} \sim \overline{\text{Model}}}(\bar{t}_{i:j} = t_{i:j} \mid \bar{t}_{1:i-1} = t_{1:i-1})$. When Model is clear from context, we typically drop Model from the notation, so that $H_e^{[i:j]}(t) := H_e^{[i:j]}(t, \text{Model})$, and moreover write $H_e^i(t) := H_e^i(t, \text{Model})$, $H_e^{[i:j)}(t) := H_e^{[i:j-1]}(t)$. Finally, note that from the chain rule of probability, $H_e^{[i:j]}(t) = \sum_{a=i}^{j} H_e^a(t)$.

Our main goal in this paper is to construct a *watermarking scheme* for Model, which is a tuple $\mathcal{W} = (\text{Setup}, \text{Wat}, \text{Detect})$ of probabilistic polynomial-time algorithms with the following semantics:

- $\text{Setup}(1^\lambda)$ takes as input a security parameter $\lambda \in \mathbb{N}$ and outputs a secret key $\text{sk}$.
- $\text{Wat}(1^\lambda, \text{sk})$ takes as input $\lambda, \text{sk}$ and outputs a sequence $t = t_{1:L} \in \Sigma^L$, for some $L \in \mathbb{N}$.
- $\text{Detect}(1^\lambda, \text{sk}, t)$ takes as input a sequence $t \in \Sigma^\ell$ for some $\ell \in \mathbb{N}$ and outputs either True or False, denoting whether $t$ is detected as being watermarked.

We next define the security properties we desire in our watermarks, in terms of a security parameter $\lambda \in \mathbb{N}$. Some of our results operate in the "large-alphabet" setting, meaning that the size of the alphabet for the language model can depend (polynomially) on $\lambda$. Formally, we consider a language model family indexed by $\lambda$, $(\text{Model}(\lambda))_{\lambda \in \mathbb{N}}$, where the alphabet for $\text{Model}(\lambda)$ is denoted by $\Sigma(\lambda)$. Our watermarking procedure should produce a string in $\Sigma(\lambda)^\star$ which is computationally indistinguishable from an output of $\text{Model}(\lambda)$. In order for us to establish a computational separation between the process of outputting watermarked text and an adversary running in time given by an arbitrarily large polynomial in $\lambda$, we assume that the length of an output $t \sim \overline{\text{Model}}(\lambda)$ is bounded by $L_{\max}(\lambda)$, where $L_{\max}(\lambda)$ is some function growing as $\text{poly}(\lambda)$.[9] When the value of $\lambda$ is clear from context, we drop the argument $\lambda$ and simply write $\text{Model}, \Sigma$.

**Definition B.2** (Soundness, Undetectability, $\mathscr{E}$-Robustness). Consider a language model family $\text{Model}(\lambda)$ together with a watermarking scheme $\mathcal{W} = (\text{Setup}, \text{Wat}, \text{Detect})$. We define the following properties of $\mathcal{W}$:

- $\mathcal{W}$ is *sound* if for all $\lambda \in \mathbb{N}$ and $t \in \Sigma^{\leq L_{\max}(\lambda)}$,
$$\Pr_{\text{sk} \leftarrow \text{Setup}(1^\lambda)} \left( \text{Detect}(1^\lambda, \text{sk}, t) = \text{True} \right) \leq \text{negl}(\lambda).$$

- $\mathcal{W}$ is *undetectable* if for all $\lambda \in \mathbb{N}$ and any probabilistic polynomial time algorithm Dist,
$$\left| \Pr \left( \text{Dist}^{\overline{\text{Model}}}(1^\lambda) = 1 \right) - \Pr \left( \text{Dist}^{\text{Wat}(1^\lambda, \text{sk})}(1^\lambda) = 1 \right) \right| \leq \text{negl}(\lambda),$$
where $\text{Dist}^{\mathcal{O}}$ means that Dist can make repeated calls to $\mathcal{O}$, which generate a sample from the corresponding distribution (either $t \sim \overline{\text{Model}}$ or $t \sim \text{Wat}(1^\lambda, \text{sk})$).

- Fix some family of channels $(\mathscr{E}(\lambda))_{\lambda \in \mathbb{N}}$, where each $\mathscr{E}(\lambda)$ is a collection of channels $\mathcal{E} : \Sigma(\lambda)^\star \to \Delta(\Sigma(\lambda)^\star)$, and a family of functions $\beta_\lambda : \mathbb{N} \to \mathbb{N}$. Then $\mathcal{W}$ is defined to be $\beta$-*substring robust* to the family $(\mathscr{E}(\lambda))_\lambda$ if for each $\lambda \in \mathbb{N}$, and each channel $\mathcal{E} \in \mathscr{E}(\lambda)$,
$$\Pr_{\substack{\text{sk} \leftarrow \text{Setup}(1^\lambda) \\ t \leftarrow \text{Wat}(1^\lambda, \text{sk}), t' \leftarrow \mathcal{E}(t)}} \left( \exists i, j \in [L_{\max}(\lambda)] \text{ s.t. } \text{Detect}(1^\lambda, \text{sk}, t') = \text{False and } H_e^{[i:j)}(t) \geq \beta_\lambda(j - i) \cdot \ln |\Sigma(\lambda)| \right) \leq \text{negl}(\lambda).$$

  Above, we use the convention that $t$ is to be interpreted as an element of $\Sigma^{L_{\max}(\lambda)}$ by padding it with instances of the terminal token END. Thus, substring robustness requires that if we choose any substring of length $\ell$ from the output of Wat and pass it through an channel $\mathcal{E}$, then the output of the channel is still detected as watermarked with high probability.

---

[9]This requirement appears to be implicit in [CG24], though it is not explicitly stated therein.

## B.2 Notation

Given $n \in \mathbb{N}$ together with a mapping $\pi : [n] \to [n]$ and a string $x = (x_1, \ldots, x_n) \in \Sigma^n$, we write $\pi \circ x := (x_{\pi(1)}, \ldots, x_{\pi(n)}) \in \Sigma^n$. Given a mapping $\phi : \mathcal{X} \to \mathcal{Y}$ for (finite) sets $\mathcal{X}, \mathcal{Y}$ and a distribution $P \in \Delta(\mathcal{X})$, we let $\phi \circ P \in \Delta(\mathcal{Y})$ be the push-forward distribution for $\phi$, i.e., $(\phi \circ P)(y) = P(\phi^{-1}(y))$.

Given a mapping $\psi : \mathcal{X} \to \mathcal{Y}$ for finite sets $\mathcal{X}, \mathcal{Y}$, we let $\psi(\mathcal{X}) := \{\psi(x) : x \in \mathcal{X}\} \subset \mathcal{Y}$, and for any $\mathcal{Y}' \subset \mathcal{Y}$, $\psi^{-1}(\mathcal{Y}') := \{x \in \mathcal{X} : \psi(x) \in \mathcal{Y}'\}$. Finally, for a tuple $x = (x_1, \ldots, x_n) \in \mathcal{X}^n$, we let $\psi(x) := (\psi(x_1), \ldots, \psi(x_n)) \in \mathcal{Y}^n$.

For $y \in \Sigma^N$ and $J = (j_1, \ldots, j_t)$, we let $y_J \in \Sigma^t$ denote the vector $y_J := (y_{j_1}, \ldots, y_{j_t})$. Given a set $\mathcal{S}$, $m \in \mathbb{N}$, and a string $y \in \mathcal{S}^m$, we define $\mathsf{Unique}(y) := \{y_i : i \in [m]\} \subset \mathcal{S}$, i.e., $\mathsf{Unique}(y)$ is the *set* of distinct elements of $y$.

# C  Secret-key substitution PRCs

## C.1  Hardness assumptions

In this section, we discuss several standard conjectures on the hardness of PAC learning which imply Assumption 3.1. We first recall the definition of PAC learning with respect to the uniform distribution:

**Definition C.1** (PAC learning for the uniform distribution)**.** Suppose that for each $n \in \mathbb{N}$, we are given a class $\mathcal{F}_n$ of boolean functions indexed by strings $s \in \{0,1\}^{\ell(n)}$, for some function $\ell : \mathbb{N} \to \mathbb{N}$; write $\mathcal{F}_n = \{F_s : \{0,1\}^n \to \{0,1\}\}_{s \in \{0,1\}^{\ell(n)}}$. Given $q \in [0, 1/2)$, we say that $\mathcal{F} = (\mathcal{F}_n)_n$ is $(\epsilon(n), \delta(n))$-*PAC learnable with noise level* $q$ if there is a $\mathrm{poly}(n)$-time algorithm $\mathsf{Alg}$ which, for some $m(n) \in \mathbb{N}$, takes as input a dataset $D = ((x_i, y_i))_{i \in [m(n)]}$ and $X \in \{0,1\}^n$, and satisfies the following with probability $1 - \delta$ over $s \sim \mathrm{Unif}(\{0,1\}^{\ell(n)}), x_i \sim \mathrm{Unif}(\{0,1\}^n), y_i = F_s(x_i)$ $(i \in [m(n)])$:

$$\Pr_{X \sim \mathrm{Unif}(\{0,1\}^n)} \left( \mathsf{Alg}((x_i, y_i)_{i \in [m(n)]}, X) \neq F_s(X) \right) \leq \epsilon.$$

**Proposition C.1.** *Suppose that a family of functions* $\mathcal{F} = (\mathcal{F}_n)_{n \in \mathbb{N}}$ *is not* $(1/3, 1/3)$*-PAC learnable with noise level* $q \in [0, 1/2)$ *(per Definition C.1). Then, with* $n(\lambda) = \lambda$*, the family of functions* $(\mathcal{F}_{n(\lambda)})_\lambda$ *is a weak PRF with noise level* $q$ *(per Definition 3.1).*

Proposition C.1 is essentially folklore (see, e.g., [BR17]), but we provide a proof sketch for completeness:

*Proof sketch of Proposition C.1.* Suppose that a polynomial-time adversary $\mathsf{Adv}$ satisfies

$$\left| \mathbb{E}_{s \sim \{0,1\}^{n(\lambda)}} \left[ \widetilde{\mathsf{Adv}}^{F_s(\cdot)}(1^{n(\lambda)}) = 1 \right] - \mathbb{E}_{F_{\mathsf{Unif}}} \left[ \widetilde{\mathsf{Adv}}^{F_{\mathsf{Unif}}}(1^{n(\lambda)}) \right] \right| \geq \lambda^{-O(1)},$$

where $F_{\mathsf{Unif}} : \{0,1\}^n \to \{0,1\}$ denotes a uniformly random function. Suppose that $\mathsf{Adv}$ makes at most $m(\lambda)$ queries to its oracle (either $F_s(\cdot)$ or $F_{\mathsf{Unif}}$). Consider the algorithm which receives $m(n)$ samples $(x_i, y_i)$ with $x_i \sim \mathrm{Unif}(\{0,1\}^n), y_i = F_s(x_i)$ for an unknown $s$ together with $X \sim \mathrm{Unif}(\{0,1\}^{n(\lambda)})$, chooses a uniformly random index $m' \in [m(\lambda) - 1]$, and then runs $\mathsf{Adv}$ with the oracle responses given by $(((x_i, y_i))_{i \in [m']}, (X, b), (x_j, y_j)_{j \in [m'+2, n(\lambda)]})$, where $x_j \sim \mathrm{Unif}(\{0,1\}^n), y_j \sim \mathrm{Unif}(\{0,1\})$ for $j \geq m' + 2$, for each value of $b \in \{0,1\}$. By a hybrid argument, for some $b \in \{0,1\}$, this algorithm must have $\lambda^{-O(1)}$ advantage at predicting $F_s(X)$. A standard boosting argument [FS95] then establishes the existence of a polynomial-time algorithm that $(1/3, 1/3)$-PAC learns the family $\mathcal{F}$. $\qquad\square$

By Proposition C.1, either of the two assumptions below implies Assumption 3.1:

**Assumption C.2** (Sparse noisy parity)**.** *There is no polynomial-time algorithm which* $(1/3, 1/3)$*-PAC learns the family* $(\mathcal{F}_n)_n$ *of* $\log(n)$*-sparse parities with noise level* $q = 1/3$*, i.e.,* $\mathcal{F}_n = \{F_s(x) = \bigoplus_{i \in [n]} x_i s_i : s \in \{0,1\}^n, \|s\|_1 = \log(n)\}$.

It is straightforward to see (using a hybrid argument) [Blu03] that Assumption C.2 is equivalent to the assumption that there is no polynomial-time algorithm which *identifies* the hidden $\log(n)$-sparse string $s \in \{0,1\}^n$ with constant probability.

**Assumption C.3** ([BFKL94], Section 2.3). *There is no polynomial-time algorithm which $(1/3, 1/3)$-PAC learns the family $(\mathcal{G}_n)_n$ defined below, with noise level $q = 0$:*

$$\mathcal{G}_n = \left\{ G_{\mathcal{S}_1, \mathcal{S}_2}(x) = \mathsf{Maj}(x_i : i \in \mathcal{S}_1) \oplus \bigoplus_{i \in \mathcal{S}_1} x_i \mid \mathcal{S}_1, \mathcal{S}_2 \subset [n] \text{ are disjoint subsets of size } \log(n)/2 \right\}.$$

Similarly to the case for Assumption C.2, Assumption C.3 is equivalent to the assumption that there is no polynomial-time algorithm which *identifies* the function $G_{\mathcal{S}_1, \mathcal{S}_2}$ with constant probability. We remark that [FGKP09, Theorem 3] shows that an algorithm which learns $\log(n)$-sparse parities at noise level $\frac{1}{2} - O(1/n)$ implies the existence of an algorithm which learns (noiseless) $\log(n)$-juntas (with both learning algorithms being over the uniform distribution). Notice, though, that Assumption C.2 assumes there is no efficient algorithm which learns $\log(n)$-sparse noisy parities with noise level $q$ bounded away from $1/2$, which is stronger than assuming hardness of learning $\log(n)$-sparse noisy parities with noise level $q = 1/2 - O(1)$. Thus, even given the reduction of [FGKP09], Assumption C.3 does not appear to imply Assumption C.2.

## C.2  Additional preliminaries

**Weakly substitution-bounded channels.**  We will show that our PRCs are robust to a slightly more broad class of channels which are only guaranteed to introduce a bounded number of substitutions with high probability with respect to a uniformly random string from the channel's alphabet. The codes of [CG24] also enjoy robustness to such channels; thus, the robustness guarantee for our substitution codes is directly comparable to that of [CG24].

**Definition C.2** (Weakly substitution bounded channel). We say that a channel $\mathcal{E}$ over an alphabet $\Sigma$ is *$p$-weakly-substitution-bounded* if for any $n \in \mathbb{N}$, $\Pr_{y \sim \mathrm{Unif}(\Sigma^n), z \sim \mathcal{E}(y)} (D_{\mathsf{Ham}}(y, z) > pn) \leq \mathsf{negl}(n)$.

**Noise sensitivity.**  For $x \in \{0,1\}^n$, we let $N_\rho(x) \in \Delta(\{0,1\}^n)$ denote the distribution over $y$, where $y_i = x_i$ with probability $1/2 + \rho/2$ (and $y_i = 1 - x_i$ with probability $1/2 - \rho/2$), independently for each $i$. Note that $y \sim N_{1-2\delta}(x)$ is generated by flipping each bit of $x$ with probability $\delta$. Define $\mathbf{NS}_\delta[f] := \Pr_{x \sim \mathrm{Unif}(\{0,1\}^n), y \sim N_{1-2\delta}(x)} (f(x) \neq f(y))$. Also write $\mathbf{W}^i[f] = \sum_{S \subseteq [n], |S|=i} \hat{f}(S)^2$.

**Lemma C.4** (Theorem 2.49 of [ODo14]). *For any $f : \{0,1\}^n \to \{-1,1\}$, we have $\mathbf{NS}_\delta[f] = \frac{1}{2} \sum_{i=0}^{n} (1 - (1-2\delta)^i) \cdot \mathbf{W}^i[f]$.*

**Corollary C.5.** *For any $f : \{0,1\}^n \to \{0,1\}$, we have $\mathbf{NS}_\delta[f] = \frac{1}{2} - \sum_{i=0}^{n} \phi_i \cdot (1-2\delta)^i$, for some $\phi_0, \ldots, \phi_n \geq 0$ satisfying $\phi_0 + \cdots + \phi_n = 1/2$.*

*Proof.* The corollary is an immediate consequence of Lemma C.4 applied to the mapping $x \mapsto 2f(x) - 1$, together with the fact that any $f : \{0,1\}^n \to \{-1,1\}$ satisfies $\sum_{i=0}^{n} \mathbf{W}^i[f] = 1$.  $\square$

## C.3  Proof of Theorem 3.2

**Proof overview.**  The proof of soundness Lemma C.6 is straightforward; see Appendix C.3 for details. The proof of undetectability (Lemma C.7) of PRF-PRC$[\mathcal{F}, p, q]$ uses Assumption 3.1: if undetectability failed to hold as witnessed by some adversary Adv, then we could construct an adversary which violates pseudorandomness of $\mathcal{F}$ (per (2)) by simulating Adv, using computational efficiency of Encode together with the PRF oracle to implement the oracle calls for Adv.

The bulk of the proof lies in establishing robustness to $p$-bounded substitution channels (Lemma C.8). Let us first make sure that $\mathsf{Decode}(1^\lambda, \mathsf{sk}, y)$ returns $\emptyset$ with high probability if its input $y$ is simply an output of $\mathsf{Encode}(1^\lambda, \mathsf{sk}, \emptyset)$ (i.e., if the channel $\mathcal{E}$ does nothing to its input). Indeed, the tuples $(x_i, w_i)$ (for $i \in [m(\lambda)]$) computed in Line 11 of Decode are exactly equal to $(x_i, F_s(x_i) \oplus e_i)$. As long as the noise rate $q$ of $e_i$ is bounded away from $1/2$, the statistic $W$ computed in Decode will be larger than $m(\lambda)/2 + \ln(m(\lambda))\sqrt{m(\lambda)}$ with all but negligible probability, as desired.

---

**Algorithm 2** PRF-PRC$[\mathcal{F}, p, q]$: Generic PRF-based PRC

---

**Require:** Weak PRF family $\mathcal{F} = (\mathcal{F}_\lambda)_\lambda$, where $\mathcal{F}_\lambda = \{F_s : \{0,1\}^{n(\lambda)} \to \{0,1\}\}$ indexed by $s \in \{0,1\}^{\ell(\lambda)}$, parameters $p, q \in (0,1)$. Functions $n(\lambda), \ell(\lambda), m(\lambda), N(\lambda) : \mathbb{N} \to \mathbb{N}$ satisfying $N(\lambda) \geq (n(\lambda) + 1)m(\lambda)$,

1: **function** KeyGen($1^\lambda$)
2:     Sample $s \sim \mathrm{Unif}(\{0,1\}^{\ell(\lambda)})$ and $z \sim \mathrm{Unif}(\{0,1\}^{N(\lambda)})$.
3:     Let $\pi : [N(\lambda)] \to [N(\lambda)]$ be a uniformly random permutation.
4:     Return $\mathsf{sk} := (s, z, \pi)$.

5: **function** Encode($1^\lambda, \mathsf{sk} = (s, z, \pi), \emptyset$)
6:     Sample $x_1, \ldots, x_{m(\lambda)} \sim \mathrm{Unif}(\{0,1\}^{n(\lambda)})$ and $e_1, \ldots, e_{m(\lambda)} \sim \mathrm{Ber}(q)$.
7:     Sample $r \sim \mathrm{Unif}(\{0,1\}^{N(\lambda)-(n(\lambda)+1)m(\lambda)})$.
8:     Define $a := ((x_1, F_s(x_1) \oplus e_1), \ldots, (x_{m(\lambda)}, F_s(x_{m(\lambda)}) \oplus e_{m(\lambda)}), r)$.
9:     **return** the string $\pi \circ (a \oplus z)$.

10: **function** Decode($1^\lambda, \mathsf{sk} = (s, z, \pi), y$)
11:     Let $x_1, \ldots, x_{m(\lambda)} \in \{0,1\}^{n(\lambda)}$ and $w_1, \ldots, w_{m(\lambda)} \in \{0,1\}$ be defined by $((x_1, w_1), \ldots, (x_{m(\lambda)}, w_{m(\lambda)})) = ((\pi^{-1} \circ y) \oplus z)_{1:(n(\lambda)+1)m(\lambda)}$.
12:     Let $W := \sum_{j=1}^{m(\lambda)} \mathbb{1}\{w_j = F_s(x_j)\}$.
13:     If $W > \frac{m(\lambda)}{2} + \ln(m(\lambda))\sqrt{m(\lambda)}$, **return** $\emptyset$; otherwise, **return** $\perp$.

---

Now what if $y$ is drawn from $\mathcal{E}(y^0)$, for $y^0 \leftarrow \mathsf{Encode}(1^\lambda, \mathsf{sk}, \emptyset)$, for some $p$-bounded substitution channel $\mathcal{E}$? Then the tuples computed in Line 11 of Decode may be written as $(x_i', w_i')$ (for $i \in [m(\lambda)]$), where $(x_i', w_i')$ are "perturbed" versions of $(x_i, w_i)$, where $x_i \sim \mathrm{Unif}(\{0,1\}^{n(\lambda)}), w_i = F_s(x_i) \oplus e_i$ are as computed in Encode. Although the channel $\mathcal{E}$ may introduce substitutions in an adversarial fashion (i.e., it may not introduce substitutions at each position independently with probability $p$), by virtue of the fact that the output string of Encode is $\pi \circ (a \oplus z)$ for uniformly random $z \in \{0,1\}^{N(\lambda)}, \pi : [N(\lambda)] \to [N(\lambda)]$, we can show that $(x_i', w_i')$ is close to being distributed by perturbing each bit of $(x_i, w_i)$ independently with some probability $p' \leq p$. The proof of this fact uses an approximation of a Hypergeometric distribution with a Binomial distribution: roughly speaking, the permutation $\pi$ allows us to "pick out", for each $i$, a uniformly random subset of $n(\lambda) + 1$ positions corresponding to $(x_i', w_i')$, out of $N(\lambda)$ total positions of the string $y$. Of these, $p' \cdot N(\lambda) \leq p \cdot N(\lambda)$ are changed by $\mathcal{E}$, and as long as $N(\lambda) \gg n(\lambda)$, applying $\mathcal{E}$ thus has nearly the same effect as changing each of the $n(\lambda) + 1$ positions of $(x_i, w_i)$ independently with probability $p'$.

Given the above, we then use the fact that if $x_i \sim \mathrm{Unif}(\{0,1\}^{n(\lambda)})$ and $x_i'$ is formed by flipping each bit of $x_i$ with probability $p' \leq p$, then $\Pr[F_s(x_i) = F_s(x_i')] \geq \frac{1}{2} + (1 - 2p)^\tau \geq \frac{1}{2} + (1 - 2p)^{\log n(\lambda)}$. This is a basic consequence of the Fourier analytic expression for the noise sensitivity of Boolean functions (Lemma C.4), together with the fact that $F_s(\cdot)$ is $\tau$-local for some $\tau \leq \log(n(\lambda))$. Note that $(1 - 2p)^{\log n(\lambda)} \geq n(\lambda)^{-\Omega_p(1)}$; thus, as long as $m(\lambda)$ is a large enough polynomial in $n(\lambda)$, we can show that the statistic $\sum_{i=1}^{m(\lambda)} \mathbb{1}\{w_i' = F_s(x_i')\}$ will be bounded away from $m(\lambda)/2$, which implies that Decode returns $\emptyset$ with high probability.

One complication of the above argument comes from the fact that, due to the adversarial nature of the channel $\mathcal{E}$, the random variables $(x_i, e_i, x_i', w_i')$ may not be independent across different $i$. Thus, to ensure concentration of the sum $\sum_{i=1}^{m(\lambda)} \mathbb{1}\{w_i' = F_s(x_i')\}$ to its mean, we use Dobrushin's concentration inequality for weakly dependent data (Theorem F.3) and bound the dependence of these random variables for different $i$ using the fact that $N(\lambda)$ is sufficiently large.

**Formal proof.** Suppose that we are given some $p < 1/2$ together with a weak PRF family $\mathcal{F} = \{F_s : \{0,1\}^{n(\lambda)} \to \{0,1\}\}_{\lambda \in \mathbb{N}, s \in \{0,1\}^{\ell(\lambda)}}$ with noise level $q < 1/2$ which verifies Assumption 3.1, i.e., for some function $\tau(\lambda) \leq \log n(\lambda)$, the family is $\tau(\lambda)$-local. We consider the construction PRF-PRC$[\mathcal{F}, p, q]$ in Algorithm 2. We let the functions $n(\lambda), \ell(\lambda)$ be as given by the PRF family $\mathcal{F}$,

and choose

$$m(\lambda) := C_0 \cdot (1 - 2q)^{-4} \cdot n(\lambda)^{4 \log \frac{1}{1-2p}}, \qquad N(\lambda) := 3m(\lambda) \cdot (n(\lambda) + 1)^2, \qquad (3)$$

where $C_0$ is a constant chosen sufficient large as specified in the proof below. By Definition 2.1, to prove Theorem 3.2, it suffices to establish soundness, undetectability, and robustness to $p$-weakly-substitution-bounded channels: we do so in Lemma C.6, Lemma C.7, Lemma C.8, respectively, below.

**Lemma C.6** (Soundness). *For any function family $\mathcal{F} = \{F_s\}_s$ on $\{0,1\}^{n(\lambda)}$ indexed by $s \in \{0,1\}^{\ell(\lambda)}$, and $p, q \in (0,1)$, $\mathsf{PRF\text{-}PRC}[\mathcal{F}, p, q]$ is sound.*

*Proof.* Fix $\lambda \in \mathbb{N}$, and write $n = n(\lambda), \ell = \ell(\lambda), m = m(\lambda), N = N(\lambda)$. Fix any $y \in \{0,1\}^N$. Let $x_1, \ldots, x_m \in \{0,1\}^n, w_1, \ldots, w_m \in \{0,1\}$ be defined as in $\mathsf{Decode}(1^\lambda, \mathsf{sk}, y)$, i.e., $((x_1, w_1), \ldots, (x_m, w_m)) = ((\pi^{-1} \circ y) \oplus z)_{1:(n+1)m}$, where $\mathsf{sk} = (s, z, \pi)$. Since $\pi, z$ are uniformly random in their respective domains, it follows that $W = \sum_{j=1}^m \mathbb{1}\{w_j = F_s(x_j)\}$ is distributed as $\mathrm{Bin}(m, 1/2)$ for any fixed $s$, meaning that, for any $\delta \in (0,1)$, $\Pr_{z,\pi}(W > m/2 + \sqrt{m \log 1/\delta}) \leq \delta$. Choosing $\delta = 2^{-\log^2 m} \leq \mathsf{negl}(\lambda)$ yields that $\Pr_{\mathsf{sk} \leftarrow \mathsf{KeyGen}(1^\lambda)}(W > m/2 + \log(m)\sqrt{m}) \leq \mathsf{negl}(\lambda)$, as desired. $\qquad \square$

**Lemma C.7** (Undetectability). *Fix $p, q \in [0, 1/2)$. Suppose that $\mathcal{F} = \{F_s\}_s$ is a weak PRF on $\{0,1\}^{n(\lambda)}$ with noise level $q$ (Definition 3.1), indexed by $s \in \{0,1\}^{\ell(\lambda)}$. Then $\mathsf{PRF\text{-}PRC}[\mathcal{F}, p, q]$ is undetectable.*

*Proof.* Fix $\lambda \in \mathbb{N}$, and write $n = n(\lambda), \ell = \ell(\lambda), m = m(\lambda), N = N(\lambda)$. Consider any probabilistic polynomial-time adversary $\mathsf{Adv}$. Let $\mathsf{sk} = (s, z, \pi) \leftarrow \mathsf{KeyGen}(1^\lambda)$ denote the secret key for the PRC. Suppose that the adversary makes a total of $Q$ queries to $\mathsf{Encode}(1^\lambda, \mathsf{sk}, \emptyset)$; its view consists of $Q$ tuples $\pi \circ (((x_1^{(i)}, F_s(x_1^{(i)}) \oplus e_1^{(i)}), \ldots, (x_m^{(i)}, F_s(x_m^{(i)}) \oplus e_m^{(i)})) \oplus z)$, for $i \in [Q]$. If $\mathsf{Adv}$ satisfies

$$\left| \Pr_{\mathsf{sk} \sim \mathsf{KeyGen}(1^\lambda)} \left( \mathsf{Adv}^{\mathsf{Encode}(1^\lambda, \mathsf{sk}, \emptyset)}(1^\lambda) = 1 \right) - \Pr_{\mathcal{U}} \left( \mathsf{Adv}^{\mathcal{U}}(1^\lambda) = 1 \right) \right| \geq \lambda^{-c} \qquad (4)$$

for some constant $c > 0$, then we could construct an adversary $\mathsf{Adv}'$ which violates security of $F_s$ (namely, Definition 3.1), as follows: $\mathsf{Adv}'$ makes $Qm$ calls to $F_s(\cdot)$ on random inputs, which gives it samples $(x_j^{(i)}, F_s(x_j^{(i)}) \oplus e_j^{(i)})$ for $j \in [m], i \in [Q]$, draws a uniformly random permutation $\pi : [N] \rightarrow [N]$ and a uniformly random string $z \in \{0,1\}^N$, and outputs the result of $\mathsf{Adv}$ given the $Q$ tuples $y^{(i)} := \pi \circ (((x_1^{(i)}, F_s(x_1^{(i)}) \oplus e_1^{(i)}), \ldots, (x_m^{(i)}, F_s(x_m^{(i)}) \oplus e_m^{(i)}), r^{(i)}) \oplus z)$, where $r^{(i)}$ are uniformly random strings of length $N - (n+1)m$ (for $i \in [Q]$). Note that if the $mQ$ calls made by $\mathsf{Adv}'$ were generated by a random function, then the strings $y^{(i)}$ are uniformly random in $\{0,1\}^N$, conditioned on the event that $x_j^{(i)}$ are all distinct, for $j \in [m], i \in [Q]$. The probability that all $x_j^{(i)}$ are distinct is at least $1 - mQ/2^n \geq 1 - \mathsf{negl}(\lambda)$, meaning that, by (4), $\mathsf{Adv}'$ achieves advantage $\lambda^{-c} - \mathsf{negl}(\lambda)$ on distinguishing a random element of the family $\{F_s\}_s$ from a uniformly random function, thus contradicting the security of $\mathcal{F}$. $\qquad \square$

Given $N, t \in \mathbb{N}$, let $\mathcal{J}_{N,t}$ denote the distribution of $t$-tuples $J = (j_1, \ldots, j_t)$ where $j_1, \ldots, j_t$ are drawn uniformly from $[N]$ *without replacement*.

**Lemma C.8** (Robustness). *Fix $p, q \in [0, 1/2)$, and that $\mathcal{F} = \{F_s\}_{s \in \{0,1\}^{\ell(\lambda)}, \lambda \in \mathbb{N}}$ is a $\tau(\lambda)$-local function family. Then $\mathsf{PRF\text{-}PRC}[\mathcal{F}p, q]$ is robust to any $p$-weakly-substitution-bounded channel.*

*Proof.* Fix $\lambda \in \mathbb{N}$, and write $n = n(\lambda), \ell = \ell(\lambda), m = m(\lambda), N = N(\lambda)$. Let $\mathcal{E}$ be a $p$-weakly-substitution-bounded channel over $\{0,1\}$. Given $x_j, e_j$ ($j \in [m]$) drawn as in Line 6 of Algorithm 2 and a uniformly random string $r \sim \mathrm{Unif}(\{0,1\}^{N-(n+1)m})$, consider the string $a := ((x_1, F_s(x_1) \oplus e_1), \ldots, (x_m, F_s(x_m) \oplus e_m), r)$, and note that the output of $\mathsf{Encode}(1^\lambda, \mathsf{sk}, \emptyset)$ given that $x_1, \ldots, x_m, e_1, \ldots, e_m, r$ are sampled is $\bar{y} := \pi \circ (a \oplus z)$. Given $v \in \{0,1\}^n$, we let,

with slight abuse of notation, $\mathcal{E}(v) \in \{0,1\}^n$ denote a random variable drawn from the eponymous distribution $\mathcal{E}(v) \in \Delta(\{0,1\}^n)$. We have

$$u := (\pi^{-1} \circ \mathcal{E}(\bar{y})) \oplus z = (\pi^{-1} \circ \mathcal{E}(\pi \circ (a \oplus z))) \oplus z$$
$$= \pi^{-1} \circ ((\pi \circ z) \oplus \mathcal{E}((\pi \circ a) \oplus (\pi \circ z))).$$

Since $z$ and $\pi$ are chosen uniformly at random in their respective domains and independent of $a$, it follows that the distribution of $u$ is the same as the distribution of

$$u' := \pi^{-1} \circ ((\pi \circ a) \oplus v \oplus \mathcal{E}(v)) = a \oplus (\pi^{-1} \circ (v \oplus \mathcal{E}(v))),$$

where $v \sim \mathrm{Unif}(\{0,1\}^n)$ is uniformly at random and independent of $\pi, a$. (In particular, we have used the reparametrization that sets $v := (\pi \circ a) \oplus (\pi \circ z)$. )

Let us unpack $(\pi^{-1} \circ (v \oplus \mathcal{E}(v)))_{1:(n+1)m} = ((x_1', w_1'), \ldots, (x_m', w_m'))$, where $x_j' \in \{0,1\}^m, w_j' \in \{0,1\}$ for $j \in [m]$. Note that the distribution of the statistic $W$ computed in $\mathsf{Decode}(1^\lambda, \mathsf{sk}, \mathcal{E}(\bar{y}))$ is exactly the distribution of

$$\sum_{j=1}^m \mathbb{1}\{w_j' \oplus e_j \oplus F_s(x_j) = F_s(x_j \oplus x_j')\}. \tag{5}$$

By our assumption that the family $F_s$ is $\tau$-local, for each $s \in \{0,1\}^\ell$ there is some $J_s \in [n]^\tau$ and $G_s : \{0,1\}^\tau \to \{0,1\}$ so that $F_s(x) = G_s(x_{J_s})$ for all $x \in \{0,1\}^n$. Define $t = \tau + 1$ and $G_s' : \{0,1\}^t \to \{0,1\}$ by $G_s'(x) = G_s(x_1, \ldots, x_{t-1}) \oplus x_t$.

For each $j \in [m]$ and fixed $s \in \{0,1\}^\ell$, we have that

$$\Pr\left(w_j' \oplus e_j \oplus F_s(x_j) = F_s(x_j \oplus x_j')\right) = (1-2q) \cdot \Pr\left(F_s(x_j) = w_j' \oplus F_s(x_j \oplus x_j')\right) + q$$
$$= (1-2q) \cdot \Pr\left(G_s'((x_j)_{J_s}, 0) = G_s'((x_j)_{J_s} \oplus (x_j')_{J_s}, w_j')\right) + q, \tag{6}$$

where the probability is taken over $x_j \sim \mathrm{Unif}(\{0,1\}^n), e_j \sim \mathrm{Ber}(q)$, $z \sim \mathrm{Unif}(\{0,1\}^n)$, the uniformly random permutation $\pi$, and the randomness in $\mathcal{E}$ (which together determine $w_j', x_j'$). We now apply Lemma C.9 to the function

$$(z_1, \ldots, z_t) \mapsto \mathbb{1}\{G_s'((x_j)_{J_s}, 0) = G_s'((z_1, \ldots, z_{t-1}) \oplus (x_j)_{J_s}, z_t)\}.$$

Note that, for fixed $v, \mathcal{E}(v)$, since $\pi$ is chosen uniformly at random (and independent of $v, \mathcal{E}(v), a$), the distribution of $((x_j')_{J_s}, w_j')$ is exactly the distribution of $y_J$ for $J \sim \mathcal{J}_{N,t}$, for $y = v \oplus \mathcal{E}(v)$. Let us write $p(y) := \mathrm{wt}(y)/N$, for $y \in \{0,1\}^N$. Thus, for any fixed $s$, we have

$$\Pr\left(G_s'((x_j)_{J_s}, 0) = G_s'((x_j)_{J_s} \oplus (x_j')_{J_s}, w_j')\right)$$
$$= \mathbb{E}_{x_j, v, \mathcal{E}(v)} \mathbb{E}_{\substack{J \sim \mathcal{J}_{N,t} \\ y = v \oplus \mathcal{E}(v)}} \left[\mathbb{1}\{G_s'((x_j)_{J_s}, 0) = G_s'(((x_j)_{J_s}, 0) \oplus \bar{y}_J)\} \mid x_j, v, \mathcal{E}(v)\right]$$
$$\geq \mathbb{E}_{x_j, v, \mathcal{E}(v)} \mathbb{E}_{\bar{x} \sim \mathrm{Ber}(p(v \oplus \mathcal{E}(v)))^t} \left[\mathbb{1}\{G_s'((x_j)_{J_s}, 0) = G_s'(((x_j)_{J_s}, 0) \oplus \bar{x})\}\right] - \frac{2t}{\sqrt{N-t}}$$
$$= \mathbb{E}_{v, \mathcal{E}(v)} \mathbb{E}_{x_j} \mathbb{E}_{\bar{x} \sim \mathrm{Ber}(p(v \oplus \mathcal{E}(v)))^t} \mathbb{1}\{G_s((x_j)_{J_s}) = G_s((x_j)_{J_s} \oplus (\bar{x}_1, \ldots, \bar{x}_{t-1})) \oplus \bar{x}_t\} - \frac{2t}{\sqrt{N-t}}$$
$$= \mathbb{E}_{v, \mathcal{E}(v)} \left[p(v \oplus \mathcal{E}(v)) + (1 - p(v \oplus \mathcal{E}(v))) \mathbb{E}_{x_j} \mathbb{E}_{\bar{x} \sim \mathrm{Ber}(p(v \oplus \mathcal{E}(v)))^{t-1}} \mathbb{1}\{G_s((x_j)_{J_s}) = G_s((x_j)_{J_s} \oplus (\bar{x}))\}\right]$$
$$\quad - \frac{2t}{\sqrt{N-t}}$$
$$= \mathbb{E}_{v, \mathcal{E}(v)} \left[p(v \oplus \mathcal{E}(v)) + (1 - 2p(v \oplus \mathcal{E}(v)) \cdot (1 - \mathbf{NS}_{p(v \oplus \mathcal{E}(v))}[G_s])\right] - \frac{2t}{\sqrt{N-t}}$$
$$\geq \mathbb{E}_{v, \mathcal{E}(v)} \left[p(v \oplus \mathcal{E}(v)) + (1 - 2p(v \oplus \mathcal{E}(v))) \cdot \left(\frac{1}{2} + \frac{1}{2} \cdot (1 - 2 \cdot p(v \oplus \mathcal{E}(v))^\tau)\right)\right] - \frac{2t}{\sqrt{N-t}}$$
$$= \mathbb{E}_{v, \mathcal{E}(v)} \left[\frac{1}{2} + \frac{1}{2} \cdot (1 - 2 \cdot p(v \oplus \mathcal{E}(v)))^{\tau+1}\right] - \frac{2t}{\sqrt{N-t}}$$
$$\geq \frac{1}{2} + \frac{1}{2} \cdot (1 - 2p)^{\tau+1} - \frac{2t}{\sqrt{N-t}} - \mathsf{negl}(N), \tag{7}$$

where the first inequality uses Lemma C.9, the fourth equality uses the fact that $x_j \sim \mathrm{Unif}(\{0,1\}^n)$ independently of $v, \mathcal{E}(v)$ and the definition of noise sensitivity, the second equality uses Corollary C.5, and the final inequality uses the fact that $\mathrm{wt}(v \oplus \mathcal{E}(v)) \leq pN$ with probability $1 - \mathsf{negl}(N)$ since the marginal distribution of $v$ is $\mathrm{Unif}(\{0,1\}^n)$ and $\mathcal{E}$ is $p$-weakly-substitution-bounded. Since we have $\frac{2t}{\sqrt{N-t}} + \mathsf{negl}(N) \leq \frac{1}{4} \cdot (1-2p)^{\tau+1}$ by our choice of $N$ in (3) (which ensures that, as long as $C_0$ is sufficiently large $N \geq C_0 \cdot (1-2p)^{-4\log(n)}$, and in particular that $N - t \geq 16t(1-2p)^{-\tau-1}$ as long as the security parameter $\lambda$ is sufficiently large) , it follows from Eqs. (6) and (7) that for each $j \in [m]$, for sufficiently large $\lambda$,

$$\Pr(w_j' \oplus e_j \oplus F_s(x_j) = F_s(x_j \oplus x_j')) \geq (1-2q) \cdot \left( \frac{1}{2} + \frac{1}{4} \cdot (1-2p)^{\tau+1} \right) + q \geq \frac{1}{2} + \frac{1}{4}(1-2q)(1-2p)^{\tau+1}. \tag{8}$$

Finally, we use Dobrushin's inequality to analyze the concentration of the sum (5); we utilize the notation of Appendix F.1 (in particular the influences defined in (43)). For each $j \in [m]$, define $\Gamma_j = (x_j, e_j, w_j', x_j')$. Let $P_{\Gamma_1, \ldots, \Gamma_m}$ denote the joint law of $(\Gamma_1, \ldots, \Gamma_m)$, and $P_{\Gamma_i | \Gamma_{-i}}$ denote the conditional law of $\Gamma_i$ conditioned on $\Gamma_{-i}$. For any distinct $i, j \in [m]$, we have that

$$I_{j \to i}(\Gamma_{1:m}) = \max_{\gamma_{-i-j}, \gamma_j, \gamma_j'} d_{\mathsf{TV}}(P_{\Gamma_i | \Gamma_{-i}}(\cdot \mid \gamma_j, \gamma_{-i-j}), P_{\Gamma_i | \Gamma_{-i}}(\cdot \mid \gamma_j', \gamma_{-i-j})) \leq \frac{(n+1)^2}{N - m(n+1)},$$

since if, for any $i \in [m]$, we condition on $x_{-i}, e_{-i}, v \oplus \mathcal{E}(v)$, and $\mathcal{S} := \{\pi^{-1}((a-1)(n+1) + b) \ : \ a \in [m] \backslash \{i\}, b \in [n+1]\}$, the distribution of $\Gamma_i = (x_i, e_i, w_i', x_i')$ is given as follows: $x_i \sim \mathrm{Unif}(\{0,1\}^n)$, $e_i \sim \mathrm{Ber}(q)$, and $(w_i', x_i')$ is distributed (independently of $x_i, e_i$) as a tuple of $n+1$ distinct elements of $v \oplus \mathcal{E}(v)$ which are not indexed by coordinates in $\mathcal{S}$. Moreover, changing the value of $x_j, e_j$, and $\{\pi^{-1}((j-1)(n+1) + b) \ : \ b \in [n+1]\}$ changes only $n+1$ elements of $\mathcal{S}$, so by Lemma C.10 with $k = n+1$ and the data processing inequality for total variation distance, changes the conditional distribution of $(w_i', x_i')$ by at most $\frac{(n+1)^2}{N-m(n+1)}$.

Thus $\sum_{j \in [m] \backslash \{i\}} I_{j \to i}(\Gamma_{1:m}) \leq \frac{m(n+1)^2}{N-m(n+1)} \leq 1/2$ and $\sum_{i \in [m] \backslash \{j\}} I_{j \to i}(\Gamma_{1:m}) \leq \frac{m(n+1)^2}{N-m(n+1)} \leq 1/2$ since we have chosen $N = 3m(n+1)^2$ in (3). It then follows from Theorem F.3 that for any $\delta > 0$, with probability $1 - \delta$ over the draw of $\Gamma_{1:m}$,

$$W = \sum_{j=1}^m \mathbb{1}\{w_j' \oplus e_j \oplus F_s(x_j) = F_s(x_j \oplus x_j')\} \geq \sum_{j=1}^m \Pr(w_j' \oplus e_j \oplus F_s(x_j) = F_s(x_j \oplus x_j')) - \sqrt{4m\ln(2/\delta)}. \tag{9}$$

Choosing $\delta = 2\exp(-\ln^2(m)) \leq \mathsf{negl}(m) \leq \mathsf{negl}(\lambda)$, and combining Eqs. (8) and (9) and the choice of $m$ in (3), we see that as long as the constant $C_0$ in (3) is sufficiently large,

$$\Pr\left( W \geq \frac{m}{2} + \ln(m)\sqrt{m} \right) \geq \Pr\left( W \geq \frac{m}{2} + m \cdot (1-2q)(1-2p)^{\tau+1}/4 - \ln(m)\sqrt{4m} \right) \geq 1 - \delta.$$

(In particular, we have used that our choice of $m$ ensures that $m \cdot (1-2q)(1-2p)^{\tau+1}/4 \geq 3\ln(m)\sqrt{m}$.) Since $\mathsf{Decode}(1^\lambda, \mathsf{sk}, y)$ outputs $\emptyset$ exactly when $W \geq \frac{m}{2} + \log(m)\sqrt{m}$, we have established robustness, as desired. $\square$

**Lemma C.9.** *Fix* $N, k \in \mathbb{N}$ *and let* $y \in \{0,1\}^N$ *be given with* $\mathrm{wt}(y) = k$. *Fix* $t \in \mathbb{N}$*, and let* $f : \{0,1\}^t \to \{0,1\}$ *be a given function. Then*

$$\left| \mathbb{E}_{J \sim \mathcal{J}_{N,t}}[f(y_J)] - \mathbb{E}_{x \sim \mathrm{Ber}(k/N)^t}[f(x)] \right| \leq \frac{2t}{\sqrt{N-t}}.$$

*Proof.* It suffices to upper bound the total variation distance between the distribution of $y_J$ and the distribution of $x \sim \mathrm{Ber}(k/N)^t$. By symmetry, this total variation distance is the total variation distance between $\mathrm{Bin}(t, k/N)$ and $\mathrm{Hyp}(N, k, t)$, where $\mathrm{Hyp}$ denotes the hypergeometric distribution (so that, in particular, $W \sim \mathrm{Hyp}(N, k, t)$ satisfies $\Pr(W = w) = \frac{\binom{k}{w}\binom{N-k}{t-w}}{\binom{N}{t}}$). By [RS23, Theorem 1] (restated in Lemma G.2), this total variation distance is bounded above by $\frac{2t}{\sqrt{N-t}}$. $\square$

**Lemma C.10.** *Let $\mathcal{S}_0, \mathcal{S}_1$ be sets of size $N$ so that $|\mathcal{S}_0 \cap \mathcal{S}_1| = N - k$, for some $k < N$. Let $n < N$ be given. For $b \in \{0, 1\}$ let $\mathcal{J}_b$ denote the distribution of a tuple $J = (j_1, \ldots, j_n)$ of $n$ elements of $\mathcal{S}_b$ drawn uniformly without replacement. Then $d_{\mathsf{TV}}(\mathcal{J}_0, \mathcal{J}_1) \leq \frac{nk}{N-n}$.*

*Proof.* For any tuple $J$ all of whose elements belong to $\mathcal{S}_0 \cap \mathcal{S}_1$ (which we write as $J \subset \mathcal{S}_0 \cap \mathcal{S}_1$), we have $\mathcal{J}_0(J) = \mathcal{J}_1(J)$ by symmetry. Thus we have

$$d_{\mathsf{TV}}(\mathcal{J}_0, \mathcal{J}_1) \leq \Pr_{J \sim \mathcal{J}_0} (J \not\subset \mathcal{S}_0 \cap \mathcal{S}_1) = 1 - \frac{\binom{N-k}{n}}{\binom{N}{n}} = 1 - \frac{(N-k) \cdots (N-k-n+1)}{N \cdots (N-n+1)}$$

$$\leq 1 - \left(\frac{N-n-k}{N-n}\right)^n \leq \frac{nk}{N-n}.$$

$\square$

# D   Insertion/Deletion PRCs from substitution PRCs

Suppose that $\mathsf{PRC}_{\mathsf{Sub}}$ is a PRC which is robust to $(1/2 - p_0)$-substitution-bounded channels, for some $p_0 > 0$. Choose $\rho := 2C_{D.8}/p_0$, where $C_{D.8}$ is the constant defined in Lemma D.8. Note that $\mathsf{PRC}_{\mathsf{Idx}}[\mathsf{PRC}_{\mathsf{Sub}}, \rho]$ (defined in Algorithm 1) has block length $m(\lambda) := \lceil n(\lambda) \cdot \ln(2) \rceil$ and alphabet size $|\Sigma(\lambda)| = q(\lambda) = \rho \cdot n(\lambda) = \frac{2C_{D.8}}{p_0} \cdot n(\lambda)$. Thus, to prove Theorem 4.1, it suffices to show that $\mathsf{PRC}_{\mathsf{Idx}}[\mathsf{PRC}_{\mathsf{Sub}}, \rho]$ satisfies undetectability (Lemma D.3), soundness (Lemma D.1), and robustness to all $(1 - C_{\mathsf{rob}} p_0, p_0)$-edit-bounded channels (Lemma D.8), where $C_{\mathsf{rob}}$ is a constant defined in Lemma D.8.

**Additional notation.** Fix $q, m \in \mathbb{N}$. For integers $j \geq 1$, we define $\mathsf{Unique}_j(y) := \{a \in [q] : |\{i : y_i = a\}| = j\}$, i.e., $\mathsf{Unique}_j(y)$ is the set of elements $a \in [q]$ so that exactly $j$ elements of $y$ are equal to $a$. Given $j \in \mathbb{N}$, we define $\mathsf{Unique}_{\geq j}(y) = \bigcup_{j' \geq j} \mathsf{Unique}_{j'}(y)$. Note that $\mathsf{Unique}(y) = \mathsf{Unique}_{\geq 1}(y)$.

**Lemma D.1** (Soundness). *Let $\mathsf{PRC}_{\mathsf{Sub}}$ be a PRC for substitutions. Then the PRC $\mathsf{PRC}_{\mathsf{Idx}}[\mathsf{PRC}_{\mathsf{Sub}}]$ in Algorithm 1 is sound.*

*Proof.* Fix $\lambda \in \mathbb{N}$, and consider $z \in [q(\lambda)]^{m(\lambda)}$. Define $y' \in \{0, 1\}^{n(\lambda)}$ as in Line 12 of Algorithm 1. Since $\mathsf{PRC}_{\mathsf{Sub}}$ is sound, we have

$$\Pr_{(\mathsf{sk}, \psi) \sim \mathsf{KeyGen}(1^\lambda)} \left(\mathsf{Decode}(1^\lambda, (\mathsf{sk}, \psi), z) = \perp\right) = \Pr_{\mathsf{sk} \sim \mathsf{KeyGen}_{\mathsf{Sub}}(1^\lambda)} \left(\mathsf{Decode}_{\mathsf{Sub}}(1^\lambda, \mathsf{sk}, y') = \perp\right) \geq 1 - \mathsf{negl}(\lambda).$$

$\square$

To establish undetectability, we first need the following lemma which states that $\mathsf{PerturbDifference}(n, m, y^0)$ outputs a uniformly random string in $[n]^m$ when its input $y^0$ is uniform over $\{0, 1\}^n$.

**Lemma D.2.** *Given positive integers $m \leq n$, the distribution of $z \leftarrow \mathsf{PerturbDifference}(n, m, y^0)$ (Algorithm 1), for $y^0 \sim \mathrm{Unif}(\{0, 1\}^n)$, is exactly $\mathrm{Unif}([n]^m)$.*

*Proof.* Suppose $y^0 \sim \mathrm{Unif}(\{0, 1\}^n)$, $y^1 \sim \mathrm{Unif}([n]^m)$, and write $\mathcal{S}^0 = \{i \in [n] : y_i^0 = 1\}$, $\mathcal{S}^1 = \mathsf{Unique}(y^1)$ as in $\mathsf{PerturbDifference}$.

Given $y \in [n]^m$, denote its *frequency mapping* $\mathsf{freq}(y) = \mathsf{f} : [m] \to \mathbb{N}_{\geq 0}$ by $\mathsf{f}(j) = |\mathsf{Unique}_j(y)|$ for $j \in [m]$. Note that $\mathsf{freq}(y^1) = \mathsf{freq}(y)$ with probability 1, where $y$ denotes the output string of $\mathsf{PerturbDifference}$ (this holds since the maps $\sigma, \tau$ are necessarily injective). Thus, the distribution of $\mathsf{freq}(y)$ is exactly the distribution of $\mathsf{freq}(y^1)$ for $y^1 \sim \mathrm{Unif}([n]^m)$.

Next, we claim that any two strings $z, z' \in [n]^m$ with $\mathsf{freq}(z) = \mathsf{freq}(z')$ have equal probability of being output by $\mathsf{PerturbDifference}$. To see this, we may choose a permutation $\pi : [n] \to [n]$ and $\sigma : [m] \to [m]$ so that $\pi(z_{\sigma(i)}) = z_i'$ for $i \in [m]$. Next, it is straightforward to see from the definition of $\mathsf{PerturbDifference}$ that the distribution of its output remains unchanged if its input $y^0$ is

replaced with the string $\tilde{y}^0$ defined by $\tilde{y}^0_{\pi(i)} := y^0_i$, $i \in [n]$, and the sample $y^1$ in Line 16 is replaced with the string $\tilde{y}^1$ defined by $\tilde{y}^1_i := \pi(y^1_{\sigma(i)})$, $i \in [m]$. The distribution of the sets $\mathcal{S}^0, \mathcal{S}^1 \subset [n]$ under this modified procedure is exactly the distribution of $\pi(\mathcal{S}^0), \pi(\mathcal{S}^1)$ when $\mathcal{S}^0, \mathcal{S}^1$ are drawn according to the original procedure. Thus, the probability of observing $z$ under the original execution of PerturbDifference is the same as the probability of observing $z'$ under this modified execution of PerturbDifference.

It follows from the two previous paragraphs that each string in $[n]^m$ has equal probability of being output by $\mathrm{PerturbDifference}(n, m, y^0)$, under $y^0 \sim \mathrm{Unif}([n]^m)$, as desired. $\qquad\square$

**Lemma D.3** (Undetectability). *Let* $\mathsf{PRC_{Sub}} = (\mathsf{KeyGen_{Sub}}, \mathsf{Encode_{Sub}}, \mathsf{Decode_{Sub}})$ *be a PRC for substitutions. Then the PRC* $\mathsf{PRC_{Idx}}[\mathsf{PRC_{Sub}}]$ *in Algorithm 1 is undetectable.*

*Proof.* Fix $\lambda \in \mathbb{N}$, and let us write $q := q(\lambda), n := n(\lambda)$. Consider any $\psi : [q] \to [n]$ which satisfies $|\psi^{-1}(j)| = q/n$ for each $j \in [n]$.

Given a string $y^0 \in \{0,1\}^n$, let $E_\psi(y^0) \in [q]^n$ denote the random variable which is the output of the following procedure: given $y^0$, let $y \leftarrow \mathsf{PerturbDifference}(n, m, y^0)$, then sample $z \in [q]^m$ as on Line 8 of Algorithm 1 (using $\psi$), and output the resulting string $z$.

**Claim D.4.** *For any fixed $\psi$ as above, the distribution of $E_\psi(y^0)$, for $y^0 \sim \mathrm{Unif}(\{0,1\}^n)$, is uniform on $[q]^m$.*

*Proof.* By Lemma D.2, the distribution of $y \leftarrow \mathsf{PerturbDifference}(n, m, y^0)$ is uniform on $[n]^m$. Since $|\psi^{-1}(j)| = q/n$ for each $j \in [n]$, it follows that the distribution of the output string $z$ is uniform on $[q]^m$. $\qquad\square$

Now consider any probabilistic polynomial-time adversary Adv, and suppose that

$$\left| \Pr_{(\mathsf{sk}, \psi) \leftarrow \mathsf{KeyGen}(1^\lambda)} \left( \mathsf{Adv}^{\mathsf{Encode}(1^\lambda, (\mathsf{sk}, \psi), \cdot)}(1^\lambda) = 1 \right) - \Pr_{\mathcal{U}} \left( \mathsf{Adv}^{\mathcal{U}}(1^\lambda) = 1 \right) \right| = \nu(\lambda)$$

for some function $\nu : \mathbb{N} \to \mathbb{R}_{\geq 0}$. We construct an adversary $\mathsf{Adv}'$ for the substitution PRC $\mathsf{PRC_{Sub}}$, as follows: $\mathsf{Adv}'$ first generates $\psi : [q] \to [n]$ conditioned on $|\psi^{-1}(j)| = q/n$ for each $j \in [n]$. $\mathsf{Adv}'$ then simulates Adv, where each time Adv calls its oracle $\mathcal{O}(\mathsf{m})$ for some message $\mathsf{m}$, $\mathsf{Adv}'$ performs the following. It calls $y^0 \leftarrow \mathcal{O}'(\mathsf{m})$ (using its oracle $\mathcal{O}'$ which is either a random oracle or $\mathsf{Encode_{Sub}}(1^\lambda, \mathsf{sk}, \mathsf{m})$), then applies $y \leftarrow \mathsf{PerturbDifference}(n, m, y^0)$, samples $z_j \sim \mathrm{Unif}(\{a : \psi(a) = y_j\})$ for each $j \in [m]$, and then uses $z$ as the simulated output for $\mathcal{O}(\mathsf{m})$. By definition of Encode in Algorithm 1, the adversary $\mathsf{Adv}'$ faithfully simulates the execution of $\mathsf{Adv}^{\mathsf{Encode}(1^\lambda, \mathsf{sk}, \mathsf{m})}$ when the oracle $\mathcal{O}'(\cdot)$ for $\mathsf{Adv}'$ is $\mathsf{Encode_{Sub}}(1^\lambda, \mathsf{sk}, \cdot)$. When the oracle $\mathcal{O}'$ for $\mathsf{Adv}'$ is a random oracle $\mathcal{U}$ (i.e., which outputs a uniformly random string $y^0 \sim \mathrm{Unif}(\{0,1\}^n)$), then Claim D.4 ensures that the adversary $\mathsf{Adv}'$ generates a uniformly random string in $[q]^m$. Thus, we have that

$$\left| \Pr_{\mathsf{sk} \leftarrow \mathsf{KeyGen_{Sub}}(1^\lambda)} \left( (\mathsf{Adv}')^{\mathsf{Encode_{Sub}}(1^\lambda, \mathsf{sk}, \cdot)}(1^\lambda) = 1 \right) - \Pr_{\mathcal{U}} \left( (\mathsf{Adv}')^{\mathcal{U}}(1^\lambda) = 1 \right) \right| = \nu(\lambda),$$

and since $\mathsf{PRC_{Sub}}$ is undetectable, we have $\nu(\lambda) = \mathsf{negl}(\lambda)$, meaning that which implies that $\mathsf{PRC_{Idx}}[\mathsf{PRC_{Sub}}]$ is undetectable. $\qquad\square$

### D.1 Lemmas for robustness

Next, we will establish several lemmas in the aim of showing robustness of $\mathsf{PRC_{Idx}}[\mathsf{PRC_{Sub}}, \rho]$. The below lemma bounds the number of replacements PerturbDifference has to perform in Line 20 or Line 23 of Algorithm 1.

**Lemma D.5.** *There is a sufficiently large constant $C_{D.5} \geq 1$ so that the following holds. Fix positive integers $n, m$ with $m = \lceil n \cdot \ln(2) \rceil$ and $\epsilon \in (0,1)$ so that $n \geq C_{D.5} \ln^2(1/\epsilon)/\epsilon^2$. For $z \in [n]^m$, write $D(z) \in \{0,1\}^n$ to be the string $D(z)_i := \mathbb{1}\{i \in \mathsf{Unique}(z)\}$. Then*

$$\Pr_{\substack{y^0 \sim \mathrm{Unif}(\{0,1\}^n) \\ z \leftarrow \mathsf{PerturbDifference}(n, m, y^0)}} \left( D_{\mathsf{Ham}}(y^0, D(z)) \geq \epsilon n \right) \leq 1 - \mathsf{negl}(n).$$

*Proof.* Given $y^0$, let $\mathcal{S}^0 = \{i \in [n] : y^0_i = 1\}$ be defined as in Line 15 of PerturbDifference. By a Chernoff bound, for any $\delta \in (0,1)$,

$$\Pr_{y^0 \sim \mathrm{Unif}(\{0,1\}^n)} \left( \left| |\mathcal{S}^0| - \frac{n}{2} \right| \geq \sqrt{n \log(2/\delta)} \right) \leq \delta \tag{10}$$

Now consider $y^1 \sim \mathrm{Unif}([n]^m)$ and $\mathcal{S}^1 = \mathsf{Unique}(y^1)$ as in Lines 15 and 16 of PerturbDifference. By our choice of $m, n$ and Lemma D.6 with $q = n$, $y^1$ is typical with probability $1 - \mathsf{negl}(m) \geq 1 - \mathsf{negl}(n)$, which implies that $\left| |\mathcal{S}^1| - \frac{n}{2} \right| \leq 2\sqrt{m}\ln(m) + 1$. Combining this fact with (10) and choosing $\delta = 2\exp(-\ln^2(n)) \leq \mathsf{negl}(n)$ gives that

$$\Pr_{y^0 \sim \mathrm{Unif}(\{0,1\}^n), y^1 \sim \mathrm{Unif}([n]^m)} \left( \left| |\mathcal{S}^0| - |\mathcal{S}^1| \right| \geq \epsilon n \right)$$

$$\leq \Pr_{y^0 \sim \mathrm{Unif}(\{0,1\}^n), y^1 \sim \mathrm{Unif}([n]^m)} \left( \left| |\mathcal{S}^0| - |\mathcal{S}^1| \right| \geq 2\sqrt{m}\ln(m) + 1 + \sqrt{n}\ln(n) \right) \leq \mathsf{negl}(n),$$

where we have used that $\epsilon n \geq 3\sqrt{n}\ln(n) + 1 \geq 2\sqrt{m}\ln(m) + 1 + \sqrt{n}\ln(n)$ as a result of our assumption that $n \geq \frac{C\ln^2(1/\epsilon)}{\epsilon^2}$ for a sufficiently large constant $C$. The conclusion of the lemma follows by noting that $\left| |\mathcal{S}^0| - |\mathcal{S}^1| \right| = D_{\mathsf{Ham}}(y^0, D(z))$ with probability 1. $\qquad\square$

**Definition D.1** (Typical string). Fix $q, m \in \mathbb{N}$. A string $z \in [q]^m$ is defined to be *typical* if the following inequalities hold:

$$q \cdot (1 - \exp(-m/q)) - 2\sqrt{m}\ln m \leq |\mathsf{Unique}(z)| \leq q \cdot (1 - \exp(-m/q)) + 2\sqrt{m}\ln m.$$

**Lemma D.6.** *Suppose that* $q \geq \sqrt{m}/\ln(m)$. *With probability* $1 - \mathsf{negl}(m)$, *a uniformly random string* $z \sim \mathrm{Unif}([q]^m)$ *is typical.*

*Proof.* Consider a string $z \in [q]^m$, and define

$$F_1(z) := |\mathsf{Unique}_{\geq 1}(z)| = \sum_{a=1}^{q} \mathbb{1}\{|\{i \in [m] : z_i = a\}| \geq 1\}.$$

Note that $F_1$ satisfies the bounded differences property with constants $c_i = 1$, for each $i \in [m]$. Note also that

$$\mathbb{E}_{z \sim \mathrm{Unif}([q]^m)}[F_1(z)] = q \cdot \left( 1 - \sum_{k=0}^{j-1} \binom{m}{k} \cdot q^{-k} \cdot (1 - 1/q)^{m-k} \right) = q \cdot \left( \sum_{k=j}^{m} \binom{m}{k} \cdot q^{-k} \cdot (1 - 1/q)^{m-k} \right).$$

We have that $\mathbb{E}[F_1(z)] = q \cdot (1 - (1 - 1/q)^m)$, and using the bounds $\exp(-1/q) \geq 1 - 1/q \geq \exp(-1/q - 1/q^2)$ for $q \geq 1$, we conclude that

$$q \cdot (1 - \exp(-m/q)) \leq \mathbb{E}[F_1(z)] \leq q \cdot (1 - \exp(-m/q - m/q^2)) \leq q \cdot (1 - \exp(-m/q)) + m/q.$$

By Theorem F.1, for any $\delta \in (0,1)$, we have that with probability $1 - \delta$ over $z \sim \mathrm{Unif}([q]^m)$,

$$|\mathbb{E}[F_1(z)] - F_1(z)| \leq \sqrt{m\ln(2/\delta)}.$$

Choose $\delta = 2\exp(-\ln^2(m))$ (so that $\delta \leq \mathsf{negl}(m)$), and note that our assumption that $q \geq \sqrt{m}/\ln m$ gives that $m/q \leq \sqrt{m\ln(2/\delta)}$. Combining the two displays above gives that with probability $1 - \mathsf{negl}(m)$ over $z \sim \mathrm{Unif}([q]^m)$, we have

$$q \cdot (1 - \exp(-m/q)) - 2\sqrt{m}\ln m \leq F_1(z) \leq q \cdot (1 - \exp(-m/q)) + 2\sqrt{m}\ln m.$$

$\qquad\square$

Given integers $n < q$ so that $q/n \in \mathbb{N}$, let $P^{\mathsf{ptn}}_{n,q}$ denote the uniform distribution over mappings $\psi : [q] \to [n]$ conditioned on the event that $|\psi^{-1}(j)| = q/n$ for each $j \in [n]$. For sets $\mathcal{S}, \mathcal{T} \subset \Omega$, define $\Delta(\mathcal{S}, \mathcal{T}) := (\mathcal{S} \backslash \mathcal{T}) \cup (\mathcal{T} \backslash \mathcal{S})$.

**Lemma D.7.** *There is a constant $C_{D.7} > 0$ so that the following holds. Consider $n, \rho, q \in \mathbb{N}$ satisfying $q/\rho = n$, let $\mathcal{Z}_1, \mathcal{Z}_2 \subset [q]$ be given, and write $\mathcal{Z} := \mathcal{Z}_1 \cap \mathcal{Z}_2$. Define $Z := |\mathcal{Z}|, Z_1 := |\mathcal{Z}_1|, Z_2 := |\mathcal{Z}_2|$, and suppose that $0 \le \epsilon \le p \le 1/10$ are given so that*

$$\frac{8}{\epsilon} \le \rho \le n^{1/4}, \quad \frac{Z_1}{n} \in [\ln(2) - \epsilon, \ln(2) + \epsilon], \quad \frac{Z_2}{n} \le 2\ln(2) + \epsilon, \quad Z \ge pn, \quad n \ge \frac{C_{D.7}}{\epsilon^3}. \tag{11}$$

*Then*

$$\Pr_{\psi \sim P_{n,q}^{\text{ptn}}} \left( |\Delta(\psi(\mathcal{Z}_1), \psi(\mathcal{Z}_2))| \ge n \cdot \left( \frac{1}{2} - \frac{p}{5} + 23\epsilon \right) \right) \le \text{negl}(n).$$

*Proof.* The fact that $2/\epsilon \le \rho$ and $\max\{Z_1/n, Z_2/n\} \le 2$ implies that $\max\{Z_1/\rho, Z_2/\rho\} \le \epsilon n$. Let us write $\zeta_1 := \exp(-Z_1/n), \zeta_2 := \exp(-Z_2/n)$ so that $\zeta_1 \in [(1-\epsilon)/2, 1/2 + \epsilon]$ and $\zeta_2 \in [(1-\epsilon)/4, 1]$.

Note that the mapping $\psi : [q] \to [n]$ is specified by the random variables $\psi^{-1}(1), \psi^{-1}(2), \dots, \psi^{-1}(n) \subset [q]$. Let us define

$$F(\psi) := |\Delta(\psi(\mathcal{Z}_1), \psi(\mathcal{Z}_2))| = \sum_{i=1}^{n} \mathbb{1}\{i \in \Delta(\psi(\mathcal{Z}_1), \psi(\mathcal{Z}_2))\} = \sum_{i=1}^{n} G(\psi^{-1}(i); \mathcal{Z}_1, \mathcal{Z}_2),$$

where $G(\mathcal{T}; \mathcal{Z}_1, \mathcal{Z}_2) \in \{0, 1\}$ (for some $\mathcal{T} \subset [q]$) is defined to be equal to 1 if and only if either (a) $\mathcal{T} \cap \mathcal{Z}_1 \ne \emptyset$ but $\mathcal{T} \cap \mathcal{Z}_2 = \emptyset$ or (b) $\mathcal{T} \cap \mathcal{Z}_2 \ne \emptyset$ but $\mathcal{T} \cap \mathcal{Z}_1 = \emptyset$.

**Step 1: Bounding the expectation of $F$.** Fix any $i \in [n]$, and note that $\psi^{-1}(i) \subset [q]$ is a uniformly random subset of size $\rho$. Note that

$$\Pr_{\psi \sim P_{n,q}^{\text{ptn}}} (\psi^{-1}(i) \cap \mathcal{Z}_1 \ne \emptyset, \ \psi^{-1}(i) \cap \mathcal{Z}_2 = \emptyset)$$

$$= \Pr_{\psi \sim P_{n,q}^{\text{ptn}}} (\psi^{-1}(i) \cap \mathcal{Z}_1 \ne \emptyset \mid \psi^{-1}(i) \cap \mathcal{Z}_2 = \emptyset) \cdot \Pr_{\psi \sim P_{n,q}^{\text{ptn}}} (\psi^{-1}(i) \cap \mathcal{Z}_2 = \emptyset). \tag{12}$$

By [Lemma D.9](), we have

$$\Pr_{\psi \sim P_{n,q}^{\text{ptn}}} (\psi^{-1}(i) \cap \mathcal{Z}_2 = \emptyset) \le \exp(-\rho Z_2/q) = \exp(-Z_2/n) = \zeta_2 \tag{13}$$

and

$$\Pr_{\psi \sim P_{n,q}^{\text{ptn}}} (\psi^{-1}(i) \cap \mathcal{Z}_1 \ne \emptyset \mid \psi^{-1}(i) \cap \mathcal{Z}_2 = \emptyset)$$

$$\le 1 - \exp\left( -\frac{\rho(Z_1 - Z)}{q - Z_2 - \rho} - \frac{\rho(Z_1 - Z)^2}{(q - Z_2 - \rho)^2} \right)$$

$$\le 1 - \exp\left( -\frac{Z_1 - Z}{n - Z_2/\rho - 1} - \frac{\rho Z_1^2}{(q - Z_2 - \rho)^2} \right)$$

$$\le 1 - \exp\left( -(1 + 4\epsilon)Z_1/n \right) \cdot \exp(p) \cdot \exp(-Z_1^2(1 + 4\epsilon)^2/(\rho n^2))$$

$$\le 1 - \exp(\ln(\zeta_1) - 8\epsilon) \cdot \exp(p) \cdot \exp(-8/\rho)$$

$$\le 1 - \zeta_1 \cdot (1 + p) \cdot \exp(-9\epsilon) \le 1 - \zeta_1 \cdot (1 + p) \cdot (1 - 9\epsilon) \le (1 - \zeta_1) - p\zeta_1 + 10\epsilon, \tag{14}$$

where we have used the fact that, conditioned on $\psi^{-1}(i) \cap \mathcal{Z}_2 = \emptyset$, $\psi^{-1}(i)$ is distributed as a uniformly random subset of $[q] \backslash \mathcal{Z}_2$, which has size $q - Z_2$. Moreover, the third inequality above uses the upper bound $Z_2/\rho \le \epsilon n$ and the lower bound $Z \ge pn$ in [(11)](), the fourth inequality uses the upper bound $Z_1/n \le 2$ from [(11)](), and the remaining inequalities simplify and use the fact that $\epsilon, p \in (0, 1/10)$ and $8/\rho \le \epsilon$.

Using Eqs. (12) to (14) together with a symmetrical argument to bound $\Pr(\psi^{-1}(i) \cap \mathcal{Z}_2 \neq \emptyset, \psi^{-1}(i) \cap \mathcal{Z}_1 = \emptyset)$,[10] we see that, for each $i \in [n]$,

$$
\begin{aligned}
\mathbb{E}_{\psi \sim P_{n,q}^{\mathsf{ptn}}}[G(\psi^{-1}(i); \mathcal{Z}_1, \mathcal{Z}_2)] &\leq \zeta_2 \cdot ((1 - \zeta_1) - p\zeta_1 + 10\epsilon) + \zeta_1 \cdot ((1 - \zeta_2) - p\zeta_2 + 10\epsilon) \\
&\leq \zeta_1 + \zeta_2 - 2\zeta_1\zeta_2(1 + p) + 20\epsilon \\
&\leq \left(\frac{1}{2} + \epsilon\right) + \zeta_2 - (1 - \epsilon)\zeta_2 \cdot (1 + p) + 20\epsilon \\
&= \left(\frac{1}{2} + \epsilon\right) + \epsilon\zeta_2 - \zeta_2(1 - \epsilon) \cdot p + 20\epsilon \\
&\leq \frac{1}{2} - \frac{p}{5} + 22\epsilon,
\end{aligned}
$$

where the second inequality uses that $\zeta_1, \zeta_2 \leq 1$, the third inequality uses that $\zeta_1 \in [(1-\epsilon)/2, 1/2+\epsilon]$, and the final inequality uses that $\zeta_2 \in [(1 - \epsilon)/4, 1]$. It follows that

$$
\mathbb{E}_{\psi \sim P_{n,q}^{\mathsf{ptn}}}[F(\psi)] \leq n \cdot \left(\frac{1}{2} - \frac{p}{5} + 22\epsilon\right). \tag{15}
$$

**Step 2: Concentration of $F$ to its expectation.** Consider any subset $\mathcal{H} \subset [n]$ of size $H_0 := |\mathcal{H}|$ satisfying $4\rho H_0 \leq n$, and define $F_{\mathcal{H}}(\psi) := \sum_{i \in \mathcal{H}} G(\psi^{-1}(i); \mathcal{Z}_1, \mathcal{Z}_2)$. Note that $F_{\mathcal{H}}$ satisfies the bounded differences property with respect to the random variables $\psi^{-1}(i)$, $i \in \mathcal{H}$, with constants $c_i = 1$ for each $i \in \mathcal{H}$. For any distinct $i, j \in \mathcal{H}$, we have

$$
\max_{\substack{\psi^{-1}(k) \subset [q],\ k \in \mathcal{H}\setminus\{i,j\} \\ \psi^{-1}(j), \tilde{\psi}^{-1}(j) \subset [q]}} d_{\mathsf{TV}}(P_{n,q}^{\mathsf{ptn}}(\psi^{-1}(i) = \cdot \mid \psi^{-1}|_{\mathcal{H}\setminus\{i,j\}}, \psi^{-1}(j)), P_{n,q}^{\mathsf{ptn}}(\psi^{-1}(i) = \cdot \mid \psi^{-1}|_{\mathcal{H}\setminus\{i,j\}}, \tilde{\psi}^{-1}(j)))
$$

$$
\leq \max_{\substack{\psi^{-1}(k) \subset [q],\ k \in \mathcal{H}\setminus\{i,j\} \\ \psi^{-1}(j), \tilde{\psi}^{-1}(j) \subset [q]}} P_{n,q}^{\mathsf{ptn}}(\psi^{-1}(i) \cap \tilde{\psi}^{-1}(j) \neq \emptyset \mid \psi^{-1}|_{\mathcal{H}\setminus\{i,j\}}, \psi^{-1}(j)) \tag{16}
$$

$$
\leq 1 - \left(1 - \frac{\rho}{q - (H_0 - 1)\rho}\right) \cdots \left(1 - \frac{\rho}{q - H_0\rho + 1}\right)
$$

$$
\leq 1 - (1 - 2\rho/q)^\rho \leq 2\rho^2/q \leq 1/(2H_0), \tag{17}
$$

where we use $\psi^{-1}|_{\mathcal{H}\setminus\{i,j\}}$ to denote the collection of tuples $(k, \psi^{-1}(k))$ for $k \in \mathcal{H}\setminus\{i,j\}$. In (16), the probability is over $\psi^{-1}(i)$, whose conditional distribution is that of a uniformly random subset of $[q]\setminus(\psi^{-1}(j) \cup (\psi^{-1}(\mathcal{H}\setminus\{i,j\})))$ of size $\rho$. The second inequality above uses Lemma D.9, and the second-to-last inequality uses the fact that $q - H_0\rho + 1 > q - H_0\rho \geq q/2$, since $2H_0\rho \leq q$ by assumption, and the final inequality uses that $q \geq 4\rho^2 H_0$, by assumption.

The above chain of inequalities (17) guarantees that $I_{j \to i}(\psi^{-1}|_{\mathcal{H}}) \leq 1/(2H_0)$ for all $i, j \in \mathcal{H}$ with $i \neq j$. By Theorem F.3, it follows that for any $\delta \in (0, 1)$,

$$
\Pr_{\psi \sim P_{n,q}^{\mathsf{ptn}}}\left(|F_{\mathcal{H}}(\psi) - \mathbb{E}_{P_{n,q}^{\mathsf{ptn}}}[F_{\mathcal{H}}(\psi)]| \geq \sqrt{4H_0 \ln(2/\delta)}\right) \leq \delta. \tag{18}
$$

Write $H := \frac{C_1 \ln^2 n}{\epsilon^2}$, for a sufficiently large constant $C_1$ to be specified below. Note that $4\rho(H+1) \leq n$ as long as $n \geq \frac{8C_1\rho \ln^2 n}{\epsilon^2}$, which in turn, since $\rho \leq n^{1/4}$, holds when $n \geq C/\epsilon^3$ for a sufficiently large constant $C$ (chosen as a function of $C_1$). Let $\mathcal{H}_1, \ldots, \mathcal{H}_{\lfloor n/H \rfloor}$ denote a partition of $[n]$ for which $|\mathcal{H}_j| \in \{H, H+1\}$. Since $4\rho(H+1) \leq n$, we may apply (18) to each $\mathcal{H}_j$ and use a union bound, which yields

$$
\Pr_{\psi \sim P_{n,q}^{\mathsf{ptn}}}\left(|F(\psi) - \mathbb{E}_{P_{n,q}^{\mathsf{ptn}}}[F(\psi)]| \geq \frac{n\sqrt{4\ln(2/\delta)}}{\sqrt{H}}\right) \leq \frac{n\delta}{H} \leq n\delta. \tag{19}
$$

Choosing $\delta = 2\exp(-\ln^2(n)) \leq \mathsf{negl}(n)$ gives that $n\sqrt{4\ln(2/\delta)}/\sqrt{H} \leq 2n\ln(n)/\sqrt{H} \leq \epsilon n$, as long as the constant $C_1$ is chosen sufficiently large. Combining Eqs. (15) and (19) yields that with probability $1 - \mathsf{negl}(n)$ over the draw of $\psi \sim P_{n,q}^{\mathsf{ptn}}$, $F(\psi) \leq n \cdot \left(\frac{1}{2} - \frac{p}{5} + 23\epsilon\right)$, which yields the claimed bound. □

---

[10]Note that, though our assumptions on $Z_1, Z_2$ in the lemma statement are not symmetric, to derive (14) we only need the inequalities $Z_2/\rho \leq \epsilon n$ and $Z_1/n \leq 2$, which hold when the roles of $Z_1, Z_2$ are flipped.

## D.2 Proof of robustness

We are ready to show robustness of our PRC $\mathsf{PRC}_{\mathsf{Idx}}[\mathsf{PRC}_{\mathsf{Sub}}]$ to edit-bounded channels (Definition 2.4).

**Lemma D.8** (Robustness). *There are some constants $C_{D.8}, C_{\mathsf{rob}} \geq 1$ so that the following holds. Consider any $\rho > 1$ and security parameter $\lambda \in \mathbb{N}$ satisfying $n(\lambda) \geq C_{D.8}\rho^4$, and $p_0 \in (C_{D.8}/\rho, 1/(10C_{\mathsf{rob}}))$ and any $(1 - C_{\mathsf{rob}}p_0)$-edit-bounded channel $\mathcal{E}$. Then for any PRC $\mathsf{PRC}_{\mathsf{Sub}}$ which is robust to $(1/2 - p_0)$-substitution-bounded channels, the PRC $\mathsf{PRC}_{\mathsf{Idx}}[\mathsf{PRC}_{\mathsf{Sub}}, \rho]$ in Algorithm 1 is robust to $\mathcal{E}$.*

*Proof of Lemma D.8.* Fix $\lambda \in \mathbb{N}$, and write $n := n(\lambda), q := q(\lambda), m := m(\lambda)$, so that $p_0 > C_{D.8}/\rho \geq 1/\rho$. Fix any $z \in [q]^m$ which is typical (per Definition D.1), and consider any $z' \in [q]^m$ which can be obtained form $z$ via a total of at most $(1 - C_{\mathsf{rob}} \cdot p_0) \cdot m$ substitutions, insertions, and deletions (which has probability 1 over $z' \sim \mathcal{E}(z)$ since $\mathcal{E}$ is $(1 - C_{\mathsf{rob}}p_0)$-edit-bounded. Write $\mathcal{Z} := \mathsf{Unique}(z) \cap \mathsf{Unique}(z')$, i.e., $\mathcal{Z}$ denotes those entries of $z$ which are preserved in $z'$. Since $z$ is typical and $m/q \leq m/(n\rho) \leq 1/\rho$, we have that

$$|\mathsf{Unique}(z)| \geq q \cdot (1 - \exp(-m/q)) - 2\sqrt{m}\ln m \qquad (20)$$
$$\geq q \cdot (m/q - (m/q)^2) - 2\sqrt{m}\ln m \geq m \cdot (1 - 1/\rho) - 2\sqrt{m}\ln m,$$

where the second inequality uses the fact that $\exp(-x) \leq 1 - x + x^2$ for $x \in [0, 1]$. Since the requirements $n \geq C_{D.8}\rho^4$ and $m = \lceil \ln(2) \cdot n \rceil$ ensure that that $2\sqrt{m}\ln(m) \leq m/\rho$ (as long as $C_{D.8}$ is large enough), we have that $|\mathsf{Unique}(z)| \geq m \cdot (1 - 2/\rho)$. Note also that it is immediate that $|\mathsf{Unique}(z)| \leq m$.

**Step 1: Using Lemma D.7.** Since $z'$ is obtained from $z$ via at most $1 - C_{\mathsf{rob}}p_0$ insertions, deletions, and substitutions, we have $|\mathcal{Z}| \geq |\mathsf{Unique}(z)| - m \cdot (1 - C_{\mathsf{rob}}p_0) \geq m \cdot (C_{\mathsf{rob}}p_0 - 2/\rho)$. We also have that $|\mathsf{Unique}(z')| \leq 2m$. We now apply Lemma D.7 with

$$\mathcal{Z}_1 = \mathsf{Unique}(z), \quad \mathcal{Z}_2 = \mathsf{Unique}(z'), \quad \epsilon = p_0, \quad p = \frac{C_{\mathsf{rob}} - 2}{2} \cdot p_0.$$

We must verify that the preconditions (11) are satisfied: first, we check that, as long as $C_{D.8} \geq 8$,

$$\frac{8}{\epsilon} = \frac{8}{p_0} < \frac{8\rho}{C_{D.8}} \leq \rho \leq n^{1/4},$$

where the final inequality follows from our choice of $n \geq \rho^4$. Next, we have

$$|\mathsf{Unique}(z)| \in [m \cdot (1 - 2/\rho), m] \subset [(\ln(2) - \epsilon) \cdot n, (\ln(2) + \epsilon) \cdot n],$$

where we have used that $m = \lceil \ln(2) \cdot n \rceil$ and the fact that $2/\rho < p_0 = \epsilon$ and $n\epsilon = np_0 \geq 1$. Similarly, we have

$$|\mathsf{Unique}(z')| \leq 2m \leq (2\ln(2) + \epsilon) \cdot n.$$

Next, we have $|\mathcal{Z}| \geq m \cdot (C_{\mathsf{rob}}p_0 - 2/\rho) \geq (C_{\mathsf{rob}} - 2)p_0 \cdot m \geq \frac{C_{\mathsf{rob}} - 2}{2} \cdot p_0 n = pn$, since $m \geq n/2$. Finally, we have that

$$n \geq \rho^4 \geq (C_{D.8}/p_0)^4 = (3C_{D.8}/\epsilon)^4 \geq C_{D.7}/\epsilon^3, \qquad (21)$$

as long as $C_{D.8}$ is chosen sufficiently large. Thus, all constraints of Lemma D.7 are satisfied. Given $\psi : [q] \to [n]$ and $z \in [q]^\star$, define $D_\psi(z) \in \{0, 1\}^n$ to be the vector defined by

$$D_\psi(z)_i := \mathbb{1}\{i \in \mathsf{Unique}(\psi(z))\}.$$

Then Lemma D.7 gives that

$$\Pr_{\psi \sim P_{n,q}^{\mathsf{ptn}}} \left( D_{\mathsf{Ham}}(D_\psi(z), D_\psi(z')) \geq n \cdot \left( \frac{1}{2} - \frac{p}{5} + 23\epsilon \right) \right)$$
$$= \Pr_{\psi \sim P_{n,q}^{\mathsf{ptn}}} \left( |\Delta(\psi(\mathsf{Unique}(z)), \psi(\mathsf{Unique}(z')))| \geq n \cdot \left( \frac{1}{2} - \frac{p}{5} + 23\epsilon \right) \right) \leq \mathsf{negl}(n). \qquad (22)$$

As long as $C_{\mathsf{rob}} \geq 300$, we have

$$\frac{1}{2} - \frac{p}{5} + 23\epsilon \leq \frac{1}{2} - \frac{C_{\mathsf{rob}} - 2}{10} \cdot p_0 + 23p_0 \leq \frac{1}{2} - 5p_0.$$

**Step 2: Averaging over $z$.** Let us consider the following "idealized" variant of $\mathsf{Encode}(1^\lambda, (\mathsf{sk}, \psi), \mathsf{m})$, which we denote by $\mathsf{Encode}'(1^\lambda, \psi)$ (as the output of the below procedure does not depend on $\mathsf{sk}$ or $\mathsf{m}$):

1. Sample $y \sim \mathrm{Unif}([n]^m)$.

2. For each $j \in [m]$, choose $z_j \sim \mathrm{Unif}(\{z \,:\, \psi(z) = y_j\})$, so that $z_j \in [q]$.

3. **return** $z = (z_1, \ldots, z_m)$.

For any fixed $z \in [q]^m$, note that $\Pr(\mathsf{Encode}'(1^\lambda, \psi) = z) = q^{-m}$, and in particular does not depend on $\psi$: this holds since each element $z_j$ of $z$ is drawn independently from the distribution which first draws $y_j \sim \mathrm{Unif}([n])$ and then draws $z_j \sim \mathrm{Unif}(\{z \,:\, \psi(z) = y_j\})$; since $|\psi^{-1}(a)| = q/n$ for each $a \in [n]$, this distribution is simply $\mathrm{Unif}([q])$. Let $Q$ denote the joint distribution of $(y, \psi, z, z')$, where $y \sim \mathrm{Unif}([n]^m)$, $\psi \sim P_{n,q}^{\mathsf{ptn}}$, $z$ is generated from $y, \psi$ as in the above procedure $\mathsf{Encode}'(1^\lambda, \psi)$, and $z' \sim \mathcal{E}(z)$. For any $z_0 \in [q]^m$, it follows that the conditional distribution (under $Q$) of $\psi$ given $z = z_0$ is $P_{n,q}^{\mathsf{ptn}}$. Since $\psi$ and $z'$ are conditionally independent given $z$, we have furthermore that for any $z_0, z_0' \in [q]^m$, the conditional distribution (under $Q$) of $\psi$ given $z = z_0, z' = z_0'$ is $P_{n,q}^{\mathsf{ptn}}$. Thus, by (22), if $z_0$ is typical and $z_0'$ can be obtained from $z_0$ with at most $(1 - C_{\mathsf{rob}} p_0) \cdot m$ subsitutions, insertions, and deletions, then

$$\Pr_Q \left( D_{\mathsf{Ham}}(D_\psi(z), D_\psi(z')) \geq n \cdot \left( \frac{1}{2} - 5p_0 \right) \mid z = z_0, z' = z_0' \right) \leq \mathsf{negl}(n).$$

Since $z$ is typical with probability $1 - \mathsf{negl}(n)$ under $Q$ (by Lemma D.6) and and $z'$ can be obtained from $z$ using at most $(1 - C_{\mathsf{rob}} p_0) \cdot m$ substitutions, insertions, and deletions with probability $1$ under $Q$, it follows that in fact

$$\Pr_Q \left( D_{\mathsf{Ham}}(D_\psi(z), D_\psi(z')) \geq n \cdot \left( \frac{1}{2} - 5p_0 \right) \right) \leq \mathsf{negl}(n) \leq \mathsf{negl}(m). \tag{23}$$

**Step 3: using pseudorandomness.** Let $\tilde{Q}$ denote the joint distribution of $(y, \psi, z, z')$ where $y$ is distributed as the random variable $y$ defined in Line 7 of Algorithm 1, $\psi \sim P_{n,q}^{\mathsf{ptn}}$, $z$ is distributed as the random variable $z$ defined in Line 8 of Algorithm 1 given the value of $\psi$ and $y$ (i.e., $z_j \sim \mathrm{Unif}(\{a \,:\, \psi(a) = y_j\})$), and $z' \sim \mathcal{E}(z)$. Using the fact a sample from $\mathcal{E}(\cdot)$ may be produced by a probabilistic polynomial-time algorithm together with Lemmas D.2 and D.3, we have that

$$\left| \Pr_Q \left( D_{\mathsf{Ham}}(D_\psi(z), D_\psi(z')) \geq n \cdot \left( \frac{1}{2} - 5p_0 \right) \right) - \Pr_{\tilde{Q}} \left( D_{\mathsf{Ham}}(D_\psi(z), D_\psi(z')) \geq n \cdot \left( \frac{1}{2} - 5p_0 \right) \right) \right| \leq \mathsf{negl}(n). \tag{24}$$

In more detail, to arrive at (24), we reason as follows: if the difference in (24) were non-negligible, then we could distinguish in polynomial time between a sample $y^0 \leftarrow \mathsf{Encode}_{\mathsf{Sub}}(1^\lambda, \mathsf{sk}, \mathsf{m})$ from a uniformly random string $y^0 \sim \mathrm{Unif}(\{0,1\}^n)$, as follows: we first generate $y \leftarrow \mathsf{PerturbDifference}(n, m, y^0)$ (as in Line 7 of Algorithm 1, then sample $\psi \sim P_{n,q}^{\mathsf{ptn}}$, then sample $z \in [q]^m$ by $z_j \sim \mathrm{Unif}(\{a \,:\, \psi(a) = y_j\})$ for $j \in [m]$, then sample $z' \sim \mathcal{E}(z)$, and finally evaluate $D_{\mathsf{Ham}}(D_\psi(z), D_\psi(z'))$. In the event that $y^0 \leftarrow \mathsf{Encode}_{\mathsf{Sub}}(1^\lambda, \mathsf{sk}, \mathsf{m})$, then the resulting distribution of $(y, \psi, z, z')$ is $\tilde{Q}$, and in the event that $y^0 \sim \mathrm{Unif}(\{0,1\}^n)$, Lemma D.2 gives that the induced distribution over $y$ is $\mathrm{Unif}([n]^m)$ and thus the resulting distribution of $(y, \psi, z, z')$ is $Q$. Thus we would get a contradiction to Lemma D.3.

**Step 4: Wrapping up.** We have from Eqs. (23) and (24) that

$$\Pr_{\tilde{Q}} \left( D_{\mathsf{Ham}}(D_\psi(z), D_\psi(z')) \geq n \cdot \left( \frac{1}{2} - 3p_0 \right) \right) \leq \mathsf{negl}(m). \tag{25}$$

By our construction of the distribution $\tilde{Q}$, $D_\psi(z) = y$ with probability 1 under $\tilde{Q}$. Moreover, we have that

$$\Pr_{(\mathsf{sk},\psi)\leftarrow\mathsf{KeyGen}(1^\lambda)} \big(\mathsf{Decode}(1^\lambda, (\mathsf{sk}, \psi), \mathcal{E}(z)) \neq \mathsf{m} \mid z \leftarrow \mathsf{Encode}(1^\lambda, (\mathsf{sk}, \psi), \mathsf{m})\big)$$

$$= \Pr_{(\mathsf{sk},\psi)\leftarrow\mathsf{KeyGen}(1^\lambda)} \big(\mathsf{Decode}_{\mathsf{Sub}}(1^\lambda, \mathsf{sk}, D_\psi(\mathcal{E}(z))) \neq \mathsf{m} \mid z \leftarrow \mathsf{Encode}(1^\lambda, (\mathsf{sk}, \psi), \mathsf{m})\big)$$

$$= \Pr_{\mathsf{sk}\sim\mathsf{KeyGen}_{\mathsf{Sub}}(1^\lambda),\psi\sim P_{n,q}^{\mathsf{ptn}}} \big(\mathsf{Decode}_{\mathsf{Sub}}(1^\lambda, \mathsf{sk}, \bar{\mathcal{E}}_\psi(y^0)) \neq \mathsf{m} \mid y^0 \leftarrow \mathsf{Encode}_{\mathsf{Sub}}(1^\lambda, \mathsf{sk}, \mathsf{m})\big) \leq \mathsf{negl}(\lambda),$$

$$(26)$$

where $\bar{\mathcal{E}}_\psi : \{0,1\}^n \to \{0,1\}^n$ denotes the (random) channel which, given $y^0 \in \{0,1\}^n$, first applies the procedure in Lines 7 and 8 of Algorithm 1 to generate $z \in [q]^m$ from $y^0$, and then outputs $D_\psi(z')$ for $z' \sim \mathcal{E}(z)$. For future reference we let this distribution over $(y^0, z)$ where $\mathsf{sk} \sim \mathsf{KeyGen}_{\mathsf{Sub}}(1^\lambda), y^0 \leftarrow \mathsf{Encode}_{\mathsf{Sub}}(1^\lambda, \mathsf{sk}, \mathsf{m})$ be denoted by $R$. The first equality in the display above uses that the output of $\mathsf{Decode}(1^\lambda, (\mathsf{sk}, \psi), z')$ is given by $\mathsf{Decode}_{\mathsf{Sub}}(1^\lambda, \mathsf{sk}, D_\psi(z'))$. The second equality uses the definition of $\mathsf{Encode}(1^\lambda, (\mathsf{sk}, \psi), \mathsf{m})$ in Algorithm 1. Finally, the inequality follows since the PRC $\mathsf{PRC}_{\mathsf{Sub}}$ is robust to $(1/2 - p_0)$-substitution bounded channels together with Lemma G.1 and the fact that with probability $1 - \mathsf{negl}(\lambda)$ over the draw of $\psi \sim P_{n,q}^{\mathsf{ptn}}, \mathsf{sk} \sim \mathsf{KeyGen}_{\mathsf{Sub}}(1^\lambda)$, $y^0 \leftarrow \mathsf{Encode}_{\mathsf{Sub}}(1^\lambda, \mathsf{sk}, \mathsf{m})$, and $z' \sim \bar{\mathcal{E}}_\psi(y^0)$, we have $D_{\mathsf{Ham}}(z', y^0) \leq (1/2 - p_0) \cdot n$. This fact follows from the following observations:

- By Lemma D.5 and undetectability of $\mathsf{PRC}_{\mathsf{Sub}}$, for any $\psi$, with probability $1 - \mathsf{negl}(n) - \mathsf{negl}(\lambda) \geq 1 - \mathsf{negl}(\lambda)$ over $(y^0, z) \sim R$, we have $D_{\mathsf{Ham}}(y^0, D_\psi(z)) \leq p_0 n$. (In particular, Lemma D.5 ensures that $D_{\mathsf{Ham}}(y^0, D_\psi(z)) \leq p_0 n$ when $y^0 \sim \mathsf{Unif}(\{0,1\})^n$ and then $z$ is generated from $y^0$ as in Lines 7 and 8 of Algorithm 1, and undetectability of $\mathsf{PRC}_{\mathsf{Sub}}$ ensures that this also holds when instead $y^0 \sim \mathsf{Encode}_{\mathsf{Sub}}(1^\lambda, \mathsf{sk}, \mathsf{m})$.) Here we have also used the fact that $\psi(z)$ is the output string $y$ of $\mathsf{PerturbDifference}(n(\lambda), m(\lambda), y^0)$ (defined in Line 7), together with the fact that $n \geq C_{D.5} \ln^2(1/\epsilon)/\epsilon^2$ by (21), as long as the constant $C_{D.8}$ is chosen sufficiently large.

- By (25) together with the fact that the distribution of $(\psi, z, z')$ where $\psi \sim P_{n,q}^{\mathsf{ptn}}, z \sim R$, and $z' \sim \mathcal{E}(z)$ is exactly the marginal distribution of $(\psi, z, z') \sim \tilde{Q}$, we have that $D_{\mathsf{Ham}}(D_\psi(z), D_\psi(z')) \leq \frac{1}{2} - 3p_0$ with probability $1 - \mathsf{negl}(n) \geq 1 - \mathsf{negl}(\lambda)$.

- Combining the two points above, we see that with probability $1 - \mathsf{negl}(\lambda)$ over $\mathsf{sk} \sim \mathsf{KeyGen}_{\mathsf{Sub}}(1^\lambda), y^0 \sim \mathsf{Encode}_{\mathsf{Sub}}(1^\lambda, \mathsf{sk}, \mathsf{m}), \psi \sim P_{n,q}^{\mathsf{ptn}}, z' \sim \bar{\mathcal{E}}_\psi(y^0), D_{\mathsf{Ham}}(y^0, z') \leq (1/2 - p_0) \cdot n$, as desired.

Summarizing, we have established (26), which yields the desired robustness guarantee. $\square$

**Remark D.2** (Removing computational efficiency of channel)**.** Though the proof of Lemma D.8 uses the fact that (per Definition 2.4), the channel $\mathcal{E}$ is sampleable in polynomial time (namely, in Step 3), this assumption is not necessary, in the following sense. If we make the slightly stronger assumpgion that $\mathsf{PRC}_{\mathsf{Sub}}$ is in fact $(1/2 - p_0)$-weakly substitution bounded (per Definition C.2), then the proof of Lemma D.8 can be modified as follows to remove the assumption that $\mathcal{E}$ is computationally efficient: instead of showing that with probability $1 - \mathsf{negl}(\lambda)$ over the draw of $y^0 \leftarrow \mathsf{Encode}_{\mathsf{Sub}}(1^\lambda, \mathsf{sk}, \mathsf{m})$ and $z' \sim \bar{\mathcal{E}}_\psi(y^0)$, we have $D_{\mathsf{Ham}}(z', y^0) \leq (1/2 - p_0) \cdot n$ and using Lemma G.1, we would show that $D_{\mathsf{Ham}}(z', y^0) \leq (1/2 - p_0) \cdot n$ with probability $1 - \mathsf{negl}(\lambda)$ when instead $y^0 \leftarrow \mathsf{Unif}(\{0,1\}^n)$, which would imply that $\bar{\mathcal{E}}_\psi$ is $(1/2 - p_0)$-weakly robust. This latter argument avoids us to avoid needing to reason about the distribution of $y^0 \leftarrow \mathsf{Encode}_{\mathsf{Sub}}(1^\lambda, \mathsf{sk}, \mathsf{m})$, and so allows us to omit Step 3 in the proof of Lemma D.8. Moreover, we note that (as discussed in Appendix C.2) Theorem 3.2 in fact establishes the existence of PRCs robust to weakly substitution bounded channels (as do [CG24, Sections 5.3 & 5.4]).

**Lemma D.9.** *Fix integers $N, \rho \in \mathbb{N}$ and a subset $\mathcal{Z} \subset [N]$ with $Z := |\mathcal{Z}|$, and suppose $Z \leq N - \rho$. Let $\mathcal{T} \subset [N]$ be a uniformly random subset of size $\rho$. Then*

$$\Pr(\mathcal{Z} \cap \mathcal{T} = \emptyset) = \left(1 - \frac{Z}{N}\right) \cdots \left(1 - \frac{Z}{N - \rho + 1}\right) \in \left[\exp\left(-\frac{\rho Z}{N - \rho} - \frac{\rho Z^2}{(N - \rho)^2}\right), \exp\left(-\frac{\rho Z}{N}\right)\right].$$

*Proof.* We may choose $\mathcal{T}$ by selecting its elements without replacement from $[N]$. After $i$ elements of $\mathcal{T}$ (not intersecting $\mathcal{Z}$) have been chosen, the $i + 1$th element of $\mathcal{T}$ is distributed uniformly over a set of size $N - i$, of which $Z$ elements belong to $\mathcal{Z}$. This establishes the equality. To see the following containment, we use the fact that $\exp(-x) \geq 1 - x \geq \exp(-x - x^2)$ for all $x \in [0, 1]$. $\qquad\square$

## E  Watermarks from PRCs over larger alphabets

### E.1  Overview of the algorithm and guarantee

In this section, we discuss how to use any PRC with security parameter $\lambda$ and alphabet size $\mathrm{poly}(\lambda)$ to produce a watermarking scheme meeting the requirements of Definition B.2. The reduction of [CG24] for this task is limited to the case where the PRC has a *binary* alphabet. To extend to the setting where the alphabet size of the PRC is larger, some new ideas are needed.

Suppose that we are given a zero-bit pseudorandom code, PRC, with block length $n(\lambda)$ over an alphabet $\Sigma_{\mathsf{PRC}}(\lambda)$, with $|\Sigma_{\mathsf{PRC}}(\lambda)| \geq n(\lambda)$, which is robust to a constant fraction of substitutions, insertions, and deletions (such a code is provided by Theorem 4.1). Given some constant $\alpha$ and a family of language models $(\mathsf{Model}(\lambda))_{\lambda \in \mathbb{N}}$ over alphabet $(\Sigma(\lambda))_{\lambda \in \mathbb{N}}$ with $|\Sigma(\lambda)| \geq |\Sigma_{\mathsf{PRC}}(\lambda)|$, we construct a watermarking scheme $\mathcal{W}[\mathsf{PRC}, \mathsf{Model}]$ (Algorithm 3), as the following tuple of algorithms $(\mathsf{Setup}, \mathsf{Wat}, \mathsf{Detect})$:

- $\mathsf{Setup}(1^\lambda)$ produces a pair $\mathsf{sk}_{\mathsf{Wat}} = (\mathsf{sk}, \phi)$ as the secret key of $\mathcal{W}$. Here $\mathsf{sk}$ is a secret key generated from $\mathsf{PRC.KeyGen}(\lambda)$, and $\phi : \Sigma(\lambda) \to \Sigma_{\mathsf{PRC}}(\lambda)$ is chosen uniformly at random, which can be interpreted as a hash function.

- $\mathsf{Wat}(1^\lambda, (\mathsf{sk}, \phi))$ generates a sequence of tokens $\mathsf{t}_1, \mathsf{t}_2, \ldots$ in blocks of length $n(\lambda)$, in the following manner: for each block of length $n(\lambda)$, we sample a codeword $x \leftarrow \mathsf{PRC.Encode}(1^\lambda, \mathsf{sk})$ (Line 13), and at each position $i$, we consider the pushforward distribution $\bar{p}_i := \phi \circ \mathsf{Model}(\mathsf{t}_i = \cdot \mid \mathsf{t}_{1:i-1}) \in \Delta(\Sigma_{\mathsf{PRC}}(\lambda))$ of the next token under the hash function $\phi$ (Line 15 of the subroutine $\mathsf{EmbedChar}$). If $\bar{p}_i$ puts enough mass on the corresponding token of the codeword $x$ (denoted by $x_j$ in Algorithm 3), then we let the output token of $\mathsf{Wat}$ at position $i$ be a uniformly random token which hashes to $x_j$ (Lines 16 and 21). Otherwise, we sample the output token of $\mathsf{Wat}$ at position $i$ in a way that will ensure it does not alter the conditional distribution of the $i$th token (Lines 19 and 21).

  To obtain some intuition for the above procedure, it is straightforward to see that if the codewords $x$ are actually uniformly random, then the output of this procedure is identical to the distribution $\overline{\mathsf{Model}}$ (Lemma E.4). Thus, if $x$ is drawn from a distribution computationally indistinguishable from random (as will be the case for the output of a PRC), it is simple to show that this procedure yields an undetectable watermark.

- $\mathsf{Detect}(1^\lambda, (\mathsf{sk}, \phi), \mathsf{t})$ functions as follows, given a sequence $\mathsf{t} = \mathsf{t}_{1:\ell} \in \Sigma(\lambda)^\ell$ of (potentially watermarked) text. It searches through all contiguous substrings of $\mathsf{t}$, denoted $\mathsf{t}_{i:j}$ and checks whether $(\phi(\mathsf{t}_i), \ldots, \phi(\mathsf{t}_j))$ decodes to $\emptyset$ under $\mathsf{PRC.Decode}$. If so, it returns $\mathsf{True}$ (and otherwise $\mathsf{False}$).

In the below theorem, we suppose that PRC is a zero-bit PRC with block length $n(\lambda)$ over alphabet $\Sigma_{\mathsf{PRC}}(\lambda)$, satisfying $|\Sigma_{\mathsf{PRC}}(\lambda)| \geq n(\lambda)$. Furthermore, we suppose that $(\mathsf{Model}(\lambda))_{\lambda \in \mathbb{N}}$ is a family of language models defined over alphabet $\Sigma(\lambda)$.

**Theorem E.1** (Watermarking from PRCs). *Suppose that $p, \alpha \in (0, 1)$ are given, that $\mathsf{PRC}, \mathsf{Model}(\lambda)$ are as described above and satisfy $|\Sigma(\lambda)| \geq (\frac{8}{\alpha} |\Sigma_{\mathsf{PRC}}(\lambda)|)^{2/\alpha}$, and that $\mathsf{PRC}$ satisfies robustness to any $(1 - \frac{\alpha}{8} + \frac{3p}{\alpha})$-edit-bounded channel. Then the watermarking scheme $\mathcal{W}[\mathsf{PRC}, \mathsf{Model}]$ (Algorithm 3) is sound, undetectable, and $\beta_\lambda(\ell)$-substring robust to any $p$-edit-bounded channel, where $\beta_\lambda(\ell) := 8n(\lambda) + 6\alpha\ell$.*

Theorem E.1 shows that for any constant $\alpha$, we can detect the watermark as long as the entropy rate of the model's text is at least $\Omega(\alpha)$ and as long as the fraction of errors (adversarial deletions/insertions) introduced to the watermarked text is at most $O(\alpha^2)$. This latter statement follows since there are PRCs robust to $(1 - \frac{\alpha}{8} + \frac{3p}{\alpha})$-edit-bounded channels for $p = O(\alpha^2)$. In turn, the alphabet $\Sigma(\lambda)$ of the language model needs to have size roughly $|\Sigma_{\mathsf{PRC}}(\lambda)|^{2/\alpha}$, which is a polynomial in $|\Sigma_{\mathsf{PRC}}(\lambda)|$ as long as $\alpha$ is a constant.

By combining Theorems 3.2, 4.1 and E.1 we can show our main theorem, which guarantees substring-robust watermarking schemes for edit-bounded channels under the existence of local weak PRFs (Assumption 3.1). To simplify notation, given constants $\alpha, q > 0$ together with a function family $\mathcal{F}$ consisting of binary-valued functions on $\{0,1\}^{n(\lambda)}$, let us define

$$\mathcal{W}^{\mathsf{comp}}[\mathcal{F}, q, \alpha] := \mathcal{W}\left[\mathsf{PRC_{Idx}}\left[\mathsf{PRF\text{-}PRC}\left[\mathcal{F}, \frac{1}{2} - \frac{\alpha}{16 C_{\mathsf{rob}}}, q\right], \frac{16 C_{\mathsf{rob}} C_0}{\alpha}\right], \mathsf{Model}\right], \quad (27)$$

where $C_0, C_{\mathsf{rob}} \geq 1$ are the constants from Theorem 4.1. In words, $\mathcal{W}^{\mathsf{comp}}[\mathcal{F}, q, \alpha]$ chains together the PRCs in Algorithms 1 and 2 and the watermarking scheme in Algorithm 3.

**Theorem E.2** (Main theorem). *There are absolute constants $c, C_1, C_2 > 0$ so that the following holds. Fix any $\alpha > 0$. Suppose there exists a function family $\mathcal{F}$, consisting of binary-valued functions on $\{0,1\}^{n(\lambda)}$, which is a $\log n(\lambda)$-local weak PRF for some noise level $q \in [0, 1/2)$, per Assumption 3.1. Then for any language model family $\mathsf{Model}(\lambda)$ over alphabets $\Sigma(\lambda)$, the watermarking scheme $\mathcal{W}^{\mathsf{comp}}[\mathcal{F}, q, \alpha]$ ((27)) is sound, undetectable, and $\beta_\lambda(\ell)$-substring robust to any $c\alpha^2$-edit-bounded channel, as long as:*

$$\beta_\lambda(\ell) := 6\alpha\ell + 8 \cdot n(\lambda)^{C_1 \log \frac{1}{\alpha}}, \qquad |\Sigma(\lambda)| \geq n(\lambda)^{C_2 \frac{1}{\alpha} \log \frac{1}{\alpha}}.$$

Notice that the exponent of $n(\lambda)$ in the $\ell$-independent term of $\beta_\lambda(\ell)$ depends on $\alpha$, as does the exponent of $n(\lambda)$ in the size of $\Sigma(\lambda)$. While $\alpha$ is a constant (and so this only leads to a polynomial blowup), it would be desirable to come up with more efficient reductions. We remark, however, that this issue is quite subtle, since our criterion of undetectability (in Definitions 2.1 and B.2) is phrased with respect to any *polynomial-time algorithm*, meaning that given a watermarking scheme $\mathcal{W} = (\mathsf{Setup}, \mathsf{Wat}, \mathsf{Detect})$ we can construct a polynomially-more efficient watermarking scheme $\mathcal{W}' = (\mathsf{Setup}', \mathsf{Wat}', \mathsf{Detect}')$ as follows. The algorithms $\mathsf{Setup}', \mathsf{Wat}', \mathsf{Detect}'$, given a security parameter $\lambda$, call the corresponding $\mathsf{Setup}, \mathsf{Wat}, \mathsf{Detect}$ algorithms with security parameter $\lambda^\alpha$, for some $\alpha < 1$. Since any function which is $\mathsf{negl}(\lambda^\alpha)$ is also $\mathsf{negl}(\lambda)$ and an algorithm running in time $\mathrm{poly}(\lambda)$ is also $\mathrm{poly}(\lambda^\alpha)$, the new scheme $\mathcal{W}'$ is sound, undetectable, and substring robust if $\mathcal{W}$ is. Moreover, its various parameters (e.g., $|\Sigma(\lambda)|$) will typically be smaller than those of $\mathcal{W}$ by a polynomial. Of course, the guarantee afforded by undetectability of $\mathcal{W}'$ (due to its use of $\lambda^\alpha$ as opposed to $\lambda$) is weaker than that of $\mathcal{W}$, though only by a polynomial.

Indeed, the approach of [CG24] implicity suffers from the same $n(\lambda)^{O(\log 1/\alpha)}$ dependence in their substring-robustness function $\beta(\ell)$, though it is not explicitly stated as such since they essentially already scale down the security parameter. In particular, the sparsity parameter $t$ in [CG24, Lemmas 4 & 5] is required to be $O\left(\frac{\log n}{\log \frac{1}{\frac{1}{2} - p}}\right)$ (where $p < 1/2$ denotes the rate of substitutions for the purpose of evaluating robustness; one can think of $\frac{1}{2} - p$ as being roughly comparable to $\alpha$ in Theorem E.2). Moreover, there is an algorithm which can distinguish the PRCs in [CG24] from uniform strings which runs in time roughly $n^t$. To formally reason about polynomial-sized differences in the parameters, we would need to make fine-grained average case hardness assumptions, a direction we do not pursue in the present work.

### E.2 Proof overview for Theorem E.1

Fix a security parameter $\lambda \in \mathbb{N}$; we will drop the argument $\lambda$ in our notation for the proof overview. In the algorithm description above we discussed the idea behind undetectability of $\mathcal{W}$, and the proof of soundness is immediate. Therefore we focus on the (substring) robustness claim. The high-level idea of the proof of robustness is to show that the procedure $\mathsf{Wat}$ can be viewed as a substitution channel which (repeatedly) takes as input an output $x$ of $\mathsf{PRC.Encode}$ and changes some of the tokens to produce its output $\mathsf{t}$. (Technically, we view it as a channel over $\Sigma_{\mathsf{PRC}}$ by applying the hash function $\phi$ to the output $\mathsf{t}$ of $\mathsf{Wat}$.) Our goal is to show that for a codeword $x \in \Sigma_{\mathsf{PRC}}^n$, $\mathsf{Wat}$ makes at most $(1 - \Omega(\alpha)) \cdot n$ substitutions if the empirical entropy of the generated text derived from $x$ (i.e., the output of the channel) is at least $\Omega(\alpha \cdot n \cdot \log |\Sigma|)$. In such a case, we say that the "empirical entropy rate" of the generated text is $\Omega(\alpha)$.

The key technical difference between our watermarking procedure and that of [CG24] is the introduction of the hash function $\phi$. Let us first see what goes wrong without such a $\phi$, i.e., if $\Sigma = \Sigma_{\mathsf{PRC}}$ and $\phi$ is the identity map. Suppose that for each $i$, $\mathsf{Model}(\mathsf{t}_i = \cdot \mid \mathsf{t}_{1:i-1})$ is uniform over a fixed subset

$\Sigma' \subset \Sigma$ of size $|\Sigma'| = |\Sigma|^{\alpha}$. Then with high probability the output of $\overline{\mathsf{Model}}$ will have empirical entropy rate $\Omega(\alpha)$. However, a codeword $x$ produced by PRC will only have the property that a roughly $|\Sigma|^{\alpha-1}$ fraction of its tokens belong to $\Sigma'$. Thus, the procedure in Algorithm 3 will only be able to maintain roughly a $|\Sigma|^{\alpha-1}$ fraction of tokens of $x$ (i.e., the **if** statement on Line 16 will only succeed with probability $|\Sigma|^{\alpha-1}$), and thus the error rate of the corresponding substitution channel will be roughly $1 - |\Sigma|^{\alpha-1} = 1 - o(1)$, since $|\Sigma| \geq n$. This error rate is too large, since our pseudorandom codes (even over large alphabets) can only correct a $1 - p$ fraction of errors, for any *constant $p > 0$*.

To circumvent this issue, we make the following observation: if the entropy of the distribution $p_i := \mathsf{Model}(\mathsf{t}_i = \cdot \mid \mathsf{t}_{1:i-1}) \in \Delta(\Sigma)$ is at least $\alpha \cdot \ln|\Sigma|$, then, if $\Sigma$ is sufficiently large compared to $\Sigma_{\mathsf{PRC}}$ (roughly $|\Sigma| \geq |\Sigma_{\mathsf{PRC}}|^{\Omega(1/\alpha)}$), then for a uniformly random hash function $\phi : \Sigma \to \Sigma_{\mathsf{PRC}}$, the pushforward distribution $\bar{p}_i = \phi \circ p_i \in \Delta(\Sigma_{\mathsf{PRC}})$ will be "close to uniform" in the folloing sense: At least an $\Omega(\alpha)$ fraction of the mass of $\bar{p}_i$ will be on tokens $\sigma \in \Sigma_{\mathsf{PRC}}$ for which $\bar{p}_i(\sigma) \leq 1/|\Sigma_{\mathsf{PRC}}|$. This claim is proven formally in Lemma E.3, and may be seen as a consequence of concentration of measure. Since each such token $\sigma$ occurs with probability roughly $1/|\Sigma|$ under a codeword output by the PRC (by undetectability), altogether such tokens account for a probability $\Omega(\alpha)$ event under which the condition on Line 16 of Algorithm 3 will evaluate to true. Thus, at least a fraction $\Omega(\alpha)$ of the tokens will *not* be substituted, as desired. This idea (together with the application of various concentration inequalities) allows us to ensure that the substitution channel induced by $\mathsf{Wat}$ introduces a fraction $1 - \Omega(\alpha)$ of errors.

Next, the $p$-edit bounded channel referred to in the statement of Theorem E.1 introduces an additional $p$ fraction of errors in the sequence of watermarked text. However, because an output of $\mathsf{Wat}$ consists of a sequence of text $\mathsf{t}$ consisting of multiple (say $M$) consecutive codewords of PRC (each of block length $n$), we run into the following issue: suppose that all of the entropy of $\mathsf{t}$ is concentrated in $\alpha \cdot M$ of the codeword blocks. Also suppose that the $p$-edit bounded channel concetrates all of its $p \cdot Mn$ errors in those same $\alpha M$ blocks, meaning that the effective rate of (edit) errors in those $\alpha M$ blocks is in fact $\frac{pMn}{\alpha Mn} = p/\alpha$. Thus, we need the PRC to in fact be robust to $(1 - \Omega(\alpha) + O(p/\alpha))$-edit-bounded channels in order to detect the watermark. It is straightforward to see that the aforementioned scenario is worst possible, meaning that such robustness is in fact sufficient. Full details of the proof may be found in Appendix E.3.

## E.3 Formal algorithm and guarantee

Algorithm 3 displays our watermarking procedure $\mathcal{W}[\mathsf{PRC}, \mathsf{Model}]$, given a pseudorandom code PRC over alphabet $\Sigma_{\mathsf{PRC}}(\lambda)$ and a family of language models $(\mathsf{Model}(\lambda))_{\lambda \in \mathbb{N}}$ over alphabet $\Sigma(\lambda)$. Whenever the security parameter $\lambda$ is clear from context, we drop the argument $\lambda$, i.e., write $\Sigma_{\mathsf{PRC}}, \Sigma, \mathsf{Model}$. To aid in the analysis, given a language model $\mathsf{Model}$ over alphabet $\Sigma$, an alphabet $\Sigma_{\mathsf{PRC}}$ for the PRC, and a mapping $\phi : \Sigma \to \Sigma_{\mathsf{PRC}}$, we define an *embedding channel* $x \mapsto \mathcal{E}^{\phi}_{\mathsf{Emb}}(x; \mathsf{t}_{1:i-1})$ for each choice of $i \in \mathbb{N}$ and $\mathsf{t}_{1:i-1} \in \Sigma^{i-1}$, which maps $x \in \Sigma^n_{\mathsf{PRC}}$ to some (random) string $\mathcal{E}^{\phi}_{\mathsf{Emb}}(x; \mathsf{t}_{1:i-1}) \in \Sigma^n$ (as the input and output alphabets are different, our use of the term "channel" is a slight abuse of terminology). Given $x \in \Sigma^n_{\mathsf{PRC}}$ and $\mathsf{t}_{1:i-1} \in \Sigma^{i-1}$, $\mathcal{E}^{\phi}_{\mathsf{Emb}}(x; \mathsf{t}_{1:i-1})$ performs the following for $1 \leq j \leq n$: for $p_j = \mathsf{Model}(\mathsf{t}_{j+i-1} = \cdot \mid \mathsf{t}_{1:j+i-2})$, it generates $\mathsf{t}_{j+i-1} \leftarrow \mathsf{EmbedChar}(x_j, p_j, \phi)$. (If some token $\mathsf{t}_{j+i-1}$ is the terminal token END, then all remaining tokens are also the terminal token END.) Note that this is exactly the procedure in Lines 9 and 10 for steps $i$ through $i + n - 1$ of $\mathsf{Wat}(1^{\lambda}, (\mathsf{sk}, \phi), \phi)$ of Algorithm 3.

**Lemma E.3.** *Suppose $\Sigma, \Sigma'$ are finite alphabets, and $P \in \Delta(\Sigma)$ is fixed. Let $\phi : \Sigma \to \Sigma'$ be a uniformly random function. Then*

$$\Pr_{\phi}\left(\sum_{\sigma' \in \Sigma'} \min\left\{\frac{1}{|\Sigma'|}, \phi \circ P(\sigma')\right\} \geq \frac{H(P)}{4 \ln|\Sigma|}\right) \geq 1 - 2^{|\Sigma'| - \frac{H(P) \cdot \exp(H(P)/2)}{4 \ln|\Sigma|}}.$$

*Proof.* Define $\eta := \frac{1}{\exp(H(P)/2)}$. Set $\mathcal{T} := \{\sigma \in \Sigma : P(\sigma) \leq \eta\}$. Note that $H(P) \leq P(\mathcal{T}) \cdot \ln|\Sigma| + P(\Sigma \backslash \mathcal{T}) \cdot H(P)/2$, meaning that $P(\mathcal{T}) \geq \frac{H(P)}{2\ln|\Sigma|}$. Writing $M := |\mathcal{T}|$, we have $M \geq \frac{H(P)}{2\ln|\Sigma| \cdot \eta}$.

**Algorithm 3** Watermarking from PRCs for general alphabets: $\mathcal{W}[\text{PRC}, \text{Model}]$

---

**Require:** Pseudorandom code PRC with security parameter $\lambda$ over alphabet $\Sigma_{\text{PRC}} = \Sigma_{\text{PRC}}(\lambda)$ and block length $n(\lambda)$, Model over alphabet $\Sigma(\lambda)$, maximum length of model text $L_{\max}(\lambda)$.

1: **function** Setup($1^\lambda$)
2:      sk $\leftarrow$ PRC.KeyGen($1^\lambda$).
3:      Let $\phi : \Sigma(\lambda) \to \Sigma_{\text{PRC}}(\lambda)$ be chosen uniformly randomly.
4:      **return** (sk, $\phi$).

5: **function** Wat($1^\lambda$, (sk, $\phi$))
6:      $x \leftarrow$ PRC.Encode($1^\lambda$, sk).            $\triangleright x \in \Sigma_{\text{PRC}}^n$
7:      $i \leftarrow 1, j \leftarrow 1$.
8:      **while** END $\notin \{t_1, \ldots, t_{i-1}\}$ **do**
9:          $p_i \leftarrow$ Model($t_i = \cdot \mid t_1, \ldots, t_{i-1}$).            $\triangleright p_i \in \Delta(\Sigma)$
10:         $t_i \leftarrow$ EmbedChar($x_j, p_i, \phi$).
11:         $i \leftarrow i + 1, j \leftarrow j + 1$.
12:         **if** $j > n(\lambda)$ **then**
13:             $j \leftarrow 1, x \leftarrow$ PRC.Encode($1^\lambda$, sk).

14: **function** EmbedChar($x_j, p_i, \phi$)           $\triangleright x_j \in \Sigma_{\text{PRC}}, p_i \in \Delta(\Sigma), \phi : \Sigma \to \Sigma_{\text{PRC}}$
15:      $\bar{p}_i \leftarrow \phi \circ p_i$.            $\triangleright \bar{p}_i \in \Delta(\Sigma_{\text{PRC}})$
16:      **if** $\text{Ber}(\min\{1, |\Sigma_{\text{PRC}}(\lambda)| \cdot \bar{p}_i(x_j)\}) = 1$ **then**
17:         Set $y_i \leftarrow x_j$.            $\triangleright y_i \in \Sigma_{\text{PRC}}$
18:      **else**
19:         Sample $y_i \sim q_i$, where $q_i(\sigma) \propto \left[\bar{p}_i(\sigma) - \frac{1}{|\Sigma_{\text{PRC}}|}\right]_+$.
20:
21:      Sample $t_i$ from the distribution of $\sigma \sim p_i \mid \phi(\sigma) = y_i$.
22:      **return** $t_i$.

23: **function** Detect($1^\lambda$, (sk, $\phi$), ($t_1, \ldots, t_\ell$))
24:      **for** $i \in [\ell], j \in [i, \min\{i + n - 1, \ell\}]$ **do**
25:         **if** PRC.Decode(sk, ($\phi(t_i), \ldots, \phi(t_j)$)) $\neq \perp$ **then**
26:             **return** True.
27:      **return** False.

---

Next, define $Q \in \Delta(\Sigma)$ by $Q(\sigma) := \frac{\mathbb{1}\{\sigma \in \mathcal{T}\} \cdot P(\sigma)}{P(\mathcal{T})}$, and let $\mathcal{U}$ denote the uniform distribution on $\Sigma'$. For any subset $\mathcal{S} \subset \Sigma'$, by Hoeffding's inequality we have that

$$\Pr_\phi \left(|\phi \circ Q(\mathcal{S}) - \mathcal{U}(\mathcal{S})| \geq \epsilon\right) = \Pr_\phi \left(\left|\mathcal{U}(\mathcal{S}) - \sum_{\sigma \in \Sigma} Q(\sigma) \cdot \mathbb{1}\{\phi(\sigma) \in \mathcal{S}\}\right| \geq \epsilon\right) \leq 2 \exp\left(-\frac{2\epsilon^2}{\sum_{\sigma \in \Sigma} Q(\sigma)^2}\right),$$

where the randomness is over the draw of a uniformly random function $\phi : \Sigma \to \Sigma_{\text{PRC}}$. Using that $\sum_{\sigma \in \Sigma} Q(\sigma)^2 \leq \max_{\sigma \in \Sigma} Q(\sigma) \leq \frac{\eta}{P(\mathcal{T})} \leq \frac{\eta \cdot 2 \ln |\Sigma|}{H(P)}$ together with a union bound over all subsets $\mathcal{S} \subset \Sigma'$,[11] we see that

$$\Pr_\phi \left(d_{\text{TV}}(\mathcal{U}, \phi \circ Q) \geq \epsilon\right) \leq 2^{|\Sigma'|} \cdot \exp\left(\frac{-2\epsilon^2 \cdot H(P)}{\eta \cdot 2 \ln |\Sigma|}\right) \leq 2^{|\Sigma'| - \frac{\epsilon^2 \cdot H(P) \cdot \exp(H(P)/2)}{\ln |\Sigma|}}.$$

Under the event that $d_{\text{TV}}(\mathcal{U}, \phi \circ Q) \leq \epsilon$, we have that $\sum_{\sigma' \in \Sigma'} \min\{1/|\Sigma'|, \phi \circ Q(\sigma')\} \geq 1 - \epsilon$, and thus, since $\phi \circ P \geq P(\mathcal{T}) \cdot (\phi \circ Q)$ pointwise, $\sum_{\sigma' \in \Sigma'} \min\{1/|\Sigma'|, \phi \circ P(\sigma')\} \geq P(\mathcal{T}) \cdot (1 - \epsilon)$. The conclusion of the lemma follows by setting $\epsilon = 1/2$.     $\square$

---

[11]Technically, we only need to do a union bound over half of all subsets, since $\phi \circ Q$ and $\mathcal{U}$ are both probability measures; hence the multiplicative factor in front is $2^{|\Sigma'|-1}$ as opposed to $2^{|\Sigma'|}$.

**Lemma E.4.** *Fix any $\phi : \Sigma \to \Sigma_{\mathsf{PRC}}$, and $i, n \in \mathbb{N}$. The distribution of $\mathcal{E}^{\phi}_{\mathsf{Emb}}(x; \mathsf{t}_{1:i-1})$, for $x \sim \mathrm{Unif}(\Sigma^n_{\mathsf{PRC}})$, is exactly the distribution of $\overline{\mathsf{Model}}(\mathsf{t}_{i:i+n-1} = \cdot \mid \mathsf{t}_{1:i-1})$.*

*Proof.* We use induction on $j \in [i, i+n-1]$. Fix any $j \in [i, i+n-1]$ together with a sequence $\mathsf{t}_{1:j-1} \in \Sigma^{j-1}$. Let $p_j \in \Delta(\Sigma)$ denote the distribution of $\mathsf{t}_j \sim \mathcal{E}^{\phi}_{\mathsf{Emb}}(x; \mathsf{t}_{1:i-1})_{j-i+1} \mid \mathsf{t}_{i:j-1}$, i.e., the distribution of the $j-i+1$th token of the output $\mathcal{E}^{\phi}_{\mathsf{Emb}}(x; \mathsf{t}_{1:i-1})$, conditioned on $\mathsf{t}_{i:j-1}$. Let $p^\star_j$ denote $\mathsf{Model}(\mathsf{t}_j = \cdot \mid \mathsf{t}_{1:j-1})$; we wish to show that $p_j = p^\star_j$.

By Line 21 of Algorithm 3, it suffices to show that $\phi \circ p_j = \phi \circ p^\star_j$. To do so, write $\bar{p}_j := \phi \circ p^\star_j$ (i.e., the quantity computed in Line 15 of Algorithm 3), and write $\rho_j := d_{\mathsf{TV}}(\bar{p}_j, \mathrm{Unif}(\Sigma_{\mathsf{PRC}})) = \sum_{\sigma \in \Sigma_{\mathsf{PRC}}} [\bar{p}_j(\sigma) - 1/|\Sigma_{\mathsf{PRC}}|]_+$, so that $\rho_j = 1 - \sum_{\sigma \in \Sigma_{\mathsf{PRC}}} \min\{\bar{p}_j(\sigma), 1/|\Sigma_{\mathsf{PRC}}|\}$. By definition, for each $\sigma \in \Sigma_{\mathsf{PRC}}$, we have

$$\phi \circ p_j(\sigma) = \frac{1}{|\Sigma_{\mathsf{PRC}}|} \cdot \min\{1, |\Sigma_{\mathsf{PRC}}| \cdot \bar{p}_j(\sigma)\} + \rho_j \cdot \frac{[\bar{p}_j(\sigma) - 1/|\Sigma_{\mathsf{PRC}}|]_+}{\rho_j} \bar{p}_j(\sigma),$$

as desired. $\qquad\square$

Given integers $a, b \in \mathbb{N}$ with $a < b$, a mapping $\phi : \Sigma \to \Sigma'$, and a sequence $\mathsf{t} = \mathsf{t}_{1:b} \in \Sigma^b$, we define the *spread* of the sequence with respect to $\phi$ to be

$$S^{\phi,[a:b)}(\mathsf{t}) := \sum_{i=a}^{b-1} \sum_{\sigma \in \Sigma'} \min\left\{ \frac{1}{|\Sigma'|}, \phi \circ P_i(\sigma) \right\}, \tag{28}$$

where $P_i(\sigma) := \mathsf{Model}(\mathsf{t}_i = \sigma \mid \mathsf{t}_{1:i-1})$.

Additionally, we define the *mean entropy* for the sequence $\mathsf{t} \in \Sigma^b$ in the interval $[a:b)$ to be

$$H^{[a:b)}_{\mathsf{m}}(\mathsf{t}) := \sum_{i=a}^{b-1} H(\mathsf{Model}(\mathsf{t}_i = \cdot \mid \mathsf{t}_{1:i-1})) = \sum_{i=a}^{b-1} \mathbb{E}[H^i_{\mathsf{e}}(\mathsf{t}) \mid \mathsf{t}_{1:i-1}].$$

**Lemma E.5.** *For any integers $a < b$, we have that*

$$\Pr_{\mathsf{t} \leftarrow \mathsf{Model}}\left( \frac{3}{2} H^{[a:b)}_{\mathsf{m}}(\mathsf{t}) + 8\ln|\Sigma| \cdot \ln^4(b-a) \geq H^{[a:b)}_{\mathsf{e}}(\mathsf{t}) \right) \geq 1 - \mathsf{negl}(b-a).$$

*Proof.* Let us write $\bar{H}^{[a:b)}_{\mathsf{e}}(\mathsf{t}) = \sum_{i=a}^{b-1} \min\{H^i_{\mathsf{e}}(\mathsf{t}), \ln|\Sigma| + \ln^2(b-a)\}$. For each $i$, we have that

$$\Pr(H^i_{\mathsf{e}}(\mathsf{t}) - \bar{H}^i_{\mathsf{e}}(\mathsf{t}) > 0) \leq |\Sigma| \cdot \exp(-\ln|\Sigma| - \ln^2(b-a)) = \exp(-\ln^2(b-a)). \tag{29}$$

By a union bound, it follows that $\Pr(H^{[a:b)}_{\mathsf{e}}(\mathsf{t}) > \bar{H}^{[a:b)}_{\mathsf{e}}(\mathsf{t})) \leq (b-a) \cdot \exp(-\ln^2(b-a))$. Next, Theorem F.2 gives that, for any $\delta > 0$,

$$\Pr_{\mathsf{t} \leftarrow \mathsf{Model}}\left( \bar{H}^{[a:b)}_{\mathsf{e}}(\mathsf{t}) - \frac{3}{2} \sum_{i=a}^{b-1} \mathbb{E}\left[ \bar{H}^i_{\mathsf{e}}(\mathsf{t}) \mid \mathsf{t}_{1:i-1} \right] > 4(\ln|\Sigma| + \ln^2(b-a)) \cdot \ln(2/\delta) \right) \leq \delta. \tag{30}$$

Finally, since $\bar{H}^i_{\mathsf{e}}(\mathsf{t}) \leq H^i_{\mathsf{e}}(\mathsf{t})$ with probability 1 for each $i \in [a, b)$, we have

$$\mathbb{E}[\bar{H}^i_{\mathsf{e}}(\mathsf{t}) \mid \mathsf{t}_{1:i-1}] \leq \mathbb{E}[H^i_{\mathsf{e}}(\mathsf{t}) \mid \mathsf{t}_{1:i-1}] = H^i_{\mathsf{m}}(\mathsf{t}). \tag{31}$$

Combining Eqs. (29) to (31) and choosing $\delta = 2\exp(-\ln^2(b-a))$, we see that with probability $1 - 2(b-a+1) \cdot \exp(-\ln^2(b-a)) \geq 1 - \mathsf{negl}(b-a)$, we have

$$H^{[a:b)}_{\mathsf{e}}(\mathsf{t}) \leq \bar{H}^{[a:b)}_{\mathsf{e}}(\mathsf{t}) \leq \frac{3}{2} \sum_{i=a}^{b-1} \mathbb{E}[H^i_{\mathsf{e}}(\mathsf{t}) \mid \mathsf{t}_{1:i-1}] + 4(\ln|\Sigma| + \ln^2(b-a)) \cdot \ln^2(2/\delta)$$

$$\leq \frac{3}{2} H^{[a:b)}_{\mathsf{m}}(\mathsf{t}) + 8\ln|\Sigma| \cdot \ln^4(b-a).$$

$\qquad\square$

**Lemma E.6.** *Let alphabets $\Sigma, \Sigma'$ be given, and let $\phi : \Sigma \to \Sigma'$ be a uniformly random function. For any integers $a < b$ and any $\alpha \in [0,1]$ satisfying $|\Sigma| \geq \left(\frac{8}{\alpha}|\Sigma'|\right)^{2/\alpha}$, we have that*

$$\Pr_{\mathsf{t} \leftarrow \overline{\mathsf{Model}}, \phi} \left( \begin{array}{c} H_{\mathsf{e}}^{[a:b]}(\mathsf{t}) \leq 3\alpha \cdot (b-a) \ln |\Sigma| + 8 \ln |\Sigma| \cdot \ln^4(b-a) \\ or\ S^{\phi,[a:b]}(\mathsf{t}) \geq \frac{\alpha \cdot (b-a)}{4} \end{array} \right) \geq 1 - \mathsf{negl}(b-a) - (b-a) \cdot 2^{-\frac{1}{8}\alpha|\Sigma|^\alpha}.$$

*Proof.* Let us fix any sequence $\mathsf{t} \in \Sigma^L$. For $i \in [L]$, define $P_i \in \Delta(\Sigma)$ by $P_i(\sigma) := \mathsf{Model}(\mathsf{t}_i = \sigma \mid \mathsf{t}_{1:i-1})$. By Lemma E.3, for each $i \in [a:b)$, we have, over a random (uniform) draw of $\phi$, that

$$\Pr_\phi \left( \sum_{\sigma' \in \Sigma'} \min\left\{ \frac{1}{|\Sigma'|}, \phi \circ P_i(\sigma') \right\} \geq \frac{H(P_i)}{4 \ln |\Sigma|} \right) \geq 1 - 2^{|\Sigma'| - \frac{H(P_i)\cdot\exp(H(P_i)/2)}{4 \ln |\Sigma|}},$$

where $c > 0$ is a universal constant. Thus, for each index $i \in [a, b-1]$ for which $H(P_i) = H_{\mathsf{m}}^i(\mathsf{t}) \geq \alpha \ln |\Sigma|$, we have

$$\Pr_\phi \left( \sum_{\sigma' \in \Sigma'} \min\left\{ \frac{1}{|\Sigma'|}, \phi \circ P_i(\sigma') \right\} \geq \frac{H(P_i)}{4 \ln |\Sigma|} \right) \geq 1 - 2^{|\Sigma'| - \frac{1}{4}\cdot\alpha|\Sigma|^{\alpha/2}} \geq 1 - 2^{-\frac{1}{8}\alpha|\Sigma|^{\alpha/2}} \quad (32)$$

where the second inequality follows by our requirement that $|\Sigma| \geq \left(\frac{8}{\alpha}|\Sigma'|\right)^{2/\alpha}$. For any sequence $\mathsf{t}$ for which $H_{\mathsf{m}}^{[a:b]}(\mathsf{t}) \geq 2\alpha \cdot (b-a) \ln |\Sigma|$, we must have that $\sum_{i \in [a,b):\ H_{\mathsf{m}}^i(\mathsf{t}) \geq \alpha \ln |\Sigma|} H_{\mathsf{m}}^i(\mathsf{t}) \geq \alpha \cdot (b-a) \ln |\Sigma|$. Thus, for any such $\mathsf{t}$, (32) together with a union bound gives that

$$\Pr_\phi \left( S^{\phi,[a:b]}(\mathsf{t}) \geq \frac{\alpha \cdot (b-a)}{4} \right) \geq 1 - (b-a) \cdot 2^{-\frac{1}{8}\alpha|\Sigma|^\alpha}.$$

Next, Lemma E.5 gives that

$$\Pr_{\mathsf{t} \leftarrow \overline{\mathsf{Model}}} \left( H_{\mathsf{e}}^{[a:b]}(\mathsf{t}) \leq 3\alpha \cdot (b-a) \ln |\Sigma| + 8 \ln |\Sigma| \cdot \ln^4(b-a) \text{ or } H_{\mathsf{m}}^{[a:b]}(\mathsf{t}) \geq 2\alpha \cdot (b-a) \ln |\Sigma| \right) \geq 1 - \mathsf{negl}(b-a).$$

Combining the two above displays, we see that

$$\Pr_{\mathsf{t} \leftarrow \overline{\mathsf{Model}}, \phi} \left( \begin{array}{c} H_{\mathsf{e}}^{[a:b]}(\mathsf{t}) \leq 3\alpha \cdot (b-a) \ln |\Sigma| + 8 \ln |\Sigma| \cdot \ln^4(b-a) \\ or\ S^{\phi,[a:b]}(\mathsf{t}) \geq \frac{\alpha \cdot (b-a)}{4} \end{array} \right) \geq 1 - \mathsf{negl}(b-a) - (b-a) \cdot 2^{-\frac{1}{8}\alpha|\Sigma|^\alpha}.$$

$\square$

**Lemma E.7.** *Fix $L \in \mathbb{N}$, alphabets $\Sigma, \Sigma_{\mathsf{PRC}}$, and integers $a, b \in [L]$. Suppose $\alpha \in [0,1]$ satisfies $\alpha \geq \frac{32 \cdot \ln^4(b-a)}{b-a}$ and $|\Sigma| \geq (\frac{8}{\alpha}|\Sigma'|)^{2/\alpha}$. Then for $x \sim \mathrm{Unif}(\Sigma_{\mathsf{PRC}}^L)$ and $\phi : \Sigma \to \Sigma_{\mathsf{PRC}}$ drawn uniformly, it holds that*

$$\Pr_{\substack{\phi, x \\ \mathsf{t}_{1:a-1} \sim \overline{\mathsf{Model}}(\cdot|\emptyset) \\ \mathsf{t}_{a:b-1} \sim \mathcal{E}_{\mathsf{Emb}}^\phi(x_{a:b-1}; \mathsf{t}_{1:a-1})}} \left( \begin{array}{c} H_{\mathsf{e}}^{[a:b]}(\mathsf{t}) \leq 4\alpha \cdot (b-a) \ln |\Sigma| \\ or\ D_{\mathsf{Ham}}(x_{a:b-1}, \phi(\mathsf{t}_{a:b-1})) \leq (b-a) \cdot (1-\alpha/8) \end{array} \right) \geq 1 - \mathsf{negl}(b-a) - (b-a) \cdot 2^{-\frac{1}{8}\alpha|\Sigma|^\alpha},$$

*where we recall that $\phi(\mathsf{t}_{a:b-1})$ denotes the string $(\phi(\mathsf{t}_a), \ldots, \phi(\mathsf{t}_{b-1}))$.*

*Proof.* For any fixed $\phi : \Sigma \to \Sigma_{\mathsf{PRC}}$ and $\mathsf{t}_{1:a-1} \in \Sigma$, Lemma E.4 gives that the distribution of $\mathcal{E}_{\mathsf{Emb}}^\phi(x_{a:b-1}; \mathsf{t}_{1:a-1})$, under $x_{a:b-1} \sim \mathrm{Unif}(\Sigma_{\mathsf{PRC}}^{b-a})$, is exactly the distribution of $\overline{\mathsf{Model}}(\mathsf{t}_{a:b-1} = \cdot \mid \mathsf{t}_{1:a-1})$. Thus, by Lemma E.6 and the fact that $\alpha \cdot (b-a) \ln |\Sigma| \geq 8 \ln |\Sigma| \cdot \ln^4(b-a)$, we have that

$$\Pr_{\substack{\phi, x \\ \mathsf{t}_{1:a-1} \sim \overline{\mathsf{Model}}(\cdot|\emptyset) \\ \mathsf{t}_{a:b-1} \sim \mathcal{E}_{\mathsf{Emb}}^\phi(x_{a:b-1}; \mathsf{t}_{1:a-1})}} \left( H_{\mathsf{e}}^{[a:b]}(\mathsf{t}) \leq 4\alpha \cdot (b-a) \ln |\Sigma| \text{ or } S^{\phi,[a:b]}(\mathsf{t}) \geq \frac{\alpha \cdot (b-a)}{4} \right)$$

$$\geq \Pr_{\substack{\phi, x \\ \mathsf{t}_{1:a-1} \sim \overline{\mathsf{Model}}(\cdot|\emptyset) \\ \mathsf{t}_{a:b-1} \sim \mathcal{E}_{\mathsf{Emb}}^\phi(x_{a:b-1}; \mathsf{t}_{1:a-1})}} \left( \begin{array}{c} H_{\mathsf{e}}^{[a:b]}(\mathsf{t}) \leq 3\alpha \cdot (b-a) \ln |\Sigma| + 8 \ln |\Sigma| \cdot \ln^4(b-a) \\ or\ S^{\phi,[a:b]}(\mathsf{t}) \geq \frac{\alpha \cdot (b-a)}{4} \end{array} \right)$$

$$\geq 1 - \mathsf{negl}(b-a) - (b-a) \cdot 2^{-\frac{1}{8}\alpha|\Sigma|^\alpha}. \quad (33)$$

(To be precise, the first inequality above uses the lower bound on $\alpha$ in the lemma statement and the second inequality uses Lemma E.6.) For each $i \in [L]$, let $p_i \in \Delta(\Sigma)$ denote the distribution $\mathsf{Model}(\mathsf{t}_i = \cdot \mid \mathsf{t}_1, \ldots, \mathsf{t}_{i-1})$, which is itself a random variable (depending on $\mathsf{t}_1, \ldots, \mathsf{t}_{i-1}$). For $i \in [a, b-1]$, let $Z_i \sim \mathrm{Ber}(\min\{1, |\Sigma_{\mathsf{PRC}}| \cdot (\phi \circ p_i)(x_{i-a+1})\})$ be the random variable used in the definition of $\mathcal{E}_{\mathsf{Emb}}^{\phi}(x_{a:b-1}; \mathsf{t}_{1:a-1})$ (i.e., corresponding to Line 16 of Algorithm 3). Note that the output $\mathsf{t}_{a:b-1} = \mathcal{E}_{\mathsf{Emb}}^{\phi}(x_{a:b-1}; \mathsf{t}_{1:a-1})$ of the embedding channel satisfies $\phi(\mathsf{t}_{i-a+1}) = x_{i-a+1}$ if $Z_i = 1$ for each $i \in [a, b-1]$. Therefore,

$$D_{\mathsf{Ham}}(x_{a:b-1}, \phi(\mathsf{t}_{a:b-1})) \leq \sum_{i=a}^{b-1}(1 - Z_i). \tag{34}$$

For each $i \in [a, b-1]$, note that

$$\mathbb{E}_{x \sim \mathrm{Unif}(\Sigma_{\mathsf{PRC}}^L)}[Z_i \mid \phi, \mathsf{t}_{1:i-1}, Z_{1:i-1}] = \sum_{\sigma \in \Sigma_{\mathsf{PRC}}} \frac{1}{|\Sigma_{\mathsf{PRC}}|} \cdot \min\{1, |\Sigma_{\mathsf{PRC}}| \cdot (\phi \circ p_i)(\sigma)\},$$

and using the definition of the spread in (28) with $\Sigma' = \Sigma_{\mathsf{PRC}}$, we see that

$$\sum_{i=a}^{b-1} \mathbb{E}_{x \sim \mathrm{Unif}(\Sigma_{\mathsf{PRC}}^L)}[Z_i \mid \phi, \mathsf{t}_{1:i-1}, Z_{1:i-1}] = \sum_{i=a}^{b-1} \sum_{\sigma \in \Sigma_{\mathsf{PRC}}} \min\left\{\frac{1}{|\Sigma_{\mathsf{PRC}}|}, (\phi \circ p_i)(\sigma)\right\} = S^{\phi, [a:b)}(\mathsf{t}).$$

By Theorem F.2, for any fixed $\phi$ and for any $\delta \in (0, 1)$, with probability $1 - \delta$ over the draw of $\mathsf{t}_{1:a-1} \sim \overline{\mathsf{Model}}(\cdot \mid \emptyset)$, $x \sim \mathrm{Unif}(\Sigma_{\mathsf{PRC}}^L)$, $\mathsf{t}_{a:b-1} \sim \mathcal{E}_{\mathsf{Emb}}(x_{a:b-1}; \mathsf{t}_{1:a-1})$, and $Z_i$, it holds that

$$\sum_{i=a}^{b-1} Z_i \geq \frac{1}{2} S^{\phi, [a:b)}(\mathsf{t}) - 4\log(2/\delta). \tag{35}$$

Choose $\delta = 2\exp(-\ln^2(b-a)) \leq \mathsf{negl}(b-a)$. In the event that $S^{\phi, [a:b)}(\mathsf{t}) \geq \frac{\alpha \cdot (b-a)}{4}$ and (35) both hold, using the lower bound on $\alpha$ in the lemma statement together with (34), we see that

$$D_{\mathsf{Ham}}(x_{a:b-1}, \phi(\mathsf{t}_{a:b-1})) \leq (b-a) - \frac{\alpha \cdot (b-a)}{4} + 4\ln^2(b-a) \leq (b-a) \cdot (1 - \alpha/8). \tag{36}$$

Combining (33) and the fact that (35) holds with probability $1 - \mathsf{negl}(b-a)$ together with (36), we see that

$$\Pr_{\substack{\phi, x \\ \mathsf{t}_{1:a-1} \sim \overline{\mathsf{Model}}(\cdot \mid \emptyset) \\ \mathsf{t}_{a:b-1} \sim \mathcal{E}_{\mathsf{Emb}}^{\phi}(x_{a:b-1}; \mathsf{t}_{1:a-1})}} \left(\begin{array}{c} H_e^{[a:b)}(\mathsf{t}) \leq 4\alpha \cdot (b-a)\ln|\Sigma| \\ \text{or } D_{\mathsf{Ham}}(x_{a:b-1}, \phi(\mathsf{t}_{a:b-1})) \leq (b-a) \cdot (1-\alpha/8) \end{array}\right) \geq 1 - \mathsf{negl}(b-a) - (b-a) \cdot 2^{-\frac{1}{8}\alpha|\Sigma|^{\alpha}}.$$

$\square$

**Lemma E.8** (Substring robustness of the watermark). *Suppose that $p, \alpha \in (0, 1)$ are given, and* $\mathsf{PRC}$ *is zero-bit PRC with block length $n(\lambda)$ over alphabet $\Sigma_{\mathsf{PRC}}(\lambda)$, satisfying $|\Sigma_{\mathsf{PRC}}(\lambda)| \geq n(\lambda)$. Suppose further that $\mathsf{PRC}$ is robust to any $(1 - \frac{\alpha}{8} + \frac{3p}{\alpha})$-edit-bounded channel. Let $\mathsf{Model}(\lambda)$ be defined over some alphabet $\Sigma(\lambda)$ satisfying $|\Sigma(\lambda)| \geq (\frac{8}{\alpha}|\Sigma_{\mathsf{PRC}}(\lambda)|)^{2/\alpha}$. Then the watermarking scheme $\mathcal{W}[\mathsf{PRC}, \mathsf{Model}]$ (defined in Algorithm 3) is $\beta_\lambda(\ell)$-substring robust to any $p$-edit-bounded channel $\mathcal{E}$, where $\beta_\lambda(\ell) = 8n(\lambda) + 6\alpha\ell$.*

*Proof.* For convenience we write $p_1 := \alpha/8$, $p_0 := 3p/\alpha$; then $\mathsf{PRC}$ is robust to any $(1 - p_1 + p_0)$-bounded channel. Let $\lambda$ denote the security parameter for the given $\mathsf{PRC}$. We will show that for all $i, j \in [L_{\mathsf{max}}(\lambda)]$,

$$\Pr_{\substack{(\mathsf{sk}, \phi) \leftarrow \mathsf{Setup}(1^\lambda) \\ \mathsf{t} \leftarrow \mathsf{Wat}(1^\lambda, (\mathsf{sk}, \phi)), \mathsf{t}' \leftarrow \mathcal{E}(\mathsf{t})}} \left(\mathsf{Detect}(1^\lambda, (\mathsf{sk}, \phi), \mathsf{t}') = \mathsf{False} \text{ and } H_e^{[i:j)}(\mathsf{t}) \geq \beta_\lambda(j-i) \cdot \ln|\Sigma(\lambda)|\right) \leq \mathsf{negl}(\lambda).$$

(37)

Using (37), the fact that $L_{\mathsf{max}}(\lambda) \leq \mathrm{poly}(\lambda)$, and a union bound over all possible choices of $i, j$ will yield the desired claim of $\beta_\lambda(\ell)$-substring robustness. To establish (37), fix $\lambda \in \mathbb{N}$, and set $\mathsf{Model} = \mathsf{Model}(\lambda)$, $n = n(\lambda)$, $\Sigma = \Sigma(\lambda)$, $\Sigma_{\mathsf{PRC}} = \Sigma_{\mathsf{PRC}}(\lambda)$, as well as $i, j \in [L_{\mathsf{max}}(\lambda)]$; we argue in two parts:

**Step 1: defining channels.** Let us write $\ell := j - i$ and let $a_1 < a_2 < \cdots < a_h$ be the indices in $[i, j-1]$ denoting the start positions of blocks for the PRC in the execution of $\mathsf{Wat}(1^\lambda, (\mathsf{sk}, \phi))$ (in particular, $a_1, \ldots, a_h$ are simply the integers in $[i, j-1]$ congruent to $1 \pmod n$). We will write $\mathsf{t}' := \mathcal{E}(\mathsf{t}_{i:j-1})$ to denote the output of the edit-bounded channel $\mathcal{E}$ given $\mathsf{t}_{i:j-1}$ as input. Since $\mathcal{E}$ is $p_0$-edit-bounded, we have $\mathsf{len}(\mathsf{t}') \leq \ell' := \lfloor (1 + p_0) \cdot \ell \rfloor$. Given a mapping $\phi : \Sigma \to \Sigma_{\mathsf{PRC}}$, we let $\mathcal{F}^\phi(x, \mathsf{t})$ (given $x \in \Sigma_{\mathsf{PRC}}^L, \mathsf{t} \in \Sigma^L$) denote the joint distribution over random variables $(y_{g,b,b'})_{g \in [h-1], b, b' \in [\ell']: \, b' - b \in [n(1-p_0), n(1+p_0)]}$, defined as follows:

- It draws $\mathsf{t}' \sim \mathcal{E}(\mathsf{t}_{i:j-1})$.

- For each choice of $g, b, b'$ as above, it lets $y_{g,b,b'}$ to be either: (a) the substring $\mathsf{t}'_{b:b'-1}$, if $\phi(\mathsf{t}'_{b:b'-1})$ can be obtained from $x_{a_g:a_{g+1}-1}$ via a sequence of at most $(1 - p_1 + p_0)n$ substitutions, insertions, and deletions, or (b) if not, any string satisfying $\phi(y_{g,b,b'}) = x_{a_g:a_{g+1}-1}$.[12]

Next, given $\phi, g, b, b'$ as above, we let $\mathcal{E}_{g,b,b'}^\phi$ be the channel which, given as input a string $x \in \Sigma_{\mathsf{PRC}}^n$, performs the following operations:

- First, it generates $(\mathsf{sk}, \phi) \sim \mathsf{Setup}(1^\lambda)$, and generates $\mathsf{t} \sim \mathsf{Wat}(1^\lambda, (\mathsf{sk}, \phi))$, with the following modification: when generating output for the block starting at index $a_g$, instead of using a fresh output of $\mathsf{PRC.Encode}(1^\lambda, \mathsf{sk})$, it uses the given input string $x$. Let $x^1, \ldots, x^{\lceil L_{\mathsf{max}}(\lambda)/n(\lambda) \rceil} \in \Sigma_{\mathsf{PRC}}^n$ denote the codewords (output by $\mathsf{PRC.Encode}(1^\lambda, \mathsf{sk})$) used in the $\mathsf{Wat}$ procedure, so that in particular $x^{a_g} = x$. Write $\bar{x} = (x^1, \ldots, x^{\lceil L_{\mathsf{max}}(\lambda)/n(\lambda) \rceil})$.

- Then, it samples from the marginal $y_{g,b,b'} \sim \mathcal{F}^\phi(\bar{x}, \mathsf{t})$ and outputs $\phi(y_{g,b,b'})$.

**Claim E.9.** *For any $g \in [h-1]$, $b, b' \in [\ell']$, and $\phi : \Sigma \to \Sigma_{\mathsf{PRC}}$, the channel $\mathcal{E}_{g,b,b'}^\phi$ is $(1 - p_1 + p_0)$-edit-bounded.*

*Proof.* It is immediate from the definition of $\mathcal{E}_{g,b,b'}^\phi$ and $\mathcal{F}^\phi$ that, almost surely, for a sample $y_{g,b,b'} \sim \mathcal{F}^\phi(x, \mathsf{t})$, $\phi(y_{g,b,b'})$ can be obtained from $x$ via a sequence of at most $(1 - p_1 + p_0)n$ substitutions, insertions, and deletions.

Finally, an output of $\mathcal{E}_{g,b,b'}^\phi$ can be sampled in polynomial time: here we use that $\mathsf{t} \sim \mathsf{Wat}(1^\lambda, (\mathsf{sk}, \phi))$ can be sampled in polynomial time (as $\mathsf{Model}$ is assumed to be computationally efficient), as well as the fact that it can be determined in polynomial time if one string (namely, $\phi(\mathsf{t}'_{b:b'-1})$ above) can be obtained from another string (namely, $x_{a_g:a_{g+1}-1}$ above) via a sequence of a given number (i.e., $(1 - p_1 + p_0)n$) substitutions, insertions, and deletions. $\square$

**Claim E.10.** *For each $g \in [h-1]$, it holds that*

$$\Pr_{\substack{(\mathsf{sk}, \phi) \leftarrow \mathsf{Setup}(1^\lambda) \\ x^1, x^2, \ldots, x^{\lceil L_{\mathsf{max}}(\lambda)/n(\lambda) \rceil} \leftarrow \mathsf{PRC.Encode}(1^\lambda, \mathsf{sk}) \\ x = (x^1, x^2, \ldots, x^{\lceil L_{\mathsf{max}}(\lambda)/n(\lambda) \rceil}) \\ \mathsf{t} \sim \mathcal{E}_{\mathsf{Emb}}^\phi(x; \emptyset)}} \left( \begin{array}{c} H_e^{[a_g:a_{g+1})}(\mathsf{t}) \leq 4\alpha \cdot n \ln |\Sigma| \\ or\ D_{\mathsf{Ham}}(x_{a_g:a_{g+1}-1}, \phi(\mathsf{t}_{a_g:a_{g+1}-1})) \leq n \cdot (1 - \alpha/8) \end{array} \right) \geq 1 - \mathsf{negl}(\lambda).$$

(38)

We emphasize that the notation $x^1, x^2, \ldots \leftarrow \mathsf{PRC.Encode}(1^\lambda, \mathsf{sk})$, $\mathsf{t} \sim \mathcal{E}_{\mathsf{Emb}}^\phi(x; \emptyset)$ means that $\mathsf{PRC.Encode}(1^\lambda, \mathsf{sk})$ is called repeatedly to produce strings $x^1, x^2, \ldots \in \Sigma_{\mathsf{PRC}}^n$, and the concatenated string $x = (x^1, x^2, \ldots, x^{\lceil L_{\mathsf{max}}(\lambda)/n(\lambda) \rceil})$ is used as input to the embedding channel $\mathcal{E}_{\mathsf{Emb}}^\phi(x; \emptyset)$. Notice that this procedure is identical to $\mathsf{Wat}(1^\lambda, (\mathsf{sk}, \phi))$, meaning that the distribution of the output string $\mathsf{t}$ is identical to the output distribution of $\mathsf{Wat}(1^\lambda, (\mathsf{sk}, \phi))$.

---

[12]If $b$ or $b'$ are outside the length bounds of $\mathsf{t}'$, then we let $\mathsf{t}'_{b:b'-1}$ denote the substring $\mathsf{t}'_{\min\{b, \mathsf{len}(\mathsf{t}')\}:\min\{b'-1, \mathsf{len}(\mathsf{t}')\}}$, which may be empty.

*Proof of Claim E.10.* Let $\mathcal{A}$ denote the event that $H_{\mathsf{e}}^{[a_g : a_{g+1}]}(\mathsf{t}) \leq 4\alpha n \ln |\Sigma|$ or $D_{\mathsf{Ham}}(x_{a_g : a_{g+1}-1}, \phi(\mathsf{t}_{a_g : a_{g+1}-1})) \leq n \cdot (1 - \alpha/8)$. We have that $\frac{32 \cdot \ln^4(n)}{n} \leq \alpha$ as long as $n = n(\lambda) \geq \tilde{\Omega}(1/\alpha)$, which holds for all $\lambda$ greater than a sufficiently large constant depending on $\alpha$ (and it is sufficient for us to only consider such $\lambda$ as the claimed failure probability is $\mathsf{negl}(\lambda)$). Thus, we may apply Lemma E.7 with $a = i, b = j, \Sigma' = \Sigma_{\mathsf{PRC}}$; doing so, we see that if $\phi : \Sigma \to \Sigma_{\mathsf{PRC}}$ is drawn uniformly and $x \sim \mathrm{Unif}(\Sigma^{L_{\max}(\lambda)})$, the event $\mathcal{A}$ occurs with probability $1 - \mathsf{negl}(n) - n \cdot 2^{-\frac{1}{8}\alpha |\Sigma|^\alpha} \geq 1 - \mathsf{negl}(\lambda) - n \cdot 2^{-\frac{1}{8}\alpha |\Sigma|^\alpha}$ (here we have used $n \geq \lambda$). Since a draw of $\mathsf{t} \sim \mathcal{E}_{\mathsf{Emb}}^\phi(x; \emptyset)$ may be implemented in polynomial time (as we have assumed that Model is computationally efficient), and since $H_{\mathsf{e}}^{[a_g : a_{g+1}]}(\mathsf{t}), D_{\mathsf{Ham}}(x_{a_g : a_{g+1}-1}, \phi(\mathsf{t}_{a_g : a_{g+1}-1}))$ can be computed in polynomial time, it follows that $\mathcal{A}$ must occur with probability $1 - \mathsf{negl}(\lambda) - n \cdot 2^{-\frac{1}{8}\alpha |\Sigma|^\alpha}$ when $x^1, x^2, \ldots, x^{\lceil L_{\max}(\lambda)/n(\lambda) \rceil} \leftarrow \mathsf{PRC.Encode}(1^\lambda, \mathsf{sk})$. (Otherwise, we could violate undetectability of PRC by generating codewords $x^1, x^2, \ldots$ from the encoding oracle $\mathcal{O}$, representing either PRC.Encode or a random oracle, and checking whether $\mathcal{A}$ occurs.)

Finally, we remark that $n \cdot 2^{-\frac{1}{8}\alpha |\Sigma|^\alpha} \leq \mathsf{negl}(n) \leq \mathsf{negl}(\lambda)$ by our requirement that $|\Sigma|^\alpha \geq |\Sigma_{\mathsf{PRC}}| \geq n$. $\qquad \square$

Since the channel $\mathcal{E}$ is $p$-edit-bounded, for most values of $g \in [h-1]$ indexing full blocks of $\mathsf{t}_{i:j-1}$, there must be some $b, b'$ so that $\mathsf{t}'_{b:b'-1}$ can be obtained from $\mathsf{t}_{a_g : a_{g+1}-1}$ by at most $O(pn)$ insertions and deletions; this is formalized in the below claim:

**Claim E.11.** *Fix any $C \geq 1$ and suppose $h \geq 2$. For any string $\mathsf{t}_{i:j-1} \in \Sigma^\ell$, there are at least $\lfloor (h-1) \cdot (1 - 1/C) \rfloor$ values of $g \in [h-1]$ so that there exist $b, b' \in [\ell']$ for which $\mathsf{t}'_{b:b'-1}$ can be obtained from $\mathsf{t}_{a_g : a_{g+1}-1}$ by applying at most $3Cp \cdot n$ substitutions, insertions, and deletions (where we have $\mathsf{t}' = \mathcal{E}(\mathsf{t}_{i:j-1})$).*

*Proof of Claim E.11.* For each $g \in [h-1]$ and $\mathsf{t}_{i:j-1} \in \Sigma^\ell$, the edit-bounded channel $\mathcal{E}$ sends the substring $\mathsf{t}_{a_g : a_{g+1}-1}$ to some substring $\mathsf{t}'_{b_g : b_{g+1}-1}$ of the (possibly random) channel output $\mathsf{t}' = \mathcal{E}(\mathsf{t}_{i:j-1})$. (In particular, $b_g$ denotes the index in $\mathsf{t}'$ of the first token in $\mathsf{t}_{a_g : a_{g+1}-1}$ which is not deleted by $\mathcal{E}$, or, if all of $\mathsf{t}_{a_g : a_{g+1}-1}$ is deleted, the index in $\mathsf{t}'$ of the first token following $\mathsf{t}_{a_{g+1}-1}$ which is not deleted by $\mathcal{E}$.) Note that $b_g$ is a function of $\mathsf{t}_{i:j-1}$; we omit this dependence from the notation to avoid clutter.

Fix $\mathsf{t}_{i:j-1}$ and $\mathsf{t}' = \mathcal{E}(\mathsf{t}_{i:j-1})$. For each $g \in [h-1]$, let $e_g$ denote the number of substitutions, insertions, and deletions applied by $\mathcal{E}$ to obtain $\mathsf{t}'_{b_g : b_{g+1}-1}$ from $\mathsf{t}_{a_g : a_{g+1}-1}$. Since $\mathcal{E}$ is $p$-edit-bounded, we have $\sum_{g=1}^{h-1} e_g \leq p\ell \leq p \cdot (h+1)n$. Therefore, at least $\lfloor (h-1) \cdot (1 - 1/C) \rfloor$ values of $g \in [h-1]$ satisfy $e_g \leq \frac{C}{h-1} \cdot p(h+1)n \leq 3Cp \cdot n$, as desired. $\qquad \square$

**Step 2: using robustness of the PRC.** Since each channel $\mathcal{E}_{g,b,b'}^\phi$ (for any $\phi : \Sigma \to \Sigma_{\mathsf{PRC}}$, $g \in [h-1]$, $b, b' \in [\ell']$ with $b' - b \in [n(1-p_0), n(1+p_0)]$) is $(1 - p_1 + p_0)$-bounded (by Claim E.9), it holds that

$$\Pr_{\substack{(\mathsf{sk}, \phi) \leftarrow \mathsf{Setup}(1^\lambda) \\ x \leftarrow \mathsf{PRC.Encode}(1^\lambda, \mathsf{sk}) \\ y \sim \mathcal{E}_{g,b,b'}^\phi(x)}} \left( \mathsf{PRC.Decode}(1^\lambda, \mathsf{sk}, y) \neq \perp \right) \geq 1 - \mathsf{negl}(\lambda). \tag{39}$$

Note that the distribution of $y \sim \mathcal{E}_{g,b,b'}^\phi(x)$, for $x \leftarrow \mathsf{PRC.Encode}(1^\lambda, \mathsf{sk})$, is the same as the marginal distribution of $\phi(y_{g,b,b'}) \sim \mathcal{F}^\phi(x, \mathsf{t})$, for $x_1, x_2, \ldots \leftarrow \mathsf{PRC.Encode}(1^\lambda, \mathsf{sk})$, $x := (x^1, \ldots, x^{\lceil L_{\max}(\lambda)/n(\lambda) \rceil})$, $\mathsf{t} \sim \mathcal{E}_{\mathsf{Emb}}^\phi(x; \emptyset)$. Therefore, using (39) together with a union bound over the (polynomially many) choices of $g, b, b'$, we see that

$$\Pr_{\substack{(\mathsf{sk}, \phi) \leftarrow \mathsf{Setup}(1^\lambda) \\ x^1, x^2, \ldots, x^{\lceil L_{\max}(\lambda)/n(\lambda) \rceil} \leftarrow \mathsf{PRC.Encode}(1^\lambda, \mathsf{sk}) \\ x = (x^1, x^2, \ldots, x^{\lceil L_{\max}(\lambda)/n(\lambda) \rceil}), \, \mathsf{t} \sim \mathcal{E}_{\mathsf{Emb}}^\phi(x; \emptyset) \\ (y_{g,b,b'})_{g,b,b'} \sim \mathcal{F}^\phi(x, \mathsf{t})}} \left( \forall g, b, b' : \mathsf{PRC.Decode}(1^\lambda, \mathsf{sk}, \phi(y_{g,b,b'})) \neq \perp \right) \geq 1 - \mathsf{negl}(\lambda).$$

$$\tag{40}$$

For any $i \in [L_{\mathsf{max}}(\lambda)]$, by Lemma E.5 with $a = i, b = i + n$, there is some event $\mathcal{B}_i$ that occurs with probability $1 - \mathsf{negl}(\ell) \geq 1 - \mathsf{negl}(\lambda)$ under $\mathsf{t} \leftarrow \overline{\mathsf{Model}}$ so that, under $\mathcal{B}_i$,

$$H_{\mathsf{e}}^{[i:i+n)}(\mathsf{t}) \leq \frac{3}{2} H_{\mathsf{m}}^{[i:i+n)}(\mathsf{t}) + 8 \ln |\Sigma| \cdot \ln^4(n) \leq \frac{3}{2} n \ln |\Sigma| + 8 \ln |\Sigma| \cdot \ln^4(n) \leq 2n \ln |\Sigma|,$$

where the final inequality holds as long as $\lambda$ is chosen sufficiently large so that $n(\lambda) \geq 20 \ln^4(n(\lambda))$. By undetectability of PRC, Lemma E.4, and efficiency of Model, the event $\mathcal{B}_i$ also holds with probability $1 - \mathsf{negl}(\lambda)$ under the distribution of $\mathsf{t}$ specified in (38). Let us write $\mathcal{B} := \mathcal{B}_{a_1 - n} \cap \mathcal{B}_{a_1} \cap \cdots \cap \mathcal{B}_{a_h}$; thus, under $\mathcal{B}$, we have

$$H_{\mathsf{e}}^{[i:a_1)}(\mathsf{t}) \leq 2n \ln |\Sigma|, \qquad H_{\mathsf{e}}^{[a_g:a_{g+1})} \leq 2n \ln |\Sigma| \ \forall g \in [h - 1], \qquad H_{\mathsf{e}}^{[a_h:j)}(\mathsf{t}) \leq 2n \ln |\Sigma|.$$

Consider the distribution of $(x, \mathsf{t}, \mathsf{t}')$ where $(x, \mathsf{t})$ are drawn as specified in (38) and $\mathsf{t}' \sim \mathcal{E}(\mathsf{t}_{i:j-1})$. Let $\mathcal{A}$ denote the event that for all $g \in [h - 1]$, $H_{\mathsf{e}}^{[a_g:a_{g+1})} \leq 4\alpha \cdot n \ln |\Sigma|$ or $D_{\mathsf{Ham}}(x_{a_g:a_{g+1}-1}, \phi(\mathsf{t}_{a_g:a_{g+1}-1})) \leq n \cdot (1 - \alpha/8)$; by Claim E.10 and a union bound over $g \in [h-1]$, $\mathcal{A}$ holds with probability $1 - \mathsf{negl}(\lambda)$.

Under the event $\mathcal{A} \cap \mathcal{B}$ (which occurs with probability $1 - \mathsf{negl}(\lambda)$), one of the following must be the case:

- There are at least $\lfloor \alpha \cdot (h - 1) \rfloor + 2$ values of $g \in [h - 1]$ so that $D_{\mathsf{Ham}}(x_{a_g:a_{g+1}-1}, \phi(\mathsf{t}_{a_g:a_{g+1}-1})) \leq n \cdot (1 - \alpha/8)$; let this event be denoted $\mathcal{C}_1$.

- It holds that $H_{\mathsf{e}}^{[i:j)} \leq 8n \ln |\Sigma| + 6\alpha \ell \ln |\Sigma|$; let this event be denoted $\mathcal{C}_2$.

Indeed, under $\mathcal{A} \cap \mathcal{B}$, if $\mathcal{C}_1$ does not occur, then we must have that $\mathcal{C}_2$ occurs since we would have:

$$\begin{aligned}
H_{\mathsf{e}}^{[i:j)} =&\, H_{\mathsf{e}}^{[i:a_1)}(\mathsf{t}) + H_{\mathsf{e}}^{[a_h:j)}(\mathsf{t}) + \sum_{g=1}^{h-1} H_{\mathsf{e}}^{[a_g:a_{g+1})}(\mathsf{t}) \\
\leq&\, 4n \ln |\Sigma| + (\lfloor \alpha \cdot (h - 1) \rfloor + 2) \cdot 2n \ln |\Sigma| + (h - 1 - \lfloor \alpha \cdot (h - 1) \rfloor - 1) \cdot 4\alpha n \ln |\Sigma| \\
\leq&\, 8n \ln |\Sigma| + 2\alpha \ell \ln |\Sigma| + 4\alpha \ell \ln |\Sigma| = 8n \ln |\Sigma| + 6\alpha \ell \ln |\Sigma|,
\end{aligned}$$

where the first inequality uses that $\mathcal{A} \cap \mathcal{B}$ occurs and the second inequality uses the fact that $(h - 1)n \leq \ell$.

Note that if $h < 2$, under the event $\mathcal{B}$, the event $\mathcal{C}_2$ occurs.

Next, if $h \geq 2$, by Claim E.11 with $C = 1/\alpha$, for any string $\mathsf{t}_{i:j-1} \in \Sigma^\ell$, with probability 1 over $\mathsf{t}' \sim \mathcal{E}(\mathsf{t}_{i:j-1})$, there are at least $\lfloor (h - 1) \cdot (1 - \alpha) \rfloor$ values of $g \in [h - 1]$ so that there exist $b, b' \in [\ell']$ for which $\mathsf{t}'_{b:b'-1}$ can be obtained from $\mathsf{t}_{a_g:a_{g+1}-1}$ by applying at most $\frac{3}{\alpha} \cdot pn$ substitutions, insertions and deletions. Note that $\lfloor \alpha \cdot (h - 1) \rfloor + 2 + \lfloor (h - 1) \cdot (1 - \alpha) \rfloor > h - 1$. Thus, under the event $\mathcal{A} \cap \mathcal{B} \cap \mathcal{C}_1$, there is some value of $g_\star \in [h - 1]$ and $b_\star, b'_\star \in [\ell']$ so that $D_{\mathsf{Ham}}(x_{a_{g_\star}:a_{g_\star+1}-1}, \phi(\mathsf{t}_{a_{g_\star}:a_{g_\star+1}-1})) \leq n \cdot (1 - \alpha/8) = n \cdot (1 - p_1)$ and $\mathsf{t}'_{b_\star:b'_\star-1}$ can be obtained from $\mathsf{t}_{a_{g_\star}:a_{g_\star+1}-1}$ by applying at most $p_0 n = \frac{3}{\alpha} \cdot pn$ substitutions, insertions and deletions. By definition of $\mathcal{F}^\phi(x, \mathsf{t})$, we may couple a draw of $(y_{g,b,b'})_{g,b,b'} \sim \mathcal{F}^\phi(x, \mathsf{t})$ to this distribution over $(x, \mathsf{t}, \mathsf{t}')$ so that under the event $\mathcal{A} \cap \mathcal{B} \cap \mathcal{C}_1$ that $(g_\star, b_\star, b'_\star)$ as above exist, we have $y_{g_\star, b_\star, b'_\star} = \mathsf{t}'_{b_\star:b'_\star-1}$.

Let the event of (40) (which holds with probability $1 - \mathsf{negl}(\lambda)$) be denoted by $\mathcal{D}$. Then, if $h \geq 2$, under the event $\mathcal{A} \cap \mathcal{B} \cap \mathcal{D} \cap \mathcal{C}_1$, $g_\star, b_\star, b'_\star$ are well-defined as above, and so we have that $\mathsf{PRC}.\mathsf{Decode}(1^\lambda, \mathsf{sk}, \phi(\mathsf{t}'_{g_\star, b_\star, b'_\star})) \neq \perp$, and in particular, that $\mathsf{Detect}(1^\lambda, (\mathsf{sk}, \phi), \mathsf{t}') = \mathsf{True}$. Under the event $\mathcal{A} \cap \mathcal{B} \cap \mathcal{D} \cap \mathcal{C}_2 \subset \mathcal{C}_2$ (which must occur under the event $\mathcal{B}$ if $h < 2$), we have that $H_{\mathsf{e}}^{[i:j)}(\mathsf{t}) \leq 8n \ln |\Sigma| + 6\alpha \ell \ln |\Sigma| = \beta_\lambda(\ell) \cdot \ln |\Sigma|$. Since $(\mathcal{A} \cap \mathcal{B} \cap \mathcal{D} \cap \mathcal{C}_1) \cup (\mathcal{A} \cap \mathcal{B} \cap \mathcal{D} \cap \mathcal{C}_2) \supseteq \mathcal{A} \cap \mathcal{B} \cap \mathcal{D}$ occurs with probability $1 - \mathsf{negl}(\lambda)$, we have established (37). $\qquad \square$

**Lemma E.12** (Soundness of the watermark). *For any PRC* $\mathsf{PRC}$*, the watermarking scheme* $\mathcal{W}[\mathsf{PRC}]$ *(defined in Algorithm 3) is sound.*

*Proof.* Fix any sequence $\mathsf{t} \in \Sigma^\ell$, for some $\ell \leq \lceil L_{\mathsf{max}}(\lambda)/n(\lambda) \rceil$. We wish to show that

$$\Pr_{(\mathsf{sk}, \phi) \leftarrow \mathsf{Setup}(1^\lambda)} \left( \mathsf{Detect}(1^\lambda, (\mathsf{sk}, \phi), \mathsf{t}) \neq \perp \right) \leq \mathsf{negl}(\lambda). \tag{41}$$

By definition of Detect (in Algorithm 3), $\mathsf{Detect}(1^\lambda, (\mathsf{sk}, \phi), \mathsf{t}) \neq \perp$ only if there are some $i, j \in [\ell]$ so that $\mathsf{PRC.Decode}(\mathsf{sk}, \phi(\mathsf{t}_{i:j})) \neq \perp$. Since PRC is sound, for each fixed choice of $\phi$ and $i, j$,

$$\Pr_{\mathsf{sk} \sim \mathsf{PRC.KeyGen}(1^\lambda)} \left( \mathsf{PRC.Decode}(\mathsf{sk}, \phi(\mathsf{t}_{i:j})) \neq \perp \right) \leq \mathsf{negl}(\lambda).$$

Taking expectation over $\phi$ (which is drawn independently from $\mathsf{sk}$) and a union bound over $i, j \leq \ell \leq \mathrm{poly}(\lambda)$, we see that (41) holds. $\qquad \square$

**Lemma E.13** (Undetectability of the watermark). *For any PRC* PRC*, the watermarking scheme* $\mathcal{W}[\mathsf{PRC}]$ *(defined in Algorithm 3) is undetectable.*

*Proof.* Suppose for the sake of contradiction that there is a polynomial-time adversary Adv so that

$$\left| \Pr \left( \mathsf{Adv}^{\overline{\mathsf{Model}}}(1^\lambda) = 1 \right) - \Pr_{\mathsf{sk} \sim \mathsf{Setup}(1^\lambda)} \left( \mathsf{Adv}^{\mathsf{Wat}(1^\lambda, \mathsf{sk})}(1^\lambda) = 1 \right) \right| \geq \frac{1}{\mathrm{poly}(\lambda)}. \qquad (42)$$

Using Adv, we can construct an adversary $\mathsf{Adv}'$ to break the undetectability of PRC, as follows. $\mathsf{Adv}'$ has access to an oracle $\mathcal{O}$ which is either $\mathsf{PRC.Encode}(1^\lambda, \mathsf{sk})$ (where $\mathsf{sk} \leftarrow \mathsf{PRC.KeyGen}(1^\lambda)$) or outputs a uniformly random string of length $n(\lambda)$ at each call. $\mathsf{Adv}'$ then draws $\phi : \Sigma \to \Sigma_{\mathsf{PRC}}$ uniformly at random and simulates Adv, simulating each call to $\mathsf{Wat}(1^\lambda, (\mathsf{sk}, \phi))$ as in Algorithm 3 but replacing each call to $\mathsf{PRC.Encode}(1^\lambda, \mathsf{sk})$ (Line 13 of Algorithm 3) with a call to $\mathcal{O}$. In the case that $\mathcal{O}$ is $\mathsf{PRC.Encode}(1^\lambda, \mathsf{sk})$, then this procedure is exactly $\mathsf{Adv}^{\mathsf{Wat}(1^\lambda, \mathsf{sk})}$. In the case that $\mathcal{O}$ outputs uniformly random strings in response to each call, then Lemma E.4 gives that this procedure is exactly $\mathsf{Adv}^{\overline{\mathsf{Model}}}$. Thus, (42) gives that $\mathsf{Adv}'$ can distinguish with inverse polynomial advantage between the two cases, a contradiction to undetectability of PRC. Note that in order to simulate Wat, $\mathsf{Adv}'$ needs to be able to evaluate $\mathsf{Model}(\mathsf{t}_i = \cdot \mid \mathsf{t}_{1:i-1})$ (Line 9 of Algorithm 3), which is possible in polynomial time by assumption on Model (see Appendix B). $\qquad \square$

*Proof of Theorem E.2.* Fix $\alpha > 0$ together with a function family $\mathcal{F}$ which is a $\log n(\lambda)$-local weak PRF for noise level $q \in [0, 1/2)$, and a family of language models $\mathsf{Model}(\lambda)$ as in the theorem statement. Write $p_0 := \frac{\alpha}{16 C_{\mathsf{rob}}}$, where $C_{\mathsf{rob}}$ is the constant from Theorem 4.1.

By Theorem 3.2, $\mathsf{PRC}_1 := \mathsf{PRF\text{-}PRC}[\mathcal{F}, \frac{1}{2} - p_0, q]$ is a zero-bit binary alphabet PRC with robustness to all $(\frac{1}{2} - p_0)$-bounded substitution channels. By definition in (3), the block length $N(\lambda)$ of $\mathsf{PRC}_1$ satisfies:

$$N(\lambda) = O\left( n(\lambda)^{4 \log(1/p_0)} \cdot n(\lambda)^2 \right) \leq n(\lambda)^{O(\log(1/p_0))}.$$

Next recall that $C_0, C_{\mathsf{rob}} \geq 1$ are the constants from Theorem 4.1. By Theorem 4.1, for $\rho = C_0/p_0$, $\mathsf{PRC}_2 := \mathsf{PRC}_{\mathsf{Idx}}[\mathsf{PRC}_1, \rho]$ is a zero-bit PRC with block length at most $N(\lambda)$ and alphabet $\Sigma_{\mathsf{PRC}}(\lambda)$ of size $|\Sigma_{\mathsf{PRC}}(\lambda)| \leq \lceil \frac{C_0}{p_0} \cdot N(\lambda) \rceil$ which has robustness to any $(1 - C_{\mathsf{rob}} p_0)$-edit-bounded channel. Since $1 - C_{\mathsf{rob}} p_0 \geq 1 - \frac{\alpha}{8} + \frac{3p}{\alpha}$ by our choice of $p_0$ and since $p \leq \alpha^2/48$ (which can be ensured by taking $c = 1/48$ in the theorem statement), $\mathsf{PRC}_2$ is robust to any $(1 - \frac{\alpha}{8} + \frac{3p}{\alpha})$-edit-bounded channel.

Finally, by Theorem E.1, as long as $|\Sigma(\lambda)| \geq (\frac{8}{\alpha} |\Sigma_{\mathsf{PRC}}(\lambda)|)^{2/\alpha}$ the watermarking scheme $\mathcal{W} = \mathcal{W}[\mathsf{PRC}_2, \mathsf{Model}]$ is sound, undetectable, and $\beta_\lambda(\ell)$-substring robust to any $p$-edit-bounded channel, for $\beta_\lambda(\ell) := 6\alpha\ell + 8N(\lambda) \leq 6\alpha\ell + n(\lambda)^{O(\log 1/p_0)}$. To ensure that the lower bound on $|\Sigma(\lambda)|$ holds, we note that

$$\left( \frac{8}{\alpha} |\Sigma_{\mathsf{PRC}}(\lambda)| \right)^{2/\alpha} \leq \left( \frac{8}{\alpha} \cdot \left\lceil \frac{C_0}{p_0} \cdot N(\lambda) \right\rceil \right)^{2/\alpha} \leq n(\lambda)^{O(\frac{1}{\alpha} \log \frac{1}{\alpha})},$$

which is bounded above by $n(\lambda)^{C_2 \frac{1}{\alpha} \log \frac{1}{\alpha}} \leq |\Sigma(\lambda)|$ (for sufficiently large $\lambda$) as long as the constant $C_2$ in the theorem statement is sufficiently large.

$\qquad \square$

# F  Technical tools: concentration inequalities

**Theorem F.1** (McDiarmid's inequality). *Let sets $\mathcal{X}_1, \ldots, \mathcal{X}_m$ equipped with sigma algebras be given, and suppose $f : \mathcal{X}_1 \times \cdots \times \mathcal{X}_m \to \mathbb{R}$ satisfies the* bounded differences property, *i.e., for each $j \in [m]$,*

$$\max_{\substack{x_i \in \mathcal{X}_i \ \forall i \in [m], x'_j \in \mathcal{X}_j}} |f(x_j, x_{-j}) - f(x'_j, x_{-j})| \leq c_j,$$

*for some positive real numbers $c_j$. Then given independent random variables $X_i \in \mathcal{X}_i$ ($i \in [m]$), it holds that*

$$\Pr\left(|f(X_1, \ldots, X_m) - \mathbb{E}[f(X_1, \ldots, X_m)]| \geq \epsilon\right) \leq 2 \exp\left(\frac{-2\epsilon^2}{\sum_{i=1}^m c_i^2}\right).$$

**Theorem F.2** (Corollary of Freedman's inequality; see Lemma A.3 of [FKQR23]). *Let $(X_t)_{t \leq T}$ be a sequence of random variables adapted to a filtration $(\mathscr{F}_t)_{t \in [T]}$. If $0 \leq X_t \leq R$ almost surely, then each of the below inequalities holds with probability $1 - \delta$:*

$$\sum_{t=1}^T X_t \leq \frac{3}{2} \sum_{t=1}^T \mathbb{E}_{t-1}[X_t] + 4R \ln(2/\delta)$$

$$\sum_{t=1}^T \mathbb{E}_{t-1}[X_t] \leq 2 \sum_{t=1}^T X_t + 8R \ln(2/\delta),$$

*where $\mathbb{E}_{t-1}[X_t]$ denotes $\mathbb{E}[X_t \mid \mathscr{F}_{t-1}]$.*

## F.1  Concentration with weakly dependent random variables

Fix a set $\mathcal{X}$ equipped with a sigma-algebra together with a positive integer $n$. Let $P \in \Delta(\mathcal{X}^n)$ denote a distribution (where $\mathcal{X}^n$ is equipped with the product sigma algebra). For $X = (X_1, \ldots, X_n) \sim P$, we define the *influence of $X_j$ on $X_i$* to be

$$I_{j \to i}(X) := \max_{\substack{x_{-i-j} \in \mathcal{X}^{n-2} \\ x_j, x'_j \in \mathcal{X}}} d_{\mathsf{TV}}(P(X_i = \cdot \mid X_{-i-j} = x_{-i-j}, X_j = x_j), P(X_i = \cdot \mid X_{-i-j} = x_{-i-j}, X_j = x'_j)).$$

$$(43)$$

**Definition F.1.** Given a random variable $X$ distributed according to some distribution $P$ over $\mathcal{X}^n$, define the *Dobrushin coefficient* of $X$ to be

$$\alpha(X) := \max\left\{\max_{j \in [n]} \sum_{i \neq j} I_{j \to i}(X), \max_{i \in [n]} \sum_{j \neq i} I_{j \to i}(X)\right\}.$$

**Theorem F.3** (Dobrushin's concentration inequality; e.g., Theorem 4.3 of [Cha16]). *Let $P$ be a distribution over $\mathcal{X}^n$ whose Dobrushin coefficient is $\alpha$. Let $f : \mathcal{X}^n \to \mathbb{R}$ be a real-valued function satisfying the following* bounded differences condition, *for some constants $c_1, \ldots, c_n \geq 0$:*

$$\max_{(x_1, \ldots, x_n) \in \mathcal{X}^n, x'_j \in \mathcal{X}} |f(x_j, x_{-j}) - f(x'_j, x_{-j})| \leq c_j.$$

*Then for all $\epsilon > 0$,*

$$\Pr_{X \sim P}\left(|f(X) - \mathbb{E}[f(X)]| \geq \epsilon\right) \leq 2 \exp\left(-\frac{(1-\alpha)\epsilon^2}{2\sum_{i=1}^n c_i^2}\right).$$

# G  Miscellaneous Lemmas

**Lemma G.1.** *Suppose a PRC* $\mathsf{PRC} = (\mathsf{KeyGen}, \mathsf{Encode}, \mathsf{Decode})$ *is robust to any $p$-substitution-bounded channel. Then for any channel $\mathcal{E}$ satisfying $\Pr_{\mathsf{sk} \sim \mathsf{KeyGen}(1^\lambda), x \leftarrow \mathsf{Encode}(1^\lambda, \mathsf{sk}, \mathsf{m}), y \sim \mathcal{E}(x)}(D_{\mathsf{Ham}}(x, y) \leq pn) \geq 1 - \mathsf{negl}(n)$, PRC is robust to $\mathcal{E}$.*

*Proof.* Let $\mathcal{E}'$ be the channel which samples $y \sim \mathcal{E}(x)$, outputs $y$ if $D_{\mathsf{Ham}}(y, x) \leq pn$, and otherwise outputs $x$. By assumption PRC is robust to $\mathcal{E}'$. Moreover, we can construct a joint distribution of $(\mathsf{sk}, x, y, y')$ where: (a) the marginal of $(\mathsf{sk}, x, y)$ is as follows: $\mathsf{sk} \sim \mathsf{KeyGen}(1^\lambda), x \leftarrow \mathsf{Encode}(1^\lambda, \mathsf{sk}, \mathsf{m}), y \sim \mathcal{E}(x)$; (b) the marginal of $(\mathsf{sk}, x, y')$ is identical except that $y' \sim \mathcal{E}'(x)$; (c) $y = y'$ with probability $1 - \mathsf{negl}(\lambda)$. Then the robustness criterion for $\mathcal{E}'$ yields that PRC is robust to $\mathcal{E}$. $\qquad\square$

**Lemma G.2** (Theorem 1 of [RS23]). *For any positive integers $t, k, N$ with $k \leq N$ and $t \leq N$, it holds that $d_{\mathsf{TV}}(\mathrm{Bin}(t, k/N), \mathrm{Hyp}(N, k, t)) \leq \frac{2t}{\sqrt{N-t}}$.*

For completeness, we provide the proof of Lemma G.2 below.

*Proof of Lemma G.2.* Fix $t, k, N$ as in the lemma statement and let $\mathbb{P} = \mathrm{Bin}(t, k/N) \in \Delta(\{0, 1, \ldots, t\})$ and $\mathbb{Q} = \mathrm{Hyp}(N, k, t) \in \Delta(\{0, 1, \ldots, t\})$. Also write $p = k/N$. Then we have $\mathbb{P}(w) = \binom{t}{w} \cdot p^w (1 - p)^{t-w}$ and

$$
\mathbb{Q}(w) = \frac{\binom{k}{w}\binom{N-k}{t-w}}{\binom{N}{t}} = \binom{t}{w} \cdot \frac{(N-k)\cdots(N-k-(t-w)+1) \cdot k \cdots (k-w+1)}{N \cdots (N-t+1)}
$$
$$
= \binom{t}{w} \cdot \left(\frac{k}{N}\right)^w \cdot \frac{\prod_{j=0}^{t-w-1}(1 - p - j/N) \cdot \prod_{j=0}^{w-1}(1 - j/k)}{\prod_{j=0}^{t-1}(1 - j/N)}.
$$

Then

$$
\frac{\mathbb{Q}(w)}{\mathbb{P}(w)} = \frac{\prod_{j=0}^{t-w-1}(1 - p - j/N) \cdot \prod_{j=0}^{w-1}(1 - j/k)}{(1 - p)^{t-w} \prod_{j=0}^{t-1}(1 - j/N)} \leq \prod_{j=0}^{t-1} \frac{1}{1 - j/N}, \tag{44}
$$

for each $0 \leq w \leq t$. By Pinsker's inequality,

$$
\begin{aligned}
d_{\mathsf{TV}}(\mathbb{Q}, \mathbb{P}) &\leq 2\sqrt{\sum_{w=0}^{t} \mathbb{Q}(w) \cdot \ln \frac{\mathbb{Q}(w)}{\mathbb{P}(w)}} \\
&\leq 2\sqrt{\ln \prod_{j=0}^{t-1} \frac{1}{1 - j/N}} \\
&= 2\sqrt{\sum_{j=0}^{t-1} \ln \frac{1}{1 - j/N}} \\
&\leq 2\sqrt{\sum_{j=0}^{t-1} \frac{j}{N - j}} \\
&\leq 2\sqrt{\frac{t^2}{N - t}},
\end{aligned}
$$

where the second inequality uses (44), and the third inequality uses the fact that $\ln \frac{1}{1-x} \leq \frac{1}{(1/x)-1}$ for all $x \in [0, 1]$. $\qquad\square$

