# OpenReview forum: "Edit Distance Robust Watermarks via Indexing Pseudorandom Codes"
_NeurIPS.cc/2024/Conference — NeurIPS 2024 poster_

### Official Review · Reviewer_Qm1S · 2024-07-11

**Soundness:** 3
**Presentation:** 2
**Contribution:** 4
**Rating:** 8
**Confidence:** 3

**Summary:**

This paper constructs an LLM watermarking scheme which is robust to edit distance perturbations. The construction is carried out carefully in stages. The authors first construct pseudorandom codes (PRC) that are robust to constant fraction substitutions assuming the existence of local weak pseudorandom functions (PRF) (see below for an informal definition of PRFs). Here, PRCs refer to a keyed encoding and decoding scheme $(\mathrm{Enc}_s, \mathrm{Dec}_s)$ that satisfies the following three criteria. Note that for watermarking, the only “message” that needs to be encoded is “this model is watermarked”, which the authors denote by $\emptyset$.

1. **Undetectability.** For poly-time oracle Turing machines, the encoding oracle $\mathrm{Enc}_s: {\emptyset} \to \\{0,1\\}^n$, which is a *random* function that outputs n-bit strings correlated with a *random* secret key s (unknown to the Turing machine), is indistinguishable from a random oracle that outputs purely random n-bit strings.
2. **Soundness.** For any *fixed* n-bit string y, $\Pr_{s}[\mathrm{Dec}_s(y) = \perp] \ge 1-\mathrm{negl}(n)$ (symbol representing NOT watermarked). In words, any fixed string is rejected by the decoder with all but negligible probability. Hence, a random n-bit string y drawn from the uniform distribution (and independently of the secret key s) will be rejected by $\mathrm{Dec}_s$ with high probability.
3. **Robustness.** Denoting by $\mathcal{E} : \\{0,1\\}^n \to \\{0,1\\}^n$ any corruption channel restricted to at most constant fraction substitutions. $\Pr[ \mathrm{Dec}_s(\mathcal{E}(y)) = \emptyset \mid y \leftarrow \mathrm{Enc}_s(\emptyset)] \ge 1-\mathrm{negl}(n)$.

The key property in this PRC construction is its robustness, which follows (non-trivially) from the use of local weak PRFs. Intuitively, *local* PRFs that depend only on log(n) bits are just the right primitives for constant-fraction perturbations. For example, if the string y is corrupted at random coordinates, most of the significant bits of the PRF evade corruption with high probability.

However, the above PRC is only robust to substitution perturbations, not robust to edit distance perturbations. This brings us to their second stage, which is a generic transformation that takes the above PRC to *index* PRCs. The idea behind index PRCs is simple. Because corruption in edit distance can *delete* coordinates of the given string y, we need to repeat its content in some way to protect against deletions. Given a string, say 1001 one way to represent it is through the indices of its support, {1, 4}. After all, n-bit strings are in one-to-one correspondence with subsets of [n]. Thus, one can repeat the “contents” of the given string y, by cleverly storing and repeating the “indices” of y. Significantly expanding on this observation, the authors construct a PRC that is robust to constant fraction edit distance perturbations. The caveat, of interest to practitioners, is that this PRC requires the alphabet size to increase polynomially with the security parameter, a core parameter controlling undetectability (so quality of the watermarked model), soundness, and robustness.

Based on their careful PRC constructions, the paper culminates by constructing a watermarking scheme for LLMs (i.e., general discrete autoregressive distributions) whose watermark can be robustly and reliably detected if the generated text contains a high empirical entropy substring.

**Local pseudorandom functions.** Local PRFs are PRFs which depend only on few bits of the source randomness. That is, for each secret key s, there exists a support set $J \subset [n]$ such that $F_s(x) = G_s(x_J)$, where $F_s : \\{0,1\\}^n \to \\{0,1\\}$, and $G_s : \\{0,1\\}^\ell \to \\{0,1\\}$. For robustness against corruptions to a constant fraction of the coordinates, the authors require $\ell = \log n$. It seems that the locality requirement is incompatible with “strong” PRFs in which adversary (an oracle Turing machine that attempts to distinguish the PRF oracle and the random oracle) has query access. This is a non-issue in this work since the authors use weak PRFs, in which the adversary only has random sample access to the oracles.

**Strengths:**

The contributions of this paper are significant both theoretically and practically. It is rare for a purely theoretical work, which is aesthetically and technically pleasing in its own right, to directly address an important practical problem of wide interest. I believe this paper has the potential to achieve award-worthy quality if its weaknesses are addressed.

**Addresses a practically important ML question.** The widespread influence of LLMs like ChatGPT on our daily lives has raised numerous concerns and has spurred recent interest in watermarking schemes for LLMs. While there have been several recent works on LLM watermarking, this work is the first to achieve *constant fraction* edit-distance robust watermarking schemes with *reasonable* storage requirements for the secret key. Moreover, the authors achieve this by carefully constructing highly non-trivial theoretical objects (PRCs w.r.t. edit distance) which may have other exciting theoretical applications.

**Technical novelty and significance.** The construction of PRCs based on local weak PRFs, and the generic transformation to index PRCs which are robust to edit distance perturbations is novel. In particular, the connection between PRCs and local weak PRFs seems to be novel, but they seem particularly well-suited for each other. Local weak PRFs are insensitive to perturbations in most coordinates of $(x, F_s(x))$, which is exactly what is needed for the robustness of PRCs.

The techniques involved in analyzing these schemes are non-trivial since the corruption channels can perturb *adversarially* (but restricted to at most p corruptions). Moreover, the test statistic which is used to test the existence of watermarks is a sum of (weakly) dependent terms to which standard concentration inequalities do not apply. The authors resolve such technical issues by designing the encoding scheme, using additional randomness, to mitigate the power of the perturbation adversary and using recently developed concentration inequalities that apply to weakly dependent random variables.

**Weaknesses:**

While the paper makes substantial technical contributions, its presentation falls short.

- **Insufficiently polished exposition on application to LLM watermarking (Section 5).** While the paper does a great job in presenting the PRC constructions, the culminating application to LLM watermarking in Section 5 is presented in a very rushed way. The "full detail" presentation in Appendix E is hard to follow as well. The key issue seems to be the lack of separation between the high-level intuition and the technical complications, as well as the poor choice of notation. Theorem statements in cryptography often resemble an alphabet soup, so careful and clear notation is crucial. A few suggestions are:

  1. Start Section 5 by explaining the basic idea that connects PRCs and watermarking schemes. The basic idea is that for each token generation, we first sample $y \leftarrow \Sigma$ from the PRC alphabet $\Sigma$, and use rejection sampling with the condition $\phi(v) = y$ to sample the next token $v$, where $\phi : \mathcal{V} \to \Sigma$ is a hash map.
  2. In Algorithm 3, using $p_i$ to denote a *distribution* when $p_0$ and $p$ have already been used to denote constants in $(0,1)$ is confusing. Perhaps $\mu_i$ might be less confusing?
  3. The cluttered subscript notation under the $\Pr$ sign is both misleading and hard to parse. It's misleading because it does not fully specify all the randomness. For example, in Eq. (1) (page 5, line 185) the randomness in $\Pr$ not only depends on the $\mathsf{sk} \leftarrow \mathsf{KeyGen}$, but also on the internal randomness of the "adversary". It might be less confusing to enumerate all elements of randomness as was done in Appendix C.3 (page 20, line 811). In addition, expressions like Eq. (37) in Claim E.10 are quite overwhelming.

- **Lack of formal description of the perturbation channel.** There seem to be two unrelated "adversaries" to the watermarking scheme. The "detection adversary" which is a poly-time oracle Turing machine that attempts to detect the watermark, and the "perturbation adversary", i.e., the perturbation channel, which is a (potentially randomized) function that perturbs the generated text. It might be worth highlighting that the modeled perturbation channel is *static*. That is, the same (potentially randomized) function is applied to the generated text and robustness only holds w.r.t. this *static* adversary. This seems to be an important distinction since robustness against a *dynamic* perturbation adversary which interacts with the watermarking scheme over many rounds appears to be impossible, as the authors note (page 1, line 34-35).
> *it is necessary to strike a balance in terms of the power of adversaries to which a watermarking scheme enjoys robustness*

- **Contextualizing the local weak PRF assumption.** It's not clear to me why existence of local weak PRFs is weaker than the existence of public-key cryptography. Is there a formal reduction showing that public-key cryptography implies local weak PRFs? I don't think the current explanation based on Impagliazzo's worlds is adequate. Without context, it's unclear what being in Minicrypt or Cryptomania even means. Is there a single "complete" primitive that generates all cryptographic tools in each world? Do the worlds form a strictly increasing sequence (in set inclusion)?

**Minor comments**
- In Section 4, when introducing *indexing* PRCs, it might be helpful to include a diagram demonstrating the "index" encoding. For example, one can draw a diagram showing that 100101 maps to {1, 4, 6}.

- The hyperref for "Line 8" in (page 6, line 248) is incorrect. It should link to Line 8 in Algorithm 2, not Algorithm 1.

**Questions:**

- It seems that the locality requirement is incompatible with “strong” PRFs in which adversary (an oracle Turing machine that attempts to distinguish the PRF oracle and the random oracle) has query access. For example, if an adversary has query access, then it can query the all zeros string and its 1-Hamming neighbors. Then, the adversary can decide that the unknown oracle is a PRF If the label bit is significantly insensitive to these 1-bit perturbations. Is this the right intuition?

- Is there an object analogous to a local weak PRF for continuous domains? Local PRFs seem to be the right objects for countering "sparse" perturbation channels which affect only a few coordinates of the sample. What would be appropriate for perturbation channels in continuous domains (e.g., $\ell_2$ bounded perturbations)?

---

> ### Author Rebuttal · Authors · 2024-08-06
>
> Thank you for your positive comments. Regarding your comments on presentation: we will adjust our exposition accordingly. Regarding local weak PRFs, we do not know of a formal reduction showing that public-key cryptography implies local weak PRFs. We will adjust the discussion to add some additional background from reference [30] (Impagliazzo, 1995) on the meaning of the five worlds (which is somewhat informal, as mentioned on p.3 of Impagliazzo's paper).
>
> Question 1: Yes, the locality requirement is incompatible with strong PRFs which give the adversary query access, since $\log(n)$-juntas can be learned in poly(n) time (See [BL97] below). Roughly speaking, the idea of this result is that for a $\log(n)$-junta $F(\cdot)$, an adversary finds x,y so that $F(x) \neq F(y)$, then queries vertices on a path in the hypercube between x and y, which yields at least 1 of the $\log(n)$ influential coordinates. It then recurses to find all influential coordinates. It can then learn $F$ by trying all possible n combinations of the $\log(n)$ influential coordinates.
>
> [BL97] "A. Blum and P. Langley. Selection of relevant features and examples in machine learning, 1997"
>
> Question 2: $\ell_2$ perturbations could be reasonable, though it is unclear if local PRFs can be used to achieve robustness to such perturbations. This is an interesting direction for future work.

---

> ### Comment · Reviewer_Qm1S · 2024-08-09
>
> Thank you for addressing my questions and being receptive. I am quite excited by the results and would be happy to see this paper accepted.
>
> Additionally, I believe that the theoretical merits of this paper alone justify its acceptance. The proposed watermarking scheme is has negligible Type-1 error, *provably* robust in edit distance, and satisfies "undetectability", which means that the watermarked model is *computationally indistinguishable* from the original model. The fact that we can satisfy all three criteria (soundness, edit distance robustness, and undetectability) is already surprising and highly non-trivial.

---

### Official Review · Reviewer_zeWE · 2024-07-11

**Soundness:** 3
**Presentation:** 2
**Contribution:** 3
**Rating:** 5
**Confidence:** 4

**Summary:**

This paper addresses the challenge of watermarking the outputs of language models with provable guarantees of undetectability and robustness to adversarial edits. The authors propose a novel watermarking scheme that maintains these properties even when subjected to a constant fraction of insertions, deletions, and substitutions. This work builds upon previous schemes that could handle only stochastic substitutions and deletions, thereby providing a more comprehensive solution. The key contribution is the development of pseudorandom codes (PRCs) that are robust against such edits, achieved through an indexing approach and relying on weaker computational assumptions. The paper demonstrates the theoretical foundations of this scheme and outlines its potential applications in preventing misuse of AI-generated content.

**Strengths:**

1. The paper introduces a new watermarking scheme, which can handle a broader range of adversarial edits, including insertions and deletions.
2. The authors provide a thorough theoretical analysis and proof of the robustness and undetectability of their proposed scheme, grounded in cryptographic principles.

**Weaknesses:**

1. The theoretical nature of the work might pose challenges in practical implementation, especially concerning the scalability and efficiency of the proposed watermarking scheme.
2. The paper lacks empirical evaluation of the proposed watermarking scheme. Without experiment data, it is very hard to assess the effectiveness of the proposed method in realistic scenarios.
3. The requirement for a constant entropy rate in the generated text might not be met by all language models, potentially limiting the scheme's effectiveness in low-entropy contexts.

**Questions:**

1. What are the specific steps required to implement this watermarking scheme in existing language models, and how does it impact the model's performance in terms of speed and resource consumption?
2. How does the proposed scheme perform against paraphrase and translation attacks?
3. Watermark stealing attack has been proposed recently [1,2,3], which can infer the parameters of the watermarking scheme and remove the watermark from the text. How does the proposed scheme perform against watermark stealing attack?
[1] N. Jovanović, R. Staab, and M. Vechev, “Watermark Stealing in Large Language Models.” http://arxiv.org/abs/2402.19361
[2] Q. Wu and V. Chandrasekaran, “Bypassing LLM Watermarks with Color-Aware Substitutions.” http://arxiv.org/abs/2403.14719
[3] Z. Zhang et al., “Large Language Model Watermark Stealing With Mixed Integer Programming.” http://arxiv.org/abs/2405.19677

**Limitations:**

1. The scheme's scalability to large-scale language models and extensive datasets remains untested, which might be a critical factor in practical applications.
2. The requirement for a large alphabet size might necessitate modifications in existing tokenization schemes, posing a barrier to seamless integration.
3. The effectiveness of the watermarking scheme is contingent upon the entropy rate of the generated text, which might not be uniformly high across different language models and applications.
4. While the theoretical foundations are robust, empirical validation through extensive experimentation on various language models and datasets is needed to ascertain the scheme's practical efficacy and resilience to adversarial attacks.

---

> ### Author Rebuttal · Authors · 2024-08-06
>
> Thank you for your review.
>
> Many of the weaknesses and limitations you mention relate to the lack of implementation and experiments. We want to emphasize that developing a fully practical watermarking scheme for immediate use in LLMs is not the point of this paper. Rather, the purpose is to lay the theoretical foundations for a technique to achieve edit-distance robust undetectable watermarking (for which there are not yet any practical schemes), and that additional work is needed to make our scheme fully practical.
>
> Question 1: The computational cost of generating each next token for our watermarking scheme is linear in the alphabet size of the LLM. Note that it takes linear in alphabet size time to even generate non-watermarked output. We expect that the actual empirical (constant factor) blowup in computational cost will be quite small due to the relative simplicity of our procedure. That being said, we remark that our scheme requires a large alphabet size, which poses challenges for a direct implementation of it on existing LLMs; see the response to Question 1 of Reviewer YwkB for further details on this point.
>
> Question 2: If the paraphrase attack produces text which is close to the original in edit distance (i.e., if it is not an overly aggressive paraphrasing), then our scheme would still detect paraphrased watermarked text. To deal with a translation attack, one could augment our detection algorithm to translate its input text into many common languages and output True if any of the resulting translations is detected as watermarked. This would overcome the translation attack assuming that translating a string x from one language to another and back leads to a string x' which is close to x in edit distance, which we expect to be a reasonable assumption.
>
> Question 3: Thank you for pointing out the papers on watermark stealing attacks. While ruling out such attacks is beyond the scope of our paper, we are hopeful that our PRCs and watermarking schemes (as well as those of [Christ & Gunn, 2024]) are resistant to such attacks. Roughly speaking, we believe that it might be possible to show this using the relatively strong cryptographic properties of local weak PRFs (or, in the case of [Christ & Gunn, 2024], the relatively strong cryptographic properties of the noisy parity problem). Establishing such robustness to stealing attacks is an interesting direction for future work, even for the case of subsitution PRCs as in [Christ & Gunn, 2024].

---

> > ### Comment · Reviewer_zeWE · 2024-08-09
> >
> > Thank you for answering my question. I would like to raise my score due to your solid theoretical analysis. However, I still believe that including empirical validation in your paper would enhance its overall strength.

---

### Official Review · Reviewer_ab11 · 2024-07-12

**Soundness:** 3
**Presentation:** 3
**Contribution:** 4
**Rating:** 8
**Confidence:** 2

**Summary:**

This paper proposes a new pseudorandom code (PRC) called an _indexing PRC_ over a polynomially-sized alphabet that is robust to a constant number of adversarial edit corruptions (insertions, deletions or substitutions). It is constructed as a wrapper around a substitution-robust PRC, which has been studied in prior work. The paper also proposes a new substitution-robust PRC that relies on weaker cryptographic hardness assumptions than prior work. These cryptographic primitives are then used to construct a watermarking scheme for autoregressive models (e.g. LLMs), which is proven to satisfy three important properties: undetectability, soundness and edit robustness.

**Strengths:**

**Originality and significance:**
The paper builds on recent work by Christ & Gunn (2024), who developed pseudorandom codes (PRCs) over binary alphabets that are robust to adversarial substitutions and limited deletions. This work makes several advances in PRCs, which enables an even more compelling application to watermarking of autoregressive models:
- A new substitution-robust PRC is proposed based on local weak pseudorandom functions that follows from weaker average-case hardness assumptions than Christ & Gunn’s PRC
- A new indexing PRC is proposed that is robust to a constant fraction of adversarial edits
- The indexing PRC is used to develop a watermarking scheme for autoregressive models with large alphabets (whereas Christ & Gunn considered a model with a binary alphabet).

**Clarity:** The authors have done a good job of distilling quite complex/technical work into 9 pages. Despite not having a background in cryptography, I found the paper reasonably accessible. I appreciated the summary of the main results on p. 3-4.

**Weaknesses:**

1. While the paper is motivated by watermarking of generative models, most of the paper’s contributions are in cryptography. Roughly two-thirds of the paper covers pseudorandom codes, while the final connection to watermarking (Section 5) consumes only half a page. I wonder whether another venue would be a better fit for the work – both in terms of the audience and the ability of reviewers to assess correctness.

1. After having read the paper, it’s not clear to me whether the proposed watermarking scheme can be implemented in practice or not. I think the work is valuable whether the scheme is practical or not, however it’s important to be upfront about practical limitations (if there are any) to help guide future work.

1. Related to the above point about practicality: there is no empirical evaluation of the watermarking scheme. This is not a major issue in my view, as the paper makes very strong theoretical contributions.

1. I wonder whether edit robustness is a desirable property for a watermarking scheme. Consider the case of a vendor who is concerned about safety. An adversary could query the vendor’s model, receive safe watermarked output and then substantially edit the watermarked output to make it unsafe. The watermark is preserved in the unsafe output since it is edit robust, provided the number of edits falls below some threshold. This could be problematic for the vendor’s reputation, as the adversary can assert that the vendor’s model generated unsafe content.

**Minor:**
- eqn (2): Should the dimension of s be $\ell(\lambda)$ rather than $n(\lambda)$? Inconsistency between two terms: one is an expectation over an indicator function, whereas the other is an expectation over the output of $\widetilde{\mathrm{Adv}}^{F_\mathrm{Unif}}$.
- line 217: Output of $\widetilde{\mathrm{Adv}}^G$ is undefined
- line 222: Call to $F_s(\cdot)$ returns a tuple in the space $\\{0, 1\\}^n \times \\{0, 1\\}$, which conflicts with the return type given in line 214.
- line 244: Should $z \in$ be $z \sim$?

**Questions:**

- Are there any barriers to instantiating the proposed watermarking scheme?
- Is edit distance robustness sometimes undesirable for watermarking?

**Limitations:**

Yes, limitations are discussed on p. 3.

---

> ### Author Rebuttal · Authors · 2024-08-06
>
> Thank you for your helpful comments.
>
> Minor questions:
> - Eq. (2): Yes, dimension of $s$ should be $\ell(\lambda)$. And yes, we forgot an "=1" in the second term.
> - Line 217: $\widetilde{Adv}^G$ refers to the output of the algorithm Adv (which is a 0 or 1) when, each time Adv decides to query G, it receives $(x, G(x))$ for a uniformly random $x$.
> - Line 222: Thanks, we will clarify that the oracle returns just the output of the function $F_s(x) \in \{0,1\}$ (though of course Adv knows the random input $x$ as well).
> - Line 244: Yes, should be $z \sim$.
>
> Questions:
>
> 1. Using techniques from our theoretical watermarking scheme to develop a practical implementation for use with LLMs is an important direction for future work. The main limitation of our present theoretical results which may complicate a practical implementation is as follows: The alphabet size $|\Sigma(\lambda)|$ is required to grow exponentially in the inverse of the parameter $\alpha$ (see the statement of Theorem E.2). In turn, the parameter $\alpha$ is proportional to the entropy rate of the text needed to guarantee substring robustness (see Definition 2.6 and the setting of $\beta_\lambda(\ell) = O(\alpha \cdot \ell)$ in Theorem E.2). For typical LLMs, the alphabet size is likely smaller than our required value of $|\Sigma(\lambda)|$ given the entropy rates observed empirically in natural language.
> On the other hand, we believe that future work aimed at developing modifications of our watermarking scheme with an eye towards practical implementation will be successful. One idea which seems promising is to simulate a larger alphabet by grouping tokens together, and to aim accordingly for a slightly weaker robustness guarantee.
>
> 2. It sounds like you are asking about using watermarking for public attribution. It is indeed correct that the same "Detect" function cannot be used for both (a) detecting watermarked output robustly, and (b) as a sort of signature scheme to attribute text to a certain language model. Section 7.4 of [Christ & Gunn, 2024] proposes a solution to this dilemma by constructing watermarking schemes with "unforgeable public attribution": their scheme has an Attribution function, which is not robust and indicates when a portion of its input text is copied verbatim from the model, as well as a Detect function, which is robust and indicates when a portion of its input is edit-distance near to a string output by the model. The key point is that a "True" output of Detect should not be interpreted as attributing the text to the model.

---

> > ### Comment · Reviewer_ab11 · 2024-08-08
> >
> > Thank you for responding to my questions. I continue to advocate for acceptance, as the paper provides strong theoretical contributions, even though there are practical limitations. I recommend addressing Q1 more explicitly in the paper, as it's likely to be a concern for many in the NeurIPS community (three of the four reviewers asked about it).

---

> > > ### Author Response · Authors · 2024-08-09
> > >
> > > Thank you! Yes, we will update the paper to address Q1 more explicitly.

---

### Official Review · Reviewer_YwkB · 2024-07-13

**Soundness:** 3
**Presentation:** 3
**Contribution:** 3
**Rating:** 6
**Confidence:** 4

**Summary:**

The paper presents an innovative approach to embed an undetectable and robust watermark into AI-generated texts. The authors takes three main steps to reach the goal. They first design a robust pseudorandom code (PRC) over a binary alphabet, and then turn it into a polynomial-sized alphabet PRC robust to a fraction of edits. Finally, they bridge the PRC and the LLM watermark using some carefully designed mapping function to complement the whole watermarking system. The proposed approach is theoretically proved to be undetectable and robust to a constant fraction of edit attacks.

**Strengths:**

The paper addresses a critical need for theoretically robust LLM watermark.
Originality: Using a designed mapping function to bridge the gap between PRC and LLM watermark and proposing the theoretical robustness is innovative.
Quality: The methodology is rigorously developed.
Clarity: The paper is well-structured and the writing is clear, making the complex concepts accessible.
Significance: The work has the potential to significantly impact the field of detecting AI-generated text.

**Weaknesses:**

1. Lack of real-world evaluation of the proposed approach.
2. Lack of efficiency comparison with the current watermarking framework.
3. Lack of discussion of trade-off between undetectability and robustness.
4. Lack of scalability.

**Questions:**

1. Have the authors considered evaluating the proposed method on real-world scenario to better understand its practical applicability?
2. How does the computational efficiency of the proposed method compare to existing watermarking framework?
3. Is there a trade-off between undetectability and robustness? For example, in Lemma D.8, the robustness is achieved at very low $p_0$ (since $C_{\text{rob}}$ should be a large constant value). Will undetectability be affected by low values of $p_0$? If the trade-off exists, how is the trade-off reflected in the theorems of the paper？
4. What are the potential limitations or challenges in scaling the proposed method up for large-scale NLP tasks?

**Limitations:**

Lack of evaluation in real scenario: the authors did not show any experiment done on real LLM. Testing in the real world would provide a better assessment of the method's practical applicability and robustness. Adding some real example of watermarked AI-generated text will signify the problem of undetectability. Showing the real accuracy of detection will bridge the gap between the theory and the practice.
This paper considers "constant-rate" insertion/deletion/substitution attacker model, how about attackers who perform paraphrasing or adaptive attacks？Is the proposed method robust to them?

---

> ### Author Rebuttal · Authors · 2024-08-06
>
> Thank you for your review.
>
> Question 1: Using techniques from our theoretical watermarking scheme to develop a practical implementation for use with LLMs is an important direction for future work. The main limitation of our present theoretical results which may complicate a practical implementation is as follows: The alphabet size $|\Sigma(\lambda)|$ is required to grow exponentially in the inverse of the parameter $\alpha$ (see the statement of Theorem E.2). In turn, the parameter $\alpha$ is proportional to the entropy rate of the text needed to guarantee substring robustness (see Definition 2.6 and the setting of $\beta_\lambda(\ell) = O(\alpha \cdot \ell)$ in Theorem E.2). For typical LLMs, the alphabet size is likely smaller than our required value of $|\Sigma(\lambda)|$ given the entropy rates observed empirically in natural language.
>
> On the other hand, we believe that future work aimed at developing modifications of our watermarking scheme with an eye towards practical implementation will be successful. One idea which seems promising is to simulate a larger alphabet by grouping tokens together, and to aim accordingly for a slightly weaker robustness guarantee.
>
> Question 2: Modulo the limitations discussed in Question 1, our watermarking scheme is very efficient (and comparable to, e.g., that of [Christ & Gunn, 2024]). In particular, the computational cost of generating each next token is linear in the alphabet size of the LLM. Note that it takes linear in alphabet size time to even generate non-watermarked output.
>
> Question 3: Yes, there is a joint tradeoff between undetectability, robustness, and blocklength of the watermark, even for substitution PRCs (i.e., Theorem 3.2), which then carries over to our edit-distance robust PRCs, as in Lemma D.8. The precise tradeoff is reflected in Equation (3) in the proof of Theorem 3.2: we show in the proof that the blocklength parameter $N(\lambda)$ to achieve a *fixed* undetectability guarantee must grow exponentially in $\log(1/(1-2p))$ through its dependence on $m(\lambda)$. In other words, keeping the block length fixed, taking $p \to 1/2$ will lead to a weaker undetectability guarantee. This does not manifest itself in the theorem statements since we always think of $p$ as a constant and so values of $p$ closer to $1/2$ result in a polynomially weaker undetectability guarantee for fixed blocklength, which is insignificant compared to the superpolynomial guarantee of undetectability (and we do not distinguish between different polynomials). We remark that the results of [Christ & Gunn, 2024] have essentially the same tradeoff; please see the end of Section E.1 for further discussion on this point.
>
> Question 4: The main challenge lies in handling the tension between the fact that in many large-scale NLP tasks, the entropy rate is relatively low and the resulting alphabet size that our results require would be too large compared to the number of tokens. We believe that there are promising techniques which will be able to alleviate these limitations in future work for practical applications; see Question 1 for more details.

---

> > ### Comment · Reviewer_YwkB · 2024-08-12
> >
> > We thank the authors for the response and decide to keep the score.

---

### Decision · Program_Chairs · 2024-09-25

**Decision:**

Accept (poster)

**Comment:**

The paper presents a novel method to embed an undetectable and robust watermark into LLM-generated texts. The main idea is a new pseudorandom code (PRC) called an indexing PRC over a polynomially-sized alphabet that is robust to a constant number of adversarial edit corruptions (insertions, deletions, or substitutions). The PRC is then integrated into an LLM watermark using a carefully designed mapping. The authors further prove that the resulting watermark is w.h.p. undetectable and is robust to a constant fraction of edit attacks.

All the reviewers have agreed that the paper provides a novel set of contributions and a novel watermarking methodology. While I recommend the paper to be accepted, I would like to suggest to the authors to do their best to further address the comments/limitations raised by the reviewers (e.g. evaluation, scalability, etc -- please see the reviews).